**Data Availability Statement:** Data are available on OSF with the following DOI: 10.17605/OSF.IO/SN8PJ.

**Funding:** E.A.C. was supported by the FRS-F.N.R.S (Belgium). A.D.B. was supported by the Research

# How using brain-machine interfaces influences the human sense of agency

**Emilie A. Caspar**[ID]¹☉*, **Albert De Beir**²,³☉, **Gil Lauwers**², **Axel Cleeremans**¹‡, **Bram Vanderborght**²,³‡

**1** CO3 lab, Center for Research in Cognition and Neuroscience, Université libre de Bruxelles, Brussels, Belgium, **2** Vrij Universiteit Brussels, Brussels, Belgium, **3** Flanders Make, Lommel, Belgium

☉ These authors contributed equally to this work.
‡ These authors also contributed equally to this work.
* ecaspar@ulb.ac.be

## Abstract

Brain-machine interfaces (BMI) allows individuals to control an external device by controlling their own brain activity, without requiring bodily or muscle movements. Performing voluntary movements is associated with the experience of agency ("sense of agency") over those movements and their outcomes. When people voluntarily control a BMI, they should likewise experience a sense of agency. However, using a BMI to act presents several differences compared to normal movements. In particular, BMIs lack sensorimotor feedback, afford lower controllability and are associated with increased cognitive fatigue. Here, we explored how these different factors influence the sense of agency across two studies in which participants learned to control a robotic hand through motor imagery decoded online through electroencephalography. We observed that the lack of sensorimotor information when using a BMI did not appear to influence the sense of agency. We further observed that experiencing lower control over the BMI reduced the sense of agency. Finally, we observed that the better participants controlled the BMI, the greater was the appropriation of the robotic hand, as measured by body-ownership and agency scores. Results are discussed based on existing theories on the sense of agency in light of the importance of BMI technology for patients using prosthetic limbs.

## Introduction

Most of our actions are automatic and triggered externally. Voluntary actions, however, because they are initiated endogenously on the basis of our intentions, are seen as a fundamental marker of human behaviour. Such goal-oriented behaviour is associated with the 'sense of agency,' that is, the subjective feeling that one exerts control over one's own actions and on their outcomes [1].

The sense of agency (SoA) is not a unitary phenomenon. SoA involves different aspects of the conscious experience of being the agent of one's actions; in particular feelings of agency or authorship (FoA), and judgments of agency (JoA) [2]. FoA refers to a pre-reflective sensorimotor experience of being the author of an action, while JoA refers to the explicit declaration that

Foundation Flanders (FWO) under grant number G026214N. This work was partially supported by the Flemish Government under the program "Onderzoeksprogramma Artificiële Inelligentie (AI) Vlaanderen". A.C. is a research director with the F.R.S.-FNRS (Belgium). This work was partially supported by ERC Advanced Grant #340718 "Radical" to A.C.

**Competing interests:** The authors declared that they had no conflicts of interest with respect to their authorship or the publication of this article.

an outcome was (or not) caused by our own actions. JoA is typically measured through explicit questions asked to the participants (e.g. *Are you the author of that action*?), while FoA is typically measured more implicitly, for instance by using time perception (i.e., the intentional binding paradigm). In the classical intentional binding paradigm [3], participants estimate the delay between their action (i.e. a keypress) and an outcome (i.e. a tone) by judging the moment at which each event occurs by means of a clock-like representation [4]. If the movement is performed voluntarily, the perceived time between the two events is reported to be shorter than in a condition in which the movement is performed involuntarily (for instance, when it is triggered by a TMS pulse over the motor cortex). This suggests that sense of agency modifies time perception by reducing the perceived duration of the delay that lapses between an action that we feel we carried out (vs. not) and its consequences (see [5–7] for reviews and [8]).

Today, many actions that we carry out are mediated by computers and machines, thus modifying the basic experience of performing an action through one's own body [9]. Such human-computer interactions appear to influence the sense of agency. For instance, Coyle and colleagues [10] showed that intentional binding was stronger in a condition in which participants used a skin-based input system that detected when participants were tapping on their arm to produce a resulting tone rather than in a condition in which they were tapping on a button to produce a similar tone. These results can be understood based on the comparator model ([11]; see [12] for a review), which suggests that SoA is experienced most clearly when there is a match between the predicted outcome and the actual outcome. Thus, according to the comparator model theory, internal forward models continuously predicts the sensory consequences of motor commands by computing the effects of an efference copy of such commands and comparing predicted and actual sensory outcomes. If there is a mismatch (for instance, created by a spatial or temporal inconsistency), the discrepancy between the predicted and sensory actual feedback is detected and the sense of being the author of the action and of controlling its outcome is reduced. In the experiment of Coyle et al. [10], the degree of congruence (and thus predictability) between the internally predicted outcome and the actual predicted outcome of skin-based stimulation could have been higher than when an external device is used to produce the outcome. SoA could thus have been boosted with skin-based inputs.

Another account is based on cue integration theory [13, 14], which suggests that SoA is based on multiple sources of information that are then combined online to infer agency. Based on this theory, receiving additional sensory cues from the limbs may contribute to a greater sense of agency vs. when not receiving this additional source of information. Using technologies such as brain-machine interfaces (BMI, or brain-computer interfaces, BCI) might thus reduce one's primary experience of being the author of one's own action, since reafference (i.e., the sensory consequences of producing an action) is weakened or absent in such cases.

The key principle of BMI is that participants can control external devices (such as a computer or neuroprostheses) by monitoring and controlling their mental state without the intervention of their sensorimotor system [15]. To do so, bidirectional learning takes place between the user and the computer: While the user learns to produce specific task-relevant (i.e., imagining moving one's right hand) vs. irrelevant (i.e., thinking about nothing) mental states, a decoding algorithm that monitors brain activity (i.e., electrical signals over motor cortex) is trained to differentiate between relevant and irrelevant activity so that the movements of an external device (i.e., a robotic hand) under its control reflect the motor intentions of the user. If the accuracy provided by the algorithm is sufficiently high, the user will then be able to efficiently control the BMI by using the same mental states as used during training. Thus, to generate an action of the external device, only internal cues are used; BMIs bypass the muscular system of the user [16]. According to the hypotheses generated by Limerick et al. [9], the

absence of sensory feedback during BMI-generated actions should reduce SoA in comparison with situations in which participants use their own hand to perform the action, since it involves either a reduction in the quality of predictions (i.e. the comparator model) or a reduced number of cues, since no reafferences are produced (i.e. the cue integration theory). The observation of decreased SoA during BMI-generated actions could have substantial implications for the daily life support of patients using neuroprostheses, but also for the notion of responsibility for actions carried out when using such devices [17, 18].

In light of these issues, we therefore aimed to address the following questions. First, we assessed to what extent individual performance to control the BMI has a measurable influence on both explicit (i.e. JoA) and implicit sense of agency (i.e., FoA). According to either the comparator model or the cue integration theory, using a BMI should reduce the sense of agency since no efference copy is available. Second, using a BMI can be particularly tiring. We thus also evaluated to what extent the reported difficulty to make the robotic hand move influences the sense of agency over BMI-generated actions. Finally, we investigated to what extent explicit and implicit sense of agency over BMI-generated actions and resulting outcomes influence the appropriation of the BMI device (here, a robotic hand), as measured through the embodiment of the device (see literature on the rubber hand illusion, [19, 20]). The embodiment of the device was measured with body-ownership, location and agency towards the robotic hand [21].

In a first study, we investigated whether or not the use of BMI influences SoA by comparing explicit and implicit SoA, both for body-generated actions and for BMI-generated actions. In the body-generated action condition, participants used their right index finger to press a key whenever they chose so as to trigger a resulting tone. Next, they had to estimate, in milliseconds, the duration of the delay between their keypress and the tone (i.e. implicit SoA; [21]). In the BMI-generated action condition, participants controlled an external (right) robotic hand through motor imagery. Instead of using their own hand, participants were instructed to relax and to use the robotic hand to press the key. Again, they had to judge the duration of the delay between the robotic hand keypress and the tone. They also had to indicate how much they felt in control of the movement that they had seen (i.e. explicit SoA). Based on Limerick et al. [9], a reduced SoA in the BMI-generated action condition in comparison with the body-generated action condition is expected. However, a previous study suggested that BMI-generated actions could share similar characteristics as body-generated actions [22]. These authors observed that explicit judgements of agency were higher for congruent BMI-actions than if neuro-visual delays were introduced—an effect that was already observed for body-generated actions [23]. However, these authors did not systematically compare implicit and explicit SoA for both BMI-generated and body-generated actions, which constitutes the main extension in the first study here. At the end of the experiment, participants had to complete a questionnaire assessing the extent to which they felt the robotic hand was embodied, as well as their level of cognitive fatigue before and after each experimental condition.

## Study 1—Method

### Participants

A total of 30 naïve right-handed participants were recruited. To the best of our knowledge, no previous studies directly compared SoA for manual and BCI actions. Given that the average sample size for SoA studies is about 20 participants (e.g. [20, 24]) and that a substantial proportion of people do not succeed in controlling a brain-computer interface–a phenomenon called 'BCI illiteracy' [25]—we decided to test 30 participants. Each participant received €15 for their participation. The following exclusion criteria were chosen prior to the experiment: (1) failure

to produce temporal intervals covarying monotonically with actual action-tone interval–(2) failure to reach chance level after the training session of motor imagery (accuracy score < 50%)–or (3) observing evidence for muscular activity during the motor imagery task. Three participants were excluded due to (3), see also point 3.4. Of the 27 remaining participants, 9 were males. The mean age was 23.78 (SD = 2.68). All participants provided written informed consent prior to the experiment. The study was approved by the local ethical committee of the Université libre de Bruxelles (054/2015).

## Materials and procedure

The experiment lasted about 1.5 hours and took place in a single session. In the first +/- 30 minutes, participants listened to instructions about the task and filled in the consent form. Next, the electroencephalography equipment was fitted. Participants then performed the first interval estimates task with their own hand (i.e., body-generated action condition). During the next +/- 30 minutes, participants performed the BCI training procedure, the duration of which depended on the number of training sessions necessary for participants to control the BCI. The final +/- 30 minutes were spent on the real feedback session with the robotic hand, the second interval estimates task (i.e., BMI-generated action condition) and the completion of the questionnaire assessing body-ownership, location and agency over the robotic hand as well as cognitive fatigue.

**Robotic hand.**   The robotic hand we used was an upgrade of the open source version described in [26] (for general requirements when designing such devices, see [27]). The index finger was actuated by a high voltage coreless digital servomotor (BMS-28A). The motion of the finger was triggered via a Matlab function using serial communication (USB). The latency between the classifier and the motion of the finger was measured to be about 110ms and was caused by the latency of the serial communication (10ms) as well as the latency of the classifier (100ms). The motion of the finger was pre-recorded in such a way that, once the motion was triggered, the finger would move quickly, taking less than 10ms to press the key. The robotic finger would maintain the key pressed for 900ms before returning to its initial position. This 900-ms timing made it possible to avoid creating a temporal distortion in the interval estimates carried out by participants, since the tone was always produced before the finger of the robotic hand lifted up. We used a membrane keyboard that produced a quiet but audible sound when pressed or released. In addition, the servomotor of the robotic hand also produced an audible "motor" sound when actuated. These two sounds were dampened substantially by the headphones used by participants during the task.

**Electroencephalography recordings and processing.**   Brain activity was recorded using a 64-channels electrode cap with the ActiveTwo system (BioSemi). Data were analysed using Fieldtrip software [28]. Muscular activity from left and right mastoids was also recorded and was used to re-reference the electrodes on the scalp. Amplified voltages were sampled at 2048 Hz. The classifier analysed the data from 15 electrodes over the motor cortex (FC3, FC1, FCz, FC2, FC4, C3, C1, Cz, C2, C4, CP3, CP1, CPz, CP2, CP4). Raw EEG data were bandpass filtered in the mu- and beta-bands (i.e., 8–30 Hz). Time-frequency decomposition of all epochs was obtained using Hanning taper with a fixed time window. In the training phase, the epochs consisted in the subtraction of the 3s-time window of the rest phase from the 3s-time window of the imagery phase. In the main experimental conditions, the epochs consisted in the 500-ms period preceding each keypress, either with participants' own fingers or with the robotic hand. Ocular activity was removed through Independent Component Analysis (ICA).

**Electromyography recordings and processing.**   Two external electrodes were placed on the flexor digitorum superficialis to control for finger movements. Participants were reminded

to remain motionless when using the robotic hand. The signal was monitored by the experimenter during the experimental session and participants were reminded not to move if muscular contractions were detectable. In addition, the signal of those electrodes was recorded and analysed after the session in order to remove trials in which muscular activity took place before the robotic hand movement. Data were filtered with a highpass of 10Hz and baseline-corrected with the period ranging from -2s. to -1.9s prior to the keypress.

**BMI training procedure.** The BCILAB platform was used for the classifier [29]. The classifier had to discriminate between two different states: motor imagery of the right hand and being at rest. To obtain the relevant data, participants were invited to sit in a relaxing position and to watch a screen. A cross was presented on the screen, either alone or accompanied by a red arrow pointing to the right (see **Fig 1A**). The arrow appeared during 3s. and disappeared during 3s. These two phases were followed by a 2 s. resting phase in which the arrow was not present but the cross was still displayed. Markers used for classification were placed at the beginning of the 3 s. of cross display and when the arrow appeared on the screen. When the cross was presented alone, participants were invited to think of nothing. Some participants reported difficulties to think of nothing and were thus invited to think of a landscape or a blue sky. Some participants also used their own strategy, like thinking of a banana on a table or of the screensaver of their own computer. When the red arrow appeared, participants were asked to imagine a movement of their right hand. They were told that they had to visualise the movement but also try to feel it from a somato-motor point of view, without actually performing the movement. Previous literature has indeed indicated that kinaesthetic motor imagery gives better performance than visual-motor imagery (e.g. [30]). Each training session lasted 2.5 minutes and was composed of 20 trials. Each trial was composed of 3s for the rest phase (i.e. thinking about nothing), 3s for the motor imagery phase and a 2s-break (see **Fig 1A**). The feature extraction that followed each training session was carried out with a Variant of Common Spatial Pattern (CSP), and the classifier used Linear Discriminant Analysis (LDA, see [29]). We used the Filter Bank Common Spatial Pattern (FBCSP) variant, which first separates the signal into different frequency bands before applying CSP on each band. Results in each frequency band are then automatically weighted in such a way that the two conditions are optimally discriminated. Two frequency bands were selected: 8-12Hz and 13-30Hz. FBCSP and LDA were applied on data collected within a 1s a time window. The position of this time window inside the - 3s of rest or right hand motor imagery was automatically optimized for each participant by sliding the time window, training the classifier, comparing achieved accuracy by cross-validation, and keeping the classifier giving the best classification accuracy.

These training sessions were performed at least twice. The first training session was used to build an initial data set containing the EEG data of the subject and the known associated condition: "motor imagery of the right hand" or "being at rest". This data set was then used to train the classifier. During the training of the classifier, a first theoretical indication of performance was obtained by cross-validation, which consisted in training the classifier using a subset of the data set, and of testing the obtained classifier on the remaining data. The second training session was used to properly evaluate the classification performance of the classifier. When the participant was asked to perform the training session a second time, the classifier was predicting in real time whether the participant was at rest or in right hand motor imagery condition. If classification performance was still low (around 50–60%), a new model was trained based on the additional training session, and combined with all previous training sessions. At most, six training sessions were performed. If the level of classification performance failed to reach chance level (i.e. 50%) after six sessions, the experiment was then terminated. We chose to also accept participants who were at the chance level in order to create variability in the control of the BMI to be able to study how this variability influences other factors. An

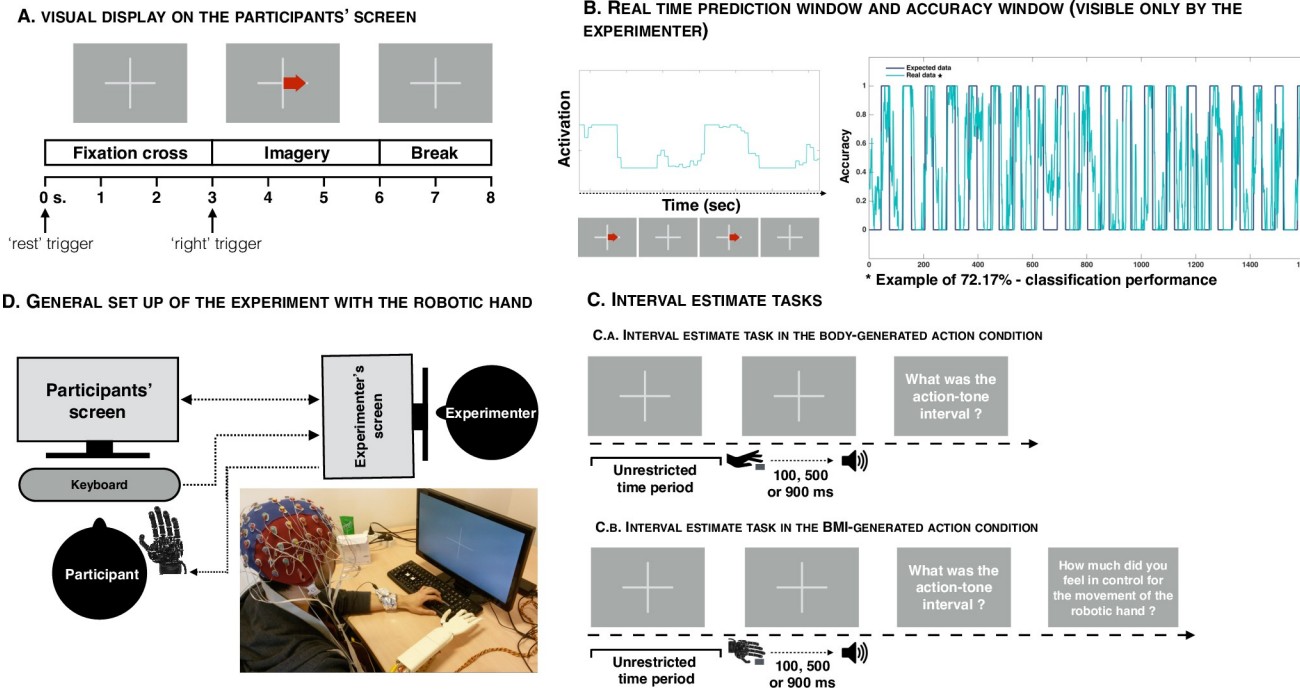

**Fig 1.** (A) Visual display of the participant's screen during the training phase. When the cross appeared alone, the participant was told to think about nothing. When the red arrow appeared, the participant had to think about a movement of the right hand. (B) Visual display on the experimenter's computer. On the left, the real-time prediction window and on the right, the final plot displaying the fit between the actual data and the expected data. (C) Graphical representation of the interval estimate tasks, in both the body-generated action condition (top) and the BMI-generated action condition (bottom). (D) General setup of the experiment with the robotic hand.

online prediction of the classifier was shown to the experimenter in real-time, in order to have an idea about the real-time accuracy of the classifier (see **Fig 1B**). This prediction oscillated between 1 (corresponding to a threshold of 0) and 2 (corresponding to a threshold of 1), 1 corresponding to rest and 2 corresponding to the imagination of the right hand. The mean threshold was set at 0.5. This means that each time the predictions were above 0.5, they were automatically transformed into a '2', and when they were below 0.5, they were automatically transformed to a '1'. The classification performance score represents the number of correct predictions divided by the number of total predictions, transformed in percentage.

**Real-time feedback session.** After the training session, participants were invited to freely control the robotic hand. They were asked to relax and to avoid performing movements with their right fingers or with their right arm. They could freely decide when to make the robotic hand move or not during 2–3 minutes, without any cues. We used the model that had reached the highest percentage of classification performance. If participants still reported that they felt that they did not control the robotic hand well enough (which happened for one participant), they again performed the BMI training procedure in order to try to increase accuracy. If participants reported being satisfied with the degree of accuracy during the real-time feedback session, we used the same model for the intentional binding task that they performed directly after.

**Interval estimates.** Participants performed the intentional binding task through the method of interval estimates [13]. This method is relatively similar to the classical intentional binding paradigm, but here, participants have to explicitly estimate and report, in ms, the duration of the delay between their keypress and the resulting tone (e.g. [21]).

When they were ready to start a trial, participants were invited to press the ENTER key on the keyboard, using their left hand. Next, a cross appeared on the screen and participants were told that they could press the '+' key with their right index finger whenever they wanted (see **Fig 1C**). They were nonetheless asked to wait a minimum of 2s. before pressing the key because of the EEG recording. After the keypress, a tone occurred randomly after 100, 500 or 900 ms, and participants were asked to estimate the delay between the keypress and the tone. To do so, they had to enter their answer with their left hand by using the keyboard's numeric keypad. They were informed that the delay would vary randomly between 1 and 1000 ms on a trial-by-trial basis (they were reminded that 1000 ms equals 1s). Participants were also told (1) to make use of all possible numbers between 1 and 1000, as appropriate, (2) to avoid restricting their answer space (i.e., not to keep using numbers falling between 1 and 100 for instance), and (3) to avoid rounding. This task was composed of 60 trials (20 for each delay).

The second interval estimates task was performed only if participants had reached at least chance level with the classification performance after the training session. The procedure was similar to the first interval estimates task, with the only difference being that participants were requested to keep their right hand in a relaxed position during the entire procedure. They were told that, as before, their left hand would serve to start each trial and to write down their interval estimations (see **Fig 1D**). To press the '+' key and thus to trigger the tone, participants were told that they had to imagine, whenever they wanted, the movement they had imagined during the BMI training session in order to make the robotic hand move. When the robotic hand moved, its right index finger pressed on the '+' key and a tone occurred. Participants were first asked to estimate the delay that occurred between the robotic hand's keypress and the tone. Then, they were invited to indicate, on a scale from 0 ('the hand moved by its own') to 10 ('the hand moved according to my own will'), if the movement of the robotic hand was caused by their own will, that is, whether the movement of the hand corresponded to the moment where they imagined the movement or not. For this task, the threshold to trigger the robotic hand was '1', meaning that they had to reach a '2' in order to trigger the movement of the hand. This was decided in order to reduce the number of false positives, since the movement of the robotic hand was harder to trigger.

*Post-session questionnaire*. Participants filled in a single questionnaire at the end of the experimental session. It assessed participants' feeling of body-ownership, location and agency

**Table 1. RHI questionnaire—appropriation of the robotic hand.**

| Ownership |
| --- |
| I felt as if I was looking at my own hand, instead of a robotic hand |
| I felt as if the robotic hand started to look like my real hand |
| I felt as if the robotic hand was my hand |
| I felt as if the robotic hand belonged to me |
| I felt as if the robotic hand was part of my body |
| **Location** |
| I felt as if my real hand was localized where the robotic hand was |
| I felt as if the robotic hand was localized where my real hand was |
| It seemed as if I were sensing the movement of my finger in the location where the robotic finger moved |
| **Agency** |
| The robotic hand moved just like I wanted it to, as if it was obeying my will |
| I felt as if I was controlling the movements of the robotic hand |
| I felt as if I was causing the movement I saw |
| Whenever I moved my finger I expected the robotic finger to move in the same way |

towards the robotic hand. This questionnaire was build based on questionnaires used in previous studies [20, 31] and adapted to the robotic hand [32]. Participants had to rate, on a scale from 0 ('totally disagree') to 6 ('totally agree') each of the 12 items of the scale (see **Table 1**). We also asked participants to complete a brief questionnaire assessing their degree of cognitive fatigue on a scale from '0' (no cognitive fatigue at all) to '10' (very high cognitive fatigue) before and after the body-generated action condition and the BMI-generated action condition.

## Study 1—Results

We performed different analyses. First, we evaluated whether or not motor imagery involves a greater desynchronization in the contralateral component in the mu and beta bands during the training phase than when being at rest. Next, we tried to ascertain if those differences correlated with the classification performance provided by the algorithm. Next, we assessed whether or not the classification performance provided by the algorithm predicted the perceived feeling of control that participants had experienced over the robotic hand during the interval estimation task. To address our main research question, we compared the interval estimates in the body-generated action condition and in the BMI-generated action condition. We additionally investigated if the variability of the perceived control of the robotic hand amongst participants could account for those results. Then, we examined whether or not this perceived control during the task would influence the global appropriation of the robotic hand, as measured through body-ownership, location and agency over this hand. Finally, we examined the potential differences between mu and beta activity during the first interval estimate task (real hand) and the second interval estimate task (robotic hand).

### Electrodes and rhythms associated with the desynchronization during the training

We conducted a repeated measures ANOVA with Rhythm (Mu, Beta) and Electrode (C3, Cz, C4) as within-subject factors on the difference in spectral power in the mu- and beta-bands between the imagery phases and the rest phases during the training session. Data were normalized by subtracting the global mean within each frequency band from each data point and by dividing the result by the global SD within the same frequency band. Since desynchronization is expected during motor imagery phases while no desynchronization is expected during rest phases, a negative value indicates that the desynchronization was stronger during motor imagery phases than during rest phases. The main effect of Rhythm was not significant ($p > .4$). The main effect of Electrode was significant ($F_{(2,52)} = 14.471$, $p < .001$, $\eta^2_{partial} = .358$). Paired comparisons indicated that the desynchronization was stronger (i.e. a lower power) over C3 (-.239, SD = .87) than over Cz (.412, SD = .83; $t_{(26)} = -4.874$, $p < .001$, Cohen's d = -.938) and, than over C4 (.035, SD = .77; $t_{(26)} = -2.250$, $p = .033$, Cohen's d = -.433), see **Fig 2A**. Desynchronization was also stronger in C4 than in Cz ($t_{(26)} = 3.495$, $p = .002$, Cohen's d = .673). This stronger desynchronization over C3 confirmed that participants were imagining a movement of the right hand. The main effect of Rhythm was not significant ($p > .4$). The interaction Electrode x Rhythm was significant ($F_{(2,52)} = 28.037$, $p < .001$, $\eta^2_{partial} = .519$). We observed that power in the mu and beta-bands did not differ for C3, Cz and C4 (all $ps > .1$). We observed that power over C3 was lower than power over Cz in the mu-band ($t_{(26)} = -7.256$, $p < .001$, Cohen's d = -1.396) but not in the beta-band ($p > .3$). Similarly, power over C4 was lower than power over Cz in the mu-band ($t_{(26)} = 6.102$, $p < .001$, Cohen's d = 1.174) but not in the beta-band ($p > .2$). Results also indicated that power over C3 was lower than power over C4 in the beta-band ($t_{(26)} = -2.088$, $p = .047$, Cohen's d = -.402) but not conclusive in the mu-band ($p = .083$).

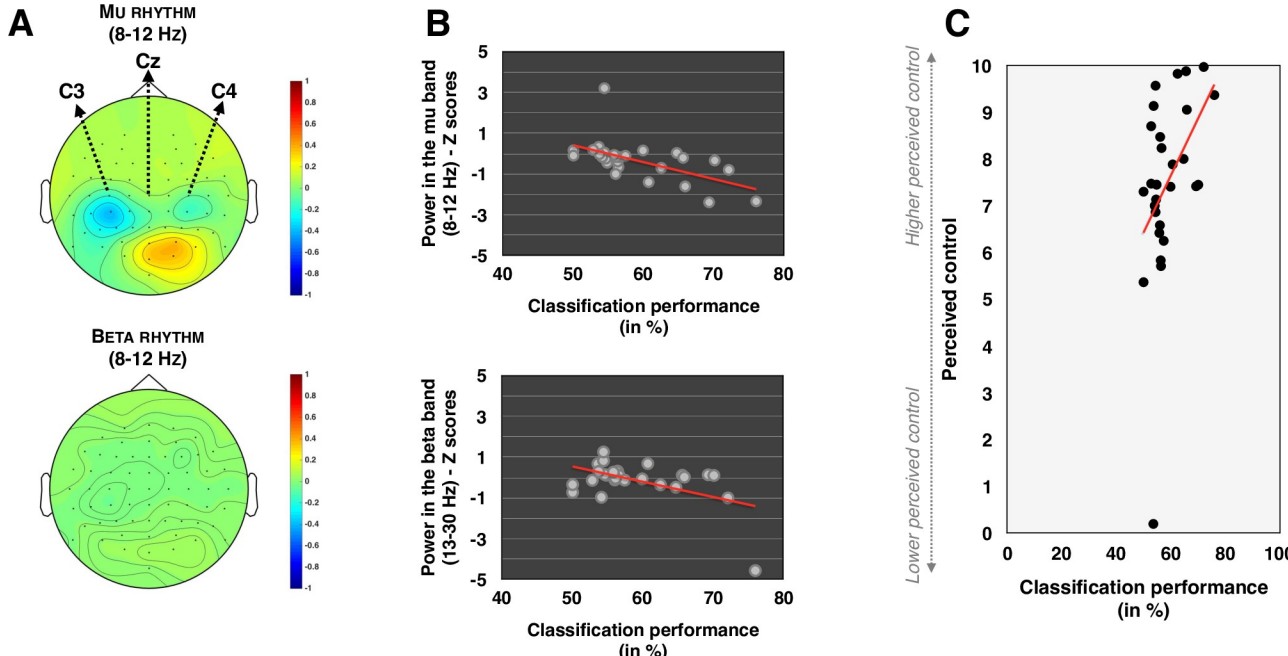

**Fig 2.** (A) Topographical representations of the power in the mu-band and of the power in the beta-band during the training phase. The color bar represents the power difference between the imagery phase and the rest phase. 1 = 100% and -1 = -100%. (B) Graphical representation of the correlations between the classification performance and the power in the mu- and the beta-bands over C3. All tests were two-tailed. (C) Graphical representation of the correlation between the classification performance and the perceived control over the movement of the robotic hand.

## Correlation between classification performance and mu/beta oscillations

We performed Pearson correlations between the classification performance and the mu and beta oscillations across C3, Cz and C4. Given that multiple correlations were performed, we applied a Bonferroni correction ($\alpha/6 = 0.008$). After this correction, we observed a significant negative correlation between the classification performance and power in the mu-band over C3 ($r = -.569$, $p = .002$) and power in the beta-band over C3 ($r = -.506$, $p = .007$), see **Fig 2B**. Other correlations were not significant (all $ps > .014$). Further, linear regressions with classification performance as the dependent variable and power in the mu- and beta-band over C3 as independent variables indicated that power in the mu-band over C3 ($t(26) = -2.305$, $p = .030$, Beta $= -.422$, VIF $= 1.321$) was a better predictor of classification performance than power in the beta-band over C3 ($p > .1$, Beta $= -.299$). One participant could be considered as an outlier since the power in the mu-band differed from more than 5 SDs from the rest of the sample. We therefore carried out the previous analysis again to ensure that the mere presence or absence of this participant would not change the results. We again observed that the power in the mu-band ($r = -.675$, $p < .001$) significantly correlated with the classification performance. Power in the beta-band on C3 marginally correlated with the classification performance ($r = -.493$, $p = .011$). Other correlations remained non-significant (all $ps > .021$).

## Does classification performance predict perceived control?

We conducted a linear regression with classification performance as the independent variable and perceived control as the dependent variable. Results indicated that the classification performance score, while remaining unknown for the participants, significantly predicted their perceived control over the robotic hand $t(26) = 2.438$, $p = .022$, Beta $= .438$), see **Fig 2C.**

### Interval estimates

Before conducting further statistical analyses, we first ensured that participants had not moved their right hand muscles when they had to control the robotic hand through motor imagery in the second intentional binding task. We observed that only 2.21/60 trials (SD = 2.99) on average contained muscular activity prior to the movement of the robotic hand, thus confirming that participants globally managed to completely relax their right hand while using the BCI. **Fig 3B** shows the averaged data (rejections-free) in the condition with participants' real hand and in the condition with the robotic hand at the moment of the keypress. Trials containing muscular activity prior to the actual movement of the robotic hand were discarded from all further statistical analyses. We then compared the interval estimates between the two tasks in order to assess whether or not the lack of sensorimotor information coming from the muscles at the moment of the keypress would reduce the implicit sense of agency. We conducted a paired-samples t-test in order to compare the interval estimates in the real hand condition and the robotic hand condition. We observed that this difference was not significant ($p > .8$), see **Fig 3A**. The Bayesian version of the same analysis supported $H_0$ ($BF_{10} = .207$). The BF value was calculated using JASP (JASP Team, 2019) and we used the default priors implemented in JASP [33]. Afterward, we investigated whether he perceived control of the robotic hand during the second interval estimate task could influence the reported interval estimates in this task, on a trial-by-trial basis. We thus performed a linear regression with the interval estimates as the dependent variable and perceived control as the independent variable. This regression was not significant ($p > .1$).

We also explore if the reported interval estimates would vary regarding actual action-tone intervals, depending on the experimental condition. We thus run a repeated-measures ANOVA with Action (real hand, robotic hand) and Delays (100, 400, 700 ms) as within-subject factors on the reported interval estimates. We observed a significant interaction between Action and Delay ($F(2,52) = 12.956$, $p < .001$, $\eta^2_{partial} = .358$). Paired-comparisons indicated that interval estimates were shorter when participants performed the action with the real hand (210 ms, SD = 163) compared to when they used the robotic hand (290 ms, SD = 137) for the 100-ms action-tone delay ($t(26) = -2.717$, $p = .012$, Cohen's d = -.523). We also observed that interval estimates were longer when participants used their real hand (624 ms, SD = 133) compared to the robotic hand (541 ms, SD = 142) for the 700ms action-tone delay ($t(26) = 3.339$, $p = .003$, Cohen's d = .643). The difference was not significant for the 500 ms action-tone delay.

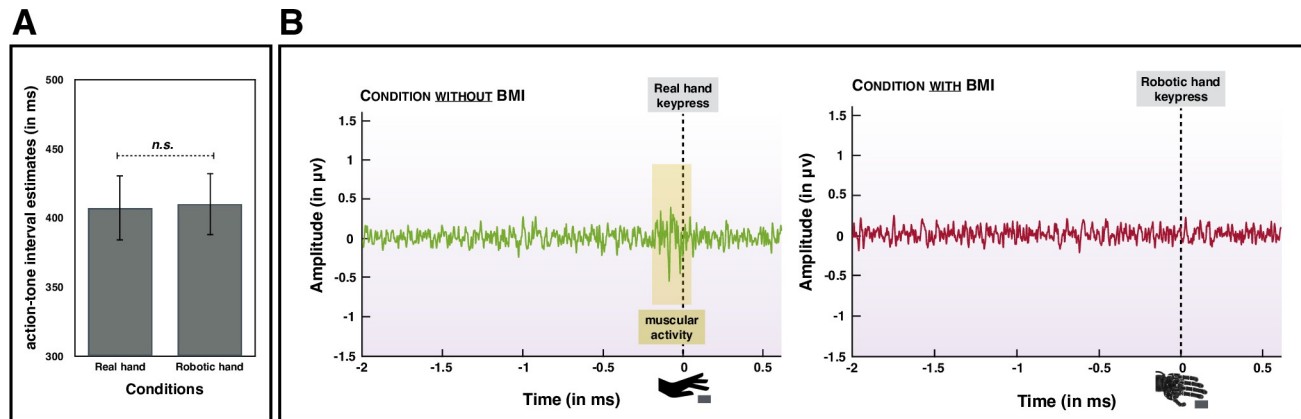

**Fig 3.** (A) Graphical representation of the comparison between interval estimates in the real hand condition and in the robotic hand condition. Test was two-tailed. (B) Graphical representation of the muscular activity before the keypress with the real hand or with the robotic hand. It confirmed that participants used motor imagery to make the robotic hand moving instead of their own muscle.

## Perceived control, interval estimates, and appropriation of the robotic hand

We first examined whether or not the perceived control of the robotic hand could predict higher appropriation of the robotic hand, as measured through scores on body-ownership, location. We took the averaged perceived control for each participant and computed non-parametric correlations (i.e. Spearman' Rho, ρ) with body-ownership, location and agency. We applied Bonferroni correction for multiple correlations (α/3 = 0.016). Results indicated that a higher score on perceived control correlated with a higher score on agency (ρ = .620, $p$ < .001). Other correlations were also positive but failed to reach significance ($p$ > .1). Then, we performed a Pearson correlation on a trial-by-trial basis between the reported perceived control and reported interval estimates. This correlation was not significant ($p$ > .1).

## Differences in mu and beta rhythms when participants used their real hand VS motor imagery

We conducted a repeated-measure ANOVA with Action (real hand, robotic hand), Rhythm (mu, beta) and Electrode (C3, Cz, C4) on the 500 ms period preceding each keypress. Importantly, we conducted those analyses without any baseline corrections. This decision was taken given that we could not ensure that baselines were absolutely identical neither between the first and the second interval estimates task nor between trials in the second interval estimates task. Participants could indeed press the key (with their real hand or with the robotic hand) whenever they wanted, thus resulting in different time periods before the actual keypress. Also, in the second interval estimates task, participants had to reach the maximum threshold ('1') to trigger the movement of the robotic hand. Thus, on some trials participants start to imagine the movement a relatively long (and undetermined) time before the actual keypress while for others the robotic hand moved even without motor imagery (false positives). Thus, we decided to perform the subsequent analyses without baseline corrections. Data were normalized by subtracting the global mean from each data point and by dividing the result by the global SD. Due to a technical failure (i.e., EEG triggers not recorded), the data of one participant were lost. The main effect of Action was not significant ($p$ > .2), suggesting no main difference between cases when the keypress was performed with participant's own finger and cases involving the robotic hand (**Fig 4**). The main effect of Rhythm was also not significant (p

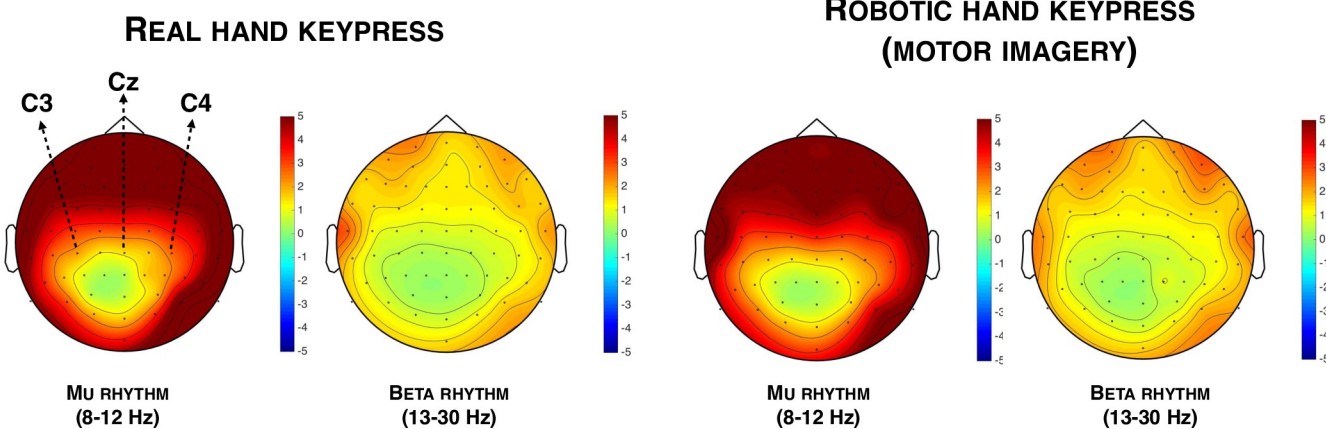

**Fig 4. Topographical representations of the power in the mu-band and of the power in the beta-band preceding the keypress in both the real hand condition and the robotic hand condition.** Data are displayed without baseline corrections.

> .9). The main effect of Electrode was significant $F(2,50) = 6.568$, $p = .003$, $\eta^2_{partial} = .208$).
Consistently with the fact that participants performed a movement with their right hand or
were imagining a movement of a right hand, paired comparisons indicated that power was
lower over C3 (-.06, SD = .83) than over C4 (.14, SD = 1.15, t(25) = -2.546, $p = .017$, Cohen's d
= -.499) and lower over Cz (-.077, SD = .81) than over C4 (t(25) = -2.946, $p = .007$, Cohen's d =
-.578). The difference was not significant between C3 and Cz ($p > .8$). We also observed a sig-
nificant interaction Rhythm x Electrode (F(2,50) = 7.285, $p = .002$, $\eta^2_{partial} = .226$). Paired
comparisons indicated that power in the mu-band did not statistically differ between C3 and
Cz, and Cz and C4 (all $ps > .1$). Power in the mu-band over C3 (-0.45, SD = .89) was margin-
ally lower than the power in the mu-band over C4 (.082, SD = 1.14, t(25) = -2.030, p = .053,
Cohen's d = -.398). Paired comparisons further indicated that power in the beta-band was
lower over C3 (-.09, SD = .82) than over C4 (.21, SD = 1.20, t(25) = -2.694, $p = .012$, Cohen's d
= -.528) and that power in the beta-band was also lower over Cz (-.11, SD = .85) than over C4
(t(25) = -4.139, $p < .001$, Cohen's d = -.812). The difference between C3 and Cz in the beta-
band was not significant ($p > .6$). The interaction Action x Electrode was also significant (F
(2,50) = 6.765, $p = .003$, $\eta^2_{partial} = .213$). Paired comparisons indicated that in the body-gener-
ated action condition, power was lower over C3 than over C4 (t(25) = -2.851, $p = .009$, Cohen's
d = -.559) and lower over Cz than over C4 (t(25) = -3.160, $p = .004$, Cohen's d = -.620). The differ-
ence was significant between power over C3 and power over Cz ($p>.9$). In the BMI-generated
action condition, power was lower over Cz than over C4 (t(25) = -2.457, $p = .021$, Cohen's d =
.482). Other comparisons were not significant (all $ps > .070$). These results are consistent with
the fact that the lateralization over a right hand movement is more marked when participants
execute a movement with their own right hand than when they imagine a movement of the
right hand. Other interactions were significant (all ps > .079).

We additionally investigated, with Pearson correlations, whether or not the difference in
power in the mu- and the beta-bands between the real hand keypress and the robotic hand
keypress on C3, Cz and C4 for each participant could predict the difference in interval esti-
mates between the same two tasks. None of these correlations were significant (all $ps > .2$),
thus suggesting that none of the differences observed in the mu- and beta-bands could predict
the difference at the implicit level of the sense of agency, as measured through interval esti-
mates. Finally, we performed Pearson correlations to check whether or not mu and/or beta
desynchronization during the second interval estimates task were associated with the per-
ceived control of the robotic hand. None of these correlations were significant, no matter the
rhythm or the electrode (all $ps > .4$).

## Cognitive fatigue after each interval estimate task

At the end of the experiment, participants were asked to rate their level of cognitive fatigue on
a scale from '0' (no cognitive fatigue at all) to '10' (very high cognitive fatigue) before and after
the body-generated action condition and the BMI-generated action condition. We conducted
a repeated-measures ANOVA with Condition (body-generated, BMI-generated) and Moment
(Before the task, After the task) on the reported level of cognitive fatigue. We observed a main
effect of Condition, with participants reporting more cognitive fatigue after the BMI-generated
action condition (2.77, SD = 2.20) than after the body-generated action condition (1.68,
SD = 1.32, F(1,25) = 86.510, p < .001, $\eta^2_{partial} = .776$). We also observed a main effect of
Moment, with more cognitive fatigue reported after each task (4.59, SD = 1.58) than before
each task (2.30, SD = 1.43, F(1,25) = 89.378, $p < .001$, $\eta^2_{partial} = .781$). The interaction Condi-
tion x Moment was also significant (F(1,25) = 5.749, $p = .024$, $\eta^2_{partial} = .187$). Paired compari-
sons indicated that participants reported a higher level of cognitive fatigue when they started

the BMI-generated action condition (3.27, SD = 1.61) than when they started the body-generated action condition (1.35, SD = 1.59, t(25) = -6.809, $p$ < .001, Cohen's d = -1.335). This result is consistent with the fact that participants always performed the BMI-generated action condition after the body-generated action condition. Participants also reported a higher level of cognitive fatigue after the BMI-generated action condition (6.12, SD = 2.1) than after the body-generated action condition (3.08, SD = 1.6, t(25) = -7.354, $p$ < .001, Cohen's d = -1.442).

## Study 1—Discussion

The main goal of Study 1 was to examine whether or not the lack of sensorimotor information when using a BMI would diminish the primary experience of agency and to investigate how the degree of control over the BMI would lead to a greater embodiment of the robotic hand.

First, we ensured that the motor imagery-based BMI that we developed was reliable. In the training session, we replicated the classical pattern of oscillations of motor imagery: We observed that the desynchronization during motor imagery was greater on C3 than on Cz and C4 in comparison with being at rest, which is consistent with the fact that we asked the participants to imagine a movement of the right hand. We additionally observed that the difference between the imagery phases and the rest phases in the mu band on C3 was the better predictor of the classification performance provided by the algorithm after the training. Even though the power in the mu- and beta-bands did not differ over C3, the fact that the power in the mu-band was a better predictor of the classification performance than power in the beta-band is consistent with past literature comparing kinaesthetic motor imagery and visual-motor imagery [30]. Power in the mu-band has indeed been associated to the sensorimotor treatment of movements [34–38] and we specifically asked our participants, not only to try to image a movement of the right hand, but also to try to 'feel' that movement. This probably explains why power in the mu-band was a better predictor of the classification performance than power in the beta-band. It is important to note that participants were not told what the classification performance score was after their training sessions. Yet, their perceived control over the movement of the robotic hand positively correlated with this classification performance score. Based on those results, we concluded that our BMI was reliable since it involved motor imagery that was similar to previous studies and that the desynchronization of the mu band during the motor imagery task predicted the classification performance score, which correlated with the perceived control.

To examine whether or not the lack of sensorimotor information when using a BMI would reduce the primary experience of agency, we compared a condition in which participants had to use their own hand to press a key versus a robotic hand controlled through motor imagery. When they use motor imagery to control an external device, participants do not receive sensorimotor feedback while using the robotic hand to press the key [39]. If sensorimotor information during the action is crucial for SoA, we would have expected longer interval estimates when participants used a robotic hand in comparison with when they used their own hand. However, we observed that this was not the case, since interval estimates did not differ between these two experimental conditions. It thus appears that the sensorimotor information is not the most important cue for generating a sense of agency (i.e. the cue integration theory). This result is consistent with previous studies, which showed that the congruence of visual cues was more important for SoA than sensorimotor feedback [22, 40]. It could nonetheless be argued that since participants used kinaesthetic motor imagery to control the BMI they still somehow received sensorimotor information during the preparation of the movement. However, this sensorimotor information was necessarily incongruent, since participants imagined a movement of their whole right hand while the robotic hand performed a simple keypress with its

index finger. Future studies could use a BMI that does not rely on kinaesthetic motor imagery (e.g. [41]) to further explore the relevance of sensorimotor information in the generation of a sense of agency.

We also observed that for small action-tone delays, the reported interval estimates were smaller for biological movements executed with one's own hand than for movements executed through the robotic hand. However, this pattern reversed for longer action-tone delays. This is suggestive that different mechanisms drive agency for biological and non-biological movements, an aspect of our findings that warrants further research.

As in previous studies, we did not observe differences in mu and beta oscillations during motor imagery and real hand movements (e.g. [42]). We nonetheless observed that the desynchronization was more contralateral during real hand movements than during motor imagery. This may be due to the fact that in the motor imagery task, even if participants were suggested to imagine a movement of their right hand, some of them could have used their own strategy, such as imagining a movement of both hands.

We also observed that the higher the perceived control over the robotic hand was, the higher the scores of agency over the robotic hand were. For body-ownership and location scores, this correlation was also positive but did not reach significance.

Learning to control a BMI, to some extent, involves learning to perform a new movement. Successfully triggering a movement of the robotic hand with a BMI requires a greater cognitive effort than simply pressing the keypress with one's own finger, which is a very common movement for healthy adults. Using a BMI is also more cognitively demanding than acting with one's own finger directly (e.g. [43]). The majority of our participants indeed reported that using the robotic hand was exhausting: They reported a higher level of cognitive fatigue in the questionnaires after the BMI-generated action condition than after the body-generated action condition. This may be due to, first, the fact that the BMI-generated action condition was always conducted after the body-generated action condition, and second, that the BMI-generated action condition was particularly tiring. These factors have been previously shown to modulate the experience of agency [44–47] and could thus also have influenced our results in the present study. To diminish the level of cognitive fatigue in Study 2, we did not include the body-generated condition. Instead we trained participants on two consecutive days with the BMI-generated action condition only. On each day, participants performed 5 training sessions instead of a variable number of sessions, such as in Study 1. This was intended to keep cognitive fatigue comparable between participants.

## Study 2 –Method

### Participants

As in Study 1, a total of 30 (new) naïve right-handed participants (13 males) were recruited. Each participant received €40 for their participation. No participants were excluded based on the same exclusion criteria as used in Study 1. However, we realised after the first 4 participants that an error in the code introduced a delay longer than the one defined between the action and the tone. Those 4 participants were excluded from further analyses associated with interval estimates, but were preserved so to evaluate the reliability of our BMI. The mean age of all included participants was 23.77 (SD = 2.76). All participants provided written informed consent prior to the experiment. The study was approved by the local ethical committee of the Université libre de Bruxelles (054/2015).

### Method and material

The method was globally similar to the one of Study 1. Participants were trained on two consecutive days, at the same hour of the day. On the first day, they first started with a 24-trial

training session for the interval estimate task. During those two experimental sessions, all participants had a fixed number of training sessions (i.e. 5). At the end of those 5 training sessions, they performed the interval estimate task with the robotic hand controlled through the BMI. After judging the action-tone interval, participants were asked, on each trial, to judge if the movement of the robotic hand was caused by their own will (such as in Study 1) and to indicate how difficult ('0'—not difficult at all to '10'—very difficult) it was to make the robotic hand move. At the end of the session, participants also completed a scale that evaluated their level of cognitive fatigue ('0'—not cognitively tired at all to '10'—very cognitively tired) before and after the interval estimate task. They also completed the questionnaire evaluating body-ownership, location and agency over the robotic hand on both days.

## Study 2 –Results

We first assessed if we could replicate the main characteristics associated with the classification performance of the BMI observed in Study 1. To do so, we first evaluated desynchronization in the mu- and beta-bands between the imagery phases and the rest phases of the training sessions on both days. Again, we then checked if power in the mu- and beta-bands correlated with the classification performance provided by the algorithm. Next, we assessed whether or not the classification performance predicted the perceived feeling of control that the participants had over the robotic hand during the interval estimate task. Then, we investigated if the implicit sense of agency when using the BMI was different on both days and if it correlated with the reported perceived control of the hand and the reported difficulty making the robotic hand perform the movement, on a trial basis. We also examined if the interval estimates correlated with the appropriation of the hand. Finally, we evaluated if perceived control, classification performance, cognitive fatigue and the appropriation of the hand differed on days 1 and 2, and how these variables correlated with each other.

### Electrodes and rhythms associated with the desynchronization during the training

We conducted a repeated-measures ANOVA with Session (Day 1, Day 2), Rhythm (mu, beta) and Electrode (C3, Cz, C4) and as within-subject factors on the difference between the imagery phases and the rest phases during the training session. The data of 2 participants were lost due to faulty electrodes in at least one training session. The main effect of Electrode was significant ($F(2,54) = 8.993$, $p < .001$, $\eta^2_{partial} = .250$). Paired comparisons indicated that the desynchronization was stronger over C3 (-.239, SD = .85) than over Cz (.22, SD = .92; $t(27) = -3.448$, $p = .002$, Cohen's d = -.652) and, than over C4 (.01, SD = .72; $t(27) = -2.517$, $p = .018$, Cohen's d = -.476). The desynchronization was also stronger in C4 than in Cz ($t(27) = 2.393$, $p = .024$, Cohen's d = .452). The higher desynchronization in C3 confirmed that participants mostly imagined a movement of the right hand, see **Fig 5**. Neither the main effect of Rhythm ($p > .9$), nor the effect of Session ($p > .9$) were significant. The interaction Electrode x Rhythm was marginal ($F(2,54) = 3.066$, $p = .055$, $\eta^2_{partial} = .102$) and thus not investigated further. Other interactions were not significant (all $p$s > .4).

### Classification performance and correlation with mu/beta oscillations

We performed Pearson correlations between the classification performance and the mu and beta oscillations across C3, Cz and C4. Analyses were performed on both days combined to achieve better statistical power, since the effect of session did not significantly influence power in the mu- and beta-bands. Given that multiple correlations were performed, we applied a Bonferroni correction ($\alpha/6 = 0.008$). After this correction, we observed a significant

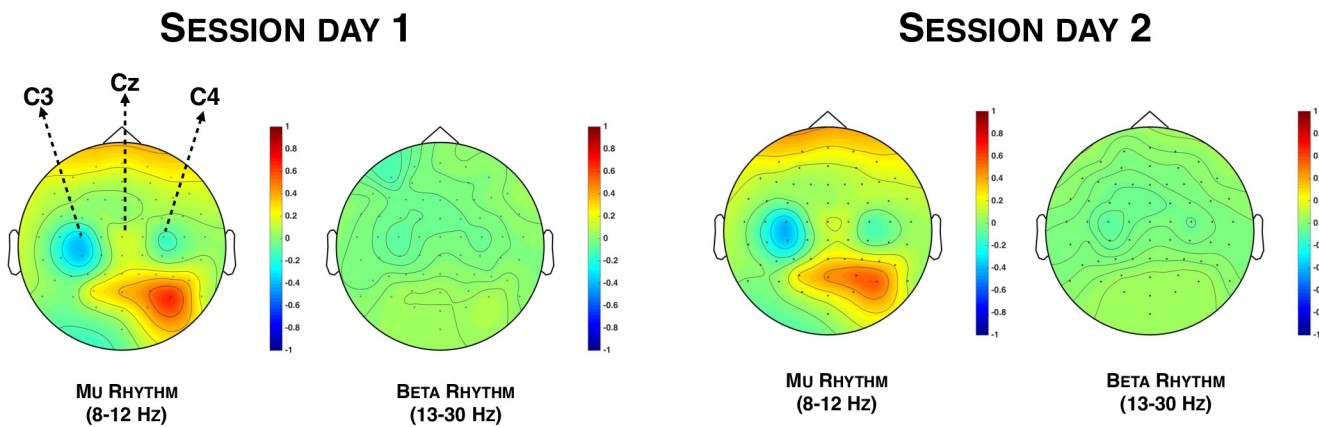

**Fig 5. Topographical representations of the power in the mu-band and in the beta-band during the training session on both days.** The color bar represents the power difference between the imagery phase and the rest phase. 1 = 100% and -1 = -100%.

correlation between classification performance and power in the mu-band (r = -.557, $p <$ .001) and power in the beta-band (r = -.488, $p <$ .001) over C3. The correlation was also significant with the power in the mu-band over C4 (r = -.417, $p$ = .001). Other correlations were not significant (all $ps >$ .064). Linear regressions with classification performance as the dependent variable and power in the mu- and beta-bands over C3 and power in the mu-band over C4 as independent variables indicated that power in the mu-band over C3 (t(55) = -2.833, $p$ = .006, Beta = -.460) was a better predictor of the classification performance than the power in the beta-band on C3 (t(55) = -2.116, $p$ = .039, Beta = -.294) and, than the power in the mu-band on C4 which was no significant anymore ($p >$ .6). Again, results were similar to those of Study 1, with a greater desynchronization in the mu-band on C3 being associated with a higher classification performance.

## Does classification performance predict perceived control?

We again conducted a linear regression with classification performance as the independent variable and perceived control as the dependent variable, again on both sessions combined. As in Study 1, results indicated that classification performance, unbeknownst to participants, significantly predicted their perceived control over the robotic hand (t(51) = 2.247, $p$ = .029, Beta = .303).

## Interval estimates

As expected, participants reported shorter interval estimates for the 100 ms-delay (442.8 ms, SD = 212.3) than for the 400 ms-delay (528.9 ms, SD = 208.7) and, than for the 700 ms-delay (575 ms, SD = 218.72). Paired comparisons indicated that interval estimates did not significantly differ between day 1 and day 2 ($p >$.06). We then performed non-parametric correlation with Spearman's rho ($\rho$) between the reported interval estimates, the perceived control over the movement of the robotic hand and the reported difficulty to make the robotic hand perform the movement, on a trial-by-trial basis. Bonferroni corrections involved a $\alpha/2$ = 0.025. Results revealed a negative significant correlation between the interval estimates and the perceived control ($\rho$ = -.106, $p <$ .001), see **Fig 6A**. Since longer interval estimates actually reflect a reduced sense of agency, this negative correlation indicates that the higher the perceived control was, the higher sense of agency was. The correlation between the interval estimates and the reported difficulty was also significant ($\rho$ = .061, $p$ = .001), suggesting that the more difficult it was for the

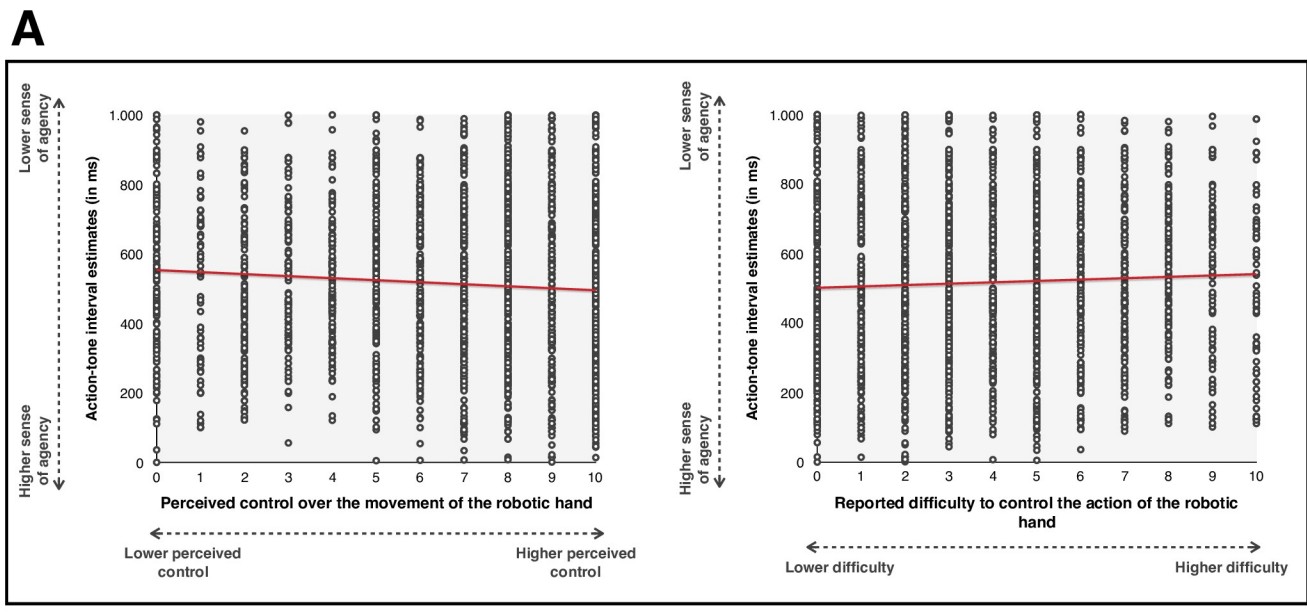

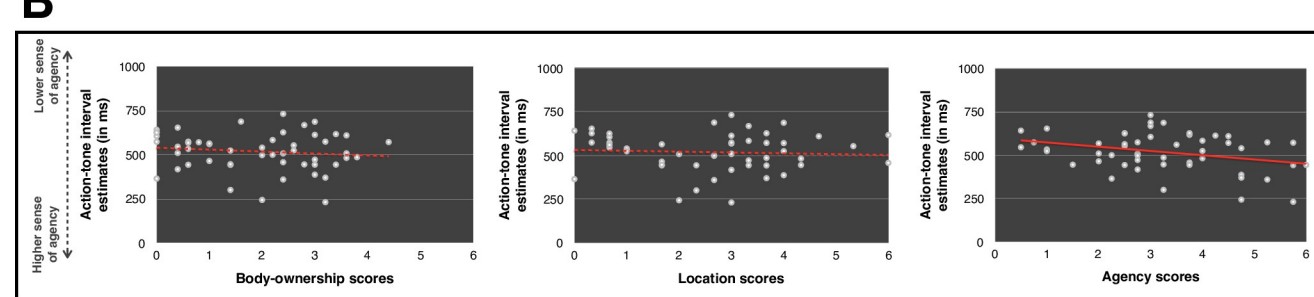

**Fig 6.** (A) Graphical representation of the correlations between interval estimates and the perceived control over the movement of the robotic hand (left) and the difficulty to control the movement of the robotic hand (right). (B) Graphical representation of the relationship between interval estimates and body-ownership, location and agency scores over the robotic hand. All tests were two-tailed. Full lines represent a significant result and dotted lines represent a non-significant result.

participants to make the robotic hand move, the lower was their sense of agency, since it corresponds to higher interval estimates. Perceived control also negatively correlated with the reported difficulty ($\rho$ = -.184, $p < .001$), suggesting that the more difficult it was for participants to make the robotic hand move, the lower was their perceived control of the hand. We then performed a linear regression with the estimated interval estimate as the dependent variable and the perceived control and the reported difficulty to make the hand performing the movement as the independent variables. Results indicated that the perceived control (t(3184) = -4.131, $p < .001$, Beta = -.074) was a better predictor of interval estimates than the reported difficulty (t(3184) = 2.195, $p = .028$, Beta = .039), see **Fig 6B**. VIFs were of 1.021, indicating an absence of collinearity. Further, we took the averaged interval estimates for each participant as the independent variable and we conducted linear regressions to assess whether it would predict scores on body-ownership, location and agency over the robotic hand. Results were again computed on sessions 1 and 2 combined. We observed that interval estimates negatively predicted the score on agency (t(51) = -2.229, $p = .031$, Beta = -.382), meaning that the higher was their implicit sense of agency during the task, the more agency participants reported over the robotic hand. Other linear regressions with body-ownership and location were not significant (all $p$s > .6).

## Classification performance, perceived control, cognitive fatigue and appropriation of the robotic hand

We first performed a paired comparison between the classification performance on day 1 and day 2. Results indicated that the classification performance was higher on day 2 (61.72%, SD = 7.43) than on day 1 (59.47%, SD = 6.79, t(29) = -2.266, $p$ = .031, Cohen's d = .414), showing that participants improved their performance from day 1 to day 2, see **Fig 7A**. However, the perceived control did not differ between the two days ($p$ > .3). We then compared the reported cognitive fatigue on both day 1 and day 2 after the use of the BMI during the interval estimate task. To do so, we first subtracted the reported score of cognitive fatigue before the task to the reported score of cognitive fatigue after the task. We then performed a paired comparison, which revealed that participants reported less cognitive fatigue on day 2 (2.96, SD = 1.93) than on day 1 (4.03, SD = 2.64, t(29) = 2.948, $p$ = .006, Cohen's d = .535). Other dependent variables associated with the appropriation of the robotic hand (i.e., body-ownership, location, agency) did not differ between the two days (all $p$s > .06).

We took the averaged perceived control for each participant and ran non-parametric correlations (i.e. Spearman' Rho, ρ) with body-ownership, location and agency in both sessions combined. We applied Bonferroni correction for multiple correlations (α/3 = 0.016). We observed a significant positive correlation between the perceived control and body-ownership scores (ρ = .453, $p$ = .001) and agency scores (ρ = .899, $p$ < .001). The correlation between the perceived control and location scores was marginal (ρ = .319, $p$ = .019), see **Fig 7B**. These results suggest that a greater perceived control over the action of the robotic hand leads to a greater appropriation of the robotic hand, with higher scores on body-ownership, location and agency.

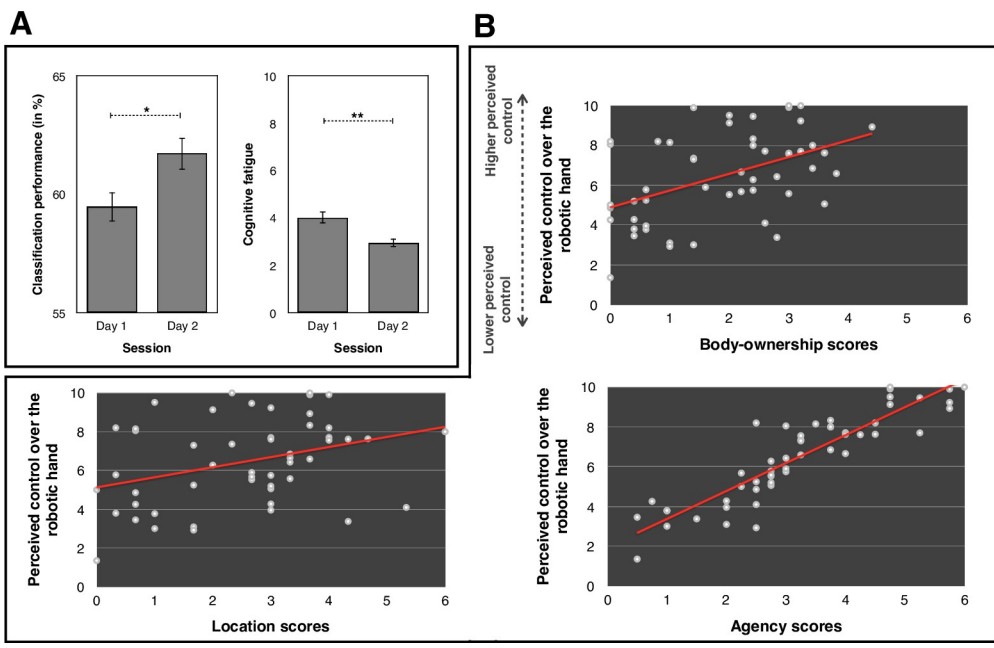

**Fig 7.** (A) Graphical representations of the difference between day 1 and day 2 on the classification performance scores and on the reported cognitive fatigue. All tests were two-tailed. * indicates a p between 01 and 05. ** represents a p between .001 and .01. (B) Graphical representation of the relationship between the perceived control over the robotic hand body-ownership, location and agency score. All tests were two-tailed.

## Study 2 –Discussion

The pattern of results of Study 2 was globally similar to Study 1. Indeed, we observed that the desynchronization in the mu band on C3 again predicted classification performance, which itself predicted perceived control over the robotic hand. Again, we observed that perceived control over the robotic hand was a reliable predictor of the appropriation of the robotic hand, as shown by significant correlations with body-ownership, location and agency.

Interestingly, classification performance was higher on day 2 than on day 1. Further, the cognitive fatigue reported was lower on day 2 than on day 1, suggesting a beneficial effect of BMI training. However, this 2-day training did not appear to be sufficient to produce an improvement in the other variables, such as the interval estimates and the scores on body-ownership, location and agency. Future studies could evaluate the impact of longer BMI training on those variables.

Results on interval estimates showed that the implicit sense of agency when using BMI was modulated by both perceived control over the external device and by the reported difficulty to make it move. We indeed observed that the more participants reported subjective control over the robotic hand, the stronger their implicit sense of agency was. Of note, this relationship was not significant in Study 1, probably due to a lower statistical power since the analyses included twice as many trials in Study 2 compared to Study 1, due to the combination of both sessions. We also observed that the more difficulty participants reported in performing the movements, the lower was their implicit sense of agency, as shown by longer interval estimates. This is consistent with previous studies that have suggested that SoA is reduced by cognitive effort (e.g. [45]).

## General discussion

In this paper, we investigated how using a BMI would influence the experience of being the author of an action for the user. BMI involves performing movements through an external device that does not imply any bodily or muscle movements. Previous theories on the sense of agency had emphasised the importance of sensorimotor cues for generating a sense of agency (SoA) [10, 13, 14, 48]. We found in Study 1 that the absence of sensorimotor information was not detrimental for the feeling of agency, as measured through interval estimates, which did not differ between a condition in which participants used their own hand and a condition in which they used the robotic hand to perform a keypress. It thus appears that participants can experience a 'disembodied agency', as long as they feel that they have an active control over the device [16]. It had also been previously suggested that SoA for BMI can be illusory [49]. Here, we observed that it may not be the case. Participants were not aware of the classification performance score that they had reached after the training session. Yet, their own perception of control over the robotic hand correlated with this classification performance score and with their implicit experience of agency, as measured through interval estimates. This result is in line with a former study, which showed that participants are good at evaluating their ability to control a BMI even if they do not receive an online feedback on their performance [50].

In both studies, we observed that experiencing a high control over the brain-computer interface predicts a greater appropriation of the robotic hand, as measured through questionnaires assessing body-ownership, location and agency. The strong relationship between the perceived control of the robotic hand during the task and the sense of agency measured during a post-session questionnaire is coherent since these two measures make it possible to assess explicit sense of agency (i.e., SoA). The fact that perceived control of the robotic hand also positively predicts body-ownership has deeper theoretical implications about the relationship between body-ownership and SoA. Thus, in [24], the authors compared ownership scores and proprioceptive drift (i.e. an implicit measure of body-ownership, [19]) for the classic rubber

hand illusion (i.e. visuo-tactile) and the active rubber hand illusion. Crucially, they found no differences between these two conditions, neither with the proprioceptive drift nor with the questionnaires. They suggested that different types of sensory information can be combined to elicit the ownership perceptions and that agency is not a modulator of body-ownership because efferent signals received in the active version of the RHI did not increase the strength of the illusion. However, this conclusion did not receive the support of other studies (e.g. [20, 51]). The present study supports that a high feeling of control can contribute to a greater feeling of body-ownership, which is an interesting finding for restorative medicine for amputee patients.

A limitation of the present study is that we did not use a control, passive condition similarly to previous studies on interval estimates. Such passive conditions involve a lack of volition in the realization of the movement, for instance by using a TMS over the motor cortex [3] or by having a motoric setup forcing the participant's finger to press the key (e.g. [13]). Neither Study 1 nor Study 2, includes such passive conditions, thus precluding our conclusions on the extent to which a BMI-generated action condition does involve a high sense of agency. The main reason for not introducing such control conditions stemmed from the tiring nature of the task asked from participants (i.e. learning to control a robotic hand through a BMI) and its duration (i.e. 2 hours). Adding additional experimental conditions would have strongly increased the fatigue of participants during the task. The existing literature on active, voluntary conditions in which participants use their own index finger to press a key whenever they want is very extensive and consistently points towards lower interval estimates in active conditions than in passive conditions, involving a higher sense of agency (see [5–7] for reviews). We thus considered that the body-generated action condition used in the present study, which is entirely similar to the active conditions previously reported in the literature, would be the baseline associated with a high sense of agency. This condition was then compared to the BMI-generated action condition in order to evaluate if using a BMI would lead to a similarly high sense of agency than in the body-generated action condition. In Study 2, we assumed that results of Study 1 were replicated for the BMI-generated action condition, thus resulting in a sense of agency on both Days 1 and 2. However, to draw more reliable conclusions on the sense of agency for MBI-generated actions, a passive control condition should be added.

Several articles have argued that demand characteristics [52] and expectancies can predict scores on the classical rubber hand illusion (RHI) questionnaires [53]. In the present study, we limited as much as possible the effects of both demand characteristics and expectancies on participants' answer to the questionnaire related to the appropriation of the robotic hand: (1) Participants were not told anything about the rubber hand illusion or the fact that some people may experience the robotic hand as a part of their body during the experiment; (2) We did not tell them in advance that they would have to fill in a questionnaire regarding their appropriation of the robotic hand at the end of the experiment; (3) We avoided creating a proper classical 'rubber hand illusion': the robotic hand was placed at a congruent place on the table regarding the participant's arm, but we did not use the blanket to cover their real arm, neither we used a paintbrush to induce an illusion. This procedure limited the possibility for participants to guess what we would assess since in the classical rubber hand illusion, placing the blanket on the participant's arm make them realize that we investigate to what extent they appropriate this hand in their body schema. However, we cannot fully rule out the influence of demand characteristics and expectancies, as participants perceiving a better control over the hand, even without knowing their own classification performance, could have intuitively indicated higher scores on the questionnaires. Control studies could thus be performed to manipulate demand characteristics in order to evaluate its effect on the appropriation of a robotic hand in a brain-machine interface procedure [53].

Finally, another limitation is that we could not entirely rule out the presence of sub-thresholded muscular activity that could have triggered the BMI during the task. The flexor digitorum superficialis where the electrodes were placed does not allow to record activity associated with the movement of the thumb. In addition, no electrodes were placed on the left arm. Those movements were only controlled visually by the experimenter during the task.

Given that the entire brain-computer interface was connected to an external robotic hand connected to the computer through an USB port, a certain latency was present during the moment where the algorithm detected a hit during motor imagery and the actual movement of the hand. This latency has been calculated and is about 110 ms. It refers to the maximum latency of the classifier (predictions were realized each 100 ms) added to the averaged latency of an USB command (+- 10 ms). One could thus argue that this delay could have negatively affected SoA when using the robotic hand in comparison with the real hand movement. However, a previous study observed that visual delays below 1s did not modulate SoA over the movement of two virtual hands controlled through motor imagery [22]. We thus consider that a 110 ms-delay did not influence our results.

One of the main and most important implications of the use of BMI is for amputees or paralysed patients. It is important to understand what factors would lead to a better appropriation of the controlled device in order to improve the well-being of those patients. In the present study, we bring the first evidence that using BMI interfaces is unlikely to negatively impact human SoA and that the better participants' control is, the greater the appropriation of the device is.

## Author Contributions

**Conceptualization:** Emilie A. Caspar, Axel Cleeremans.

**Data curation:** Emilie A. Caspar.

**Formal analysis:** Emilie A. Caspar.

**Funding acquisition:** Emilie A. Caspar, Albert De Beir.

**Investigation:** Emilie A. Caspar.

**Methodology:** Emilie A. Caspar, Albert De Beir.

**Project administration:** Emilie A. Caspar.

**Resources:** Emilie A. Caspar, Gil Lauwers, Bram Vanderborght.

**Software:** Gil Lauwers.

**Supervision:** Emilie A. Caspar, Albert De Beir, Axel Cleeremans, Bram Vanderborght.

**Validation:** Emilie A. Caspar.

**Writing – original draft:** Emilie A. Caspar.

**Writing – review & editing:** Axel Cleeremans, Bram Vanderborght.

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
