## [Decision Letter · Decision Letter 0]

15 Jun 2020

PONE-D-20-06849

How using brain-machine interfaces influences the human sense of agency

PLOS ONE

Dear Dr. Caspar,

Thank you for submitting your manuscript to PLOS ONE. After careful consideration, we feel that it has merit but does not fully meet PLOS ONE’s publication criteria as it currently stands. Therefore, we invite you to submit a revised version of the manuscript that addresses the points raised during the review process.

We look forward to receiving your revised manuscript.

Kind regards,

Jane Elizabeth Aspell, PhD

Academic Editor

PLOS ONE

Journal Requirements:

3. Please include a copy of Table 1 which you refer to in your text on page 19.

Reviewers' comments:

Reviewer's Responses to Questions

**Comments to the Author**

1. Is the manuscript technically sound, and do the data support the conclusions?

Reviewer #1: Yes

Reviewer #2: Partly

2. Has the statistical analysis been performed appropriately and rigorously? 

Reviewer #1: Yes

Reviewer #2: N/A

3. Have the authors made all data underlying the findings in their manuscript fully available?

Reviewer #1: Yes

Reviewer #2: Yes

4. Is the manuscript presented in an intelligible fashion and written in standard English?

Reviewer #1: No

Reviewer #2: Yes

5. Review Comments to the Author

Reviewer #1: PONE-D-20-06849

This study looked at the sense of agency for actions performed using an EEG-based brain-machine interface (BMI). The intentional binding effect was used as the primary dependent measure, where subjects directly estimate the interval between the cause (button press) and effect (auditory click). In two experiments there were no significant differences in interval estimates between motor actions and BMI-mediated actions, suggesting no difference in agency. The authors did, however, find relationships between other variables in the experiment, such as the sense of control, experienced difficulty, ownership of the robotic hand, and agency (as indexed by the interval estimates).

The paper is not very clearly written, and it was very difficult to make sense of it in places, often because crucial information was left out or not made clear, or jargon was undefined. It really reads like a first or second draft of a paper, rather than a paper that has been through a few rounds of internal review before being deemed ready to submit to a journal for peer review. Many of my comments are thus focused on the writing and on making things more clear. The paper does not make any strong claims, and the conclusions that are drawn seem reasonable given the data. The methods involved in controlling the robotic hand and the robotic hand itself were not given in sufficient detail (for example we do not know anything about the features used to discriminate motor imagery from rest conditions, even though this might be relevant to some of the results that involved correlation with the “theoretical accuracy”). The statistics appear to be appropriate and properly reported, except that for correlations with ordinal data (as in figure 6A) it can be better to use a non-parametric statistic like Spearman’s rho.

Specific comments (in no particular order):

p. 4, line 3: what is a “skin-based input”?

p. 6, line 21”main age” should be “mean age”

p. 7, bottom and p. 12, line 22, and figure 3

So they used bi-polar electrodes to record EMG over the flexor digitorum superficialis, but it seems that they just monitored it on-line or on a single-trial basis (says that they used the EMG traces off-line to remove trials with muscle activity). But they did not, it seems, check for sub-threshold muscle activity during motor imagery by looking at the average EMG power over all trials. To do this you have to take the sqrt of the sum of squares in a sliding window for each trial, before you average the trials together. You have to do this if you want to rule out any muscle activity, including sub-threshold muscle activity.

Also it says that EMG data were high-pass filtered at 10 Hz, but then nothing else. You can’t really see muscle activity without at least rectifying the signal, or (better) converting to power.

p. 9, line 4: They checked to see if, after six training sessions, accuracy of the classifier was below chance. But that’s not what you want to do. You want to check to see if accuracy of the classifier is significantly *different* from chance. Performance of 51% correct might still be chance! You should always say “not different from chance” rather than “below chance”.

p. 10, line 5: "red cross” should be “red arrow”

It would be helpful and useful if you computed AUC for control of the robotic hand, just to give an idea of what level of control your subjects managed to achieve.

What do you mean by the “theoretical” accuracy score? This was not clear enough. On what data set was this theoretical accuracy determined? You should have a test set and a training set, but you did not specify what you used as training data and what you used as test data. Please clarify.

If theoretical accuracy refers to the predicted accuracy on test data given the accuracy of the classifier on then training data, then of course we expect mu and beta power to be predictive, because the classifier is almost certainly using mu and beta power to discriminate between motor imagery and rest. Using CSP and LDA is a very common recipe for EEG-based BCIs, and it tends very often to converge on mu and beta desynchronization. This brings up another point which is that you should specify in your methods section which were the features that the classifier used. You don’t have to go into all of the details of the classifier, but at a minimum you should state what the features were. Did the classifier operate on the raw time series, or (more likely) on a time-frequency representation of the data?

p. 14, label says “FIGURE 3” but I think this is figure 2.

“Location” = electrode location? This was not always clear. Maybe just call it “electrode”.

What does the word “location” refer to in the section on "Perceived control and appropriation of the robotic hand”?

Are you using the word “location” in two different ways. I know that one is the electrode location, but then it seems you are using it for something else as well. This was confusing.

For spectral power calculations on the EEG data, did you normalize, i.e. divide by the standard deviation of the baseline? I realize that you did your power spectral analyses without baseline correction because the time leading up to the action was too heterogenous. But couldn’t you use a time window before even the start of the trial as your baseline, or perhaps during the two-second waiting period? Because power is typically higher at lower frequencies because of 1/f scaling, so it is more meaningful to express your power estimates as a Z score before comparing. It is not really valid to compare power at in different frequency bands. I.e. you comparing power in the mu band to power in the beta band in kind of meaningless. Mu band will always win, just because there is always more power at lower frequencies.

Figure 2 A and C: what are the units???

p. 18, line 16: What are the two variables precisely? One is the difference in interval estimates between the two tasks, but what is the other one? You just refer to the “difference between the real hand key press and the robotic hand key press”, but the difference in what? Spectral power? And what is each data point? A subject (I presume)? You just need to be more clear and specific.

p. 20, line 10: This is confusing. You write “we observed a greater desynchronization in the beta band than in the mu band.” Do you mean for real hand movements? I thought that you had observed greater desynchronization in the mu band compared to the beta band (see p. 19 line 15).

You state that there was no difference in the interval estimates for real button press versus motor imagery, but did you verify that you even obtained a canonical intentional binding effect in the first place? I did not see this anywhere. Maybe there was an IB effect for both real-hand and robotic-hand keypresses, in which case you might not see any difference. Or maybe neither of them had the effect. Don’t you need a passive viewing or involuntary-motor task to control for that?

p. 24, line 6: You talk about the “difference between the imagery phases and the rest phases”, but the difference in what?? You should always specify what variable you are talking about, even if you think it is obvious. Did you mean the difference in spectral power? If so then how measured? I.e. over what time interval / frequency range? And how did you compute your power estimates? Wavelets? Band-pass filter plus Hilbert transform? In general you have not been precise enough in describing your analyses.

Figure 5: units!!

When you talk about the “mu rhythm” and “beta rhythm” what do you mean? Do you mean power in the mu-band and power in the beta band? If so then you should say that. Just saying “mu rhythm” or “beta rhythm” is not specific enough. It’s unclear.

Figure 6A: I am not sure it is appropriate or optimal to use Pearson’s correlation coefficient on a discrete variable (like perceived control).

No detail was given about the robotic hand itself, and very little detail about the BMI that was controlling it. How long was the latency between the BMI decision and the movement of the robotic hand? This information was given in the discussion, but should appear in the methods section as well. How long did it take for the robotic hand to depress the key? Was it an abrupt movement, or did the robotic hand move slowly until the key was pressed? Or did it move at a speed that was determined by the degree of BMI control? Did the key produce an audible click sound when pressed? These might seem like trivial details, but they may play a role in determining the sense of agency over the robotic hand, and the subsequent sense of agency over the keypress made by the robotic hand.

Does the theoretical accuracy predict perceived control? This was significant in both experiments and simply suggests that better performance on the part of the classifier is associated with a stronger feeling of control over the BCI (which it should). A prior study that is directly relevant here is Schurger et al, Brain & Cognition (2016) which also looked at judgements of control over a BMI.

p. 26, lines 9-11: This sentence is simply confusing. I can’t make any sense of it. Please clarify. And what do you mean by the “estimated interval estimate”? Isn’t it just either the “estimated interval” or the “interval estimate”? And what is "the reported difficulty to make the hand performing the movement”? Does this refer to the robotic hand? Do you perhaps mean “the reported difficulty making the robotic hand perform the movement”?

p. 26, lines 24-25: "Results indicated that the perceived control was a better predictor of interval estimates than the reported difficulty.” Isn’t there guaranteed to be some collinearity in this regression since we know already that perceived control and reported difficulty are correlated? And how did you determine that perceived control was the better predictor? If they are collinear then this would be difficult to ascertain. In part it could just be that this was not clearly expressed in writing. The writing definitely could stand to be improved.

p. 27, line 5: “p = -2.360”???

Again, what are “location scores”? This was unclear to me given that you used the word “location” to refer to which electrode you were looking at.

Reviewer #2: In two studies the authors lay out the importance of one’s sense of agency (SoA) over one’s actions and in particular actions performed by assistive technology via a brain-computer interface. Furthermore, they explore how the performance of the BCI as well as one’s perceived control or fatigue affect one’s SoA for the observed actions and how this may affect one’s sense of ownership over the assistive technology.

• What are the main claims of the paper and how significant are they for the discipline?

- The main claims of the paper are that i) the performance of the BCI has immediate influence over one’s explicit SoA (judgment of agency - JoA), even though the BCI actions do not produce reafferences; ii) similarly, the performance of the BCI predicts one’s implicit SoA (or feeling of Agency – FoA), as determined using intentional binding (IB) as dependent variable; iii) the perceived effort or difficulty of completing the task with the BCI had a negative effect on the FoA, and that iv) an increased SoA also aids embodiment or ownership of the device.

Generally, these claims are very interesting for the discipline, as they attempt to disentangle different aspects of one’s SoA by combining implicit and explicit judgments and linking them to movement-related cortical desynchronization.

Overall, I am not convinced the authors have controlled for all aspects of their study to fully support these claims.

• Are the claims properly placed in the context of the previous literature? Have the authors treated the literature fairly?

- The introduction of the manuscript provides an overview of relevant literature on the sense of agency, intentional binding, and BCIs. While the authors aim to disentangle these different concepts in order to motivate their study design, I disagree with some of their key arguments here. I would appreciate some clarification on these points, as they are important for the study design, the interpretation of their results, as well as their implications.

Intentional binding

- I agree that it is important to separate the FoA from the JoA, as the authors do. However, I am not convinced that the “pre-reflective experience of” the FoA can be assessed using Intentional Binding. Indeed, I would think that e.g. Moore and Obhi (2012 Consciousness & Cognition) or Ebert and Wegner (2010, also C&C) would argue that IB is rather compatible with the JoA. (I have more comments on this for the analysis.) The FoA describes an implicit, on-going aspect of performing an action or making a movement. Neither of these points is true for IB, where the action has already been completed with the press of the button. I understand that this is a general discussion point, not specific to this study, but I think it should be considered.

- Comparing the JoA and IB also seems difficult, as the former starts with one’s movement intention and ends with the press of the button (“the hand moved according to my will”), whereas the latter only reflects the cause-and-effect of the button-press followed by a tone (“the delay that occurred between the robotic hand keypress and the tone”). It could therefore be argued that the JoA and the FoA measured here concern different processes and should not directly be compared (without further justification).

- With respect to further literature on the IB paradigm, Suzuki et al.’s findings (Psychological Science 2019) suggesting that IB may simply reflect “multisensory causal binding” would be relevant to include, particularly as the current paper does neither includes baseline estimations for action- or outcome-binding nor a passive control condition. The work by Rohde and Ernst (e.g. Behavioural Science, 2016) is also relevant.

Questionnaire Data

- Table 1 seems to be missing in my documents, so I am not entirely sure, my comments on these data are completely accurate. However, a general point to consider for the questionnaire is – what is the control condition and is there a control question? (also see my comment in the data analysis section.) Recently, the questionnaires used in the rubber hand illusion and comparable studies have come under (even more) scrutiny. It would be great, if the authors could include a brief discussion of their questionnaire results with respect to the points raised by Peter Lush (2020) “Demand Characteristics Confound the Rubber Hand Illusion” Collabra: Psychology.

• Do the data and analyses fully support the claims? If not, what other evidence is required?

- IB: Was there any evidence of an intentional binding effect? It would be interesting to see the distribution of the reported intervals – is it trimodal? (Cf. figures in Suzuki et al.) Was the interval actually reduced? As there is apparently no passive or control condition and no difference between human or robot hand movements, it is not clear if there was any effect at all. Perhaps this is something that can be explored in the existing data set. If there is no effect of binding, what does that mean with respect to the findings of study 2?

- Control condition: Moore and Obhi 2012 argue that “Out of all the factors that have been linked to intentional binding, it appears that the presence of efferent information is most central to the manifestation of intentional binding as, when efferent information is not present, such as in passive movement conditions, or passive observation of others, intentional binding is reduced or absent.” Should you not have included such a condition in order to verify an effect of IB?

- Agency Question: Were there any false positives or catch trials, in which the robotic hand moved without the user’s intention? If the hand only ever moves when participants “intend” to move, then the hand can never actually “move by its own”. Furthermore, if the question(s) are only asked after a successful triggering of the hand movement, is the latter answer not ~unattainable?

- Ownership Question: Is there a way to distinguish between ownership and agency in the current paradigm? Are the scores highly correlated? Can you exclude a counterargument such as demand characteristics being responsible for the ownership scores?

- Theoretical accuracy: The EEG (and EMG) methods are clearly described. The results with respect to mu- and beta-band suppression and electrode location are in-line with prior findings and it is good to see this BCI approach being applied to agency and ownership questions. I have a couple of questions which you can probably quite easily clarify. How does “theoretical accuracy” differ from (actual) classifier performance? Could you also briefly explain why you do not use cross-validation to measure the performance?

- On the same point, can you calculate the classifier performance during the actual experimental block? Does this match the training performance and is this a better or poorer indicator of perceived control?

- Theoretical accuracy and correlation to mu/beta oscillations: If I understand correctly, the classifier is trained to detect desynchronization in the mu/beta frequency bands. Performance is then quantified as theoretical accuracy. What does the correlation between theoretical accuracy and mu/beta oscillation actually tell us? Is this simply the confirmation that these are the trained criteria? (Apologies, if I simply missed the point here.)

- Sample size: The justification of the sample size is not very clear. Would it not be better to either calculate it based on prior or expected effect sizes and add a percentage of participants in case of BCI illiteracy? Alternatively, you could use a Bayesian approach and determine a cut-off criterion.

• PLOS ONE encourages authors to publish detailed protocols and algorithms as supporting information online. Do any particular methods used in the manuscript warrant such treatment? If a protocol is already provided, for example for a randomized controlled trial, are there any important deviations from it? If so, have the authors explained adequately why the deviations occurred?

- No issues.

• If the paper is considered unsuitable for publication in its present form, does the study itself show sufficient potential that the authors should be encouraged to resubmit a revised version?

- Not applicable.

• Are original data deposited in appropriate repositories and accession/version numbers provided for genes, proteins, mutants, diseases, etc.?

- Yes

• Are details of the methodology sufficient to allow the experiments to be reproduced?

- Yes

• Is the manuscript well organized and written clearly enough to be accessible to non-specialists?

- Yes

6. PLOS authors have the option to publish the peer review history of their article (what does this mean?). If published, this will include your full peer review and any attached files.

Reviewer #1: Yes: Aaron Schurger

Reviewer #2: No

---

## [Author Response · Author response to Decision Letter 0]

11 Aug 2020

Dear Editor,

We would like to thank you for your time on this manuscript. We have revised the manuscript according to the comments received from the two reviewers. You will find attached a point-by-point response to the reviewers’ comments. We hope that the manuscript has been improved and will be suitable for publication in Plos One. 

Sincerely,

Emilie A. Caspar, Albert De Beir, Gil Lauwers, Axel Cleeremans and Bram Vanderborght

Reviewer #1: PONE-D-20-06849

This study looked at the sense of agency for actions performed using an EEG-based brain-machine interface (BMI). The intentional binding effect was used as the primary dependent measure, where subjects directly estimate the interval between the cause (button press) and effect (auditory click). In two experiments there were no significant differences in interval estimates between motor actions and BMI-mediated actions, suggesting no difference in agency. The authors did, however, find relationships between other variables in the experiment, such as the sense of control, experienced difficulty, ownership of the robotic hand, and agency (as indexed by the interval estimates).

The paper is not very clearly written, and it was very difficult to make sense of it in places, often because crucial information was left out or not made clear, or jargon was undefined. It really reads like a first or second draft of a paper, rather than a paper that has been through a few rounds of internal review before being deemed ready to submit to a journal for peer review. Many of my comments are thus focused on the writing and on making things more clear. The paper does not make any strong claims, and the conclusions that are drawn seem reasonable given the data. The methods involved in controlling the robotic hand and the robotic hand itself were not given in sufficient detail (for example we do not know anything about the features used to discriminate motor imagery from rest conditions, even though this might be relevant to some of the results that involved correlation with the “theoretical accuracy”). The statistics appear to be appropriate and properly reported, except that for correlations with ordinal data (as in figure 6A) it can be better to use a non-parametric statistic like Spearman’s rho.

 → We thank the referee for this frank but sympathetic assessment and apologize for the quality of the language. We have revised the manuscript extensively, improving both language quality and presentation. We have also added information about the decoding algorithms we used and now hope the manuscript feels more straightforward. We reply to the referee’s specific comments below: 

Specific comments (in no particular order):

C1. p. 4, line 3: what is a “skin-based input”?

→ R1. In the paper of Coyle and colleagues, the authors used a system that was detecting when participants were tapping on their arm to produce an outcome (i.e. a tone). We have now added this description in the relevant passage, as follows: For instance, Coyle, Moore, Kristensson, Fletcher and Blackwell (2012) showed that intentional binding was stronger in a condition in which participants used a skin-based input system that detected when participants were tapping on their arm to produce a resulting tone rather than in a condition in which they were tapping on a button to produce a similar tone

C2. p. 6, line 21”main age” should be “mean age”

→ R2. Corrected.

C3. p. 7, bottom and p. 12, line 22, and figure 3

So they used bi-polar electrodes to record EMG over the flexor digitorum superficialis, but it seems that they just monitored it on-line or on a single-trial basis (says that they used the EMG traces off-line to remove trials with muscle activity). But they did not, it seems, check for sub-threshold muscle activity during motor imagery by looking at the average EMG power over all trials. To do this you have to take the sqrt of the sum of squares in a sliding window for each trial, before you average the trials together. You have to do this if you want to rule out any muscle activity, including sub-threshold muscle activity.

 Also it says that EMG data were high-pass filtered at 10 Hz, but then nothing else. You can’t really see muscle activity without at least rectifying the signal, or (better) converting to power.

→ R3. We agree with this comment. An important aspect of the methods that we did not mention in the manuscript was that there was a baseline correction for the analysis of the EMG. This information has now been added in the relevant section. However, even using a different approach to analyse the EMG data, we cannot entirely rule out the presence of muscular activity. The flexor digitorum superficialis does not control for the movement of the thumb for instance - a movement that we could only check with visual control for overts mouvements. There were also no electrodes placed on the left arm for instance. We agree that these shortcomings constitute a limitation and have now added a discussion paragraph regarding this aspect, as follows (page 42): “Finally, another limitation is that we could not entirely rule out the presence of sub-thresholded muscular activity that could have triggered the BMI during the task. The flexor digitorum superficialis where the electrodes were placed does not allow to record activity associated with the movement of the thumb. In addition, no electrodes were placed on the left arm. Those movements were only controled visually by the experimenter during the task.”. 

C4. p. 9, line 4: They checked to see if, after six training sessions, accuracy of the classifier was below chance. But that’s not what you want to do. You want to check to see if accuracy of the classifier is significantly *different* from chance. Performance of 51% correct might still be chance! You should always say “not different from chance” rather than “below chance”.

→ R4. We were not interested in including only participants that were above chance level, as is indeed usually done in the classic literature regarding BCI. Our aim was to include even people at the chance level in order to have variability in our sample, that is a distribution ranging from people only able to control the BCI at chance level to people having a good control of the BCI. This variability was of interest to us in order to evaluate how different degrees of control over the BCI result in a modulation of the sense of agency as well as in embodiment of the robotic hand. We further clarified this choice on page 12 as follows: “We chose to also accept participants who were at the chance level in order to create variability in the control of the BMI to be able to study how this variability influences other factors.” Our only criterion was excluding participants if they were lower than 50%. We agreed that performance should be significantly lower than 50% to truly consider that our exclusion criteria were respected, since indeed a score of 48 or 49% could still be considered as the chance level. We have now modified the sentence on page 12 as follows: “At most, six training sessions were performed. If the level of classification performance failed to reach chance level (i.e. 50%) after six sessions, the experiment was then terminated. ”

C5. p. 10, line 5: "red cross” should be “red arrow”

 → R5. Thank you for noticing this typo, we have now corrected it.

C6. It would be helpful and useful if you computed AUC for control of the robotic hand, just to give an idea of what level of control your subjects managed to achieve.

 → R6. We thank the referee for this suggestion. Computing the area under the curve indeed makes it possible to better describe a binary classifier. Although often used in data science, we found very few examples of this metric in our field. As the goal of this paper was to use the BCI merely as a method in behavioural research  rather than to develop a better BCI, we decided not to include the suggested AUC analyses.

C7. What do you mean by the “theoretical” accuracy score? This was not clear enough. On what data set was this theoretical accuracy determined? You should have a test set and a training set, but you did not specify what you used as training data and what you used as test data. Please clarify.

→ R7. Thank you for this comment. We agree that the theoretical accuracy score could be confusing, such as pointed out by Reviewer 2 as well. We have now decided to use the terminology “classification performance” for more clarity. We have also added more information about the training session, as follows, on page 12: “These training sessions were performed at least twice. The first training session was used to build an initial data set containing the EEG data of the subject and the known associated condition: “motor imagery of the right hand” or “being at rest”. This data set was then used to train the classifier. During the training of the classifier, a first theoretical indication of performance was obtained by cross-validation, which consisted in training the classifier using a subset of the data set, and of testing the obtained classifier on the remaining data. The second training session was used to properly evaluate the classification performance of the classifier. When the participant was asked to perform the training session a second time, the classifier was predicting in real time whether the participant was at rest or in right hand motor imagery condition. If classification performance was still low (around 50-60%), a new model was trained based on the additional training session, and combined with all previous training sessions. At most, six training sessions were performed.”

C8. If theoretical accuracy refers to the predicted accuracy on test data given the accuracy of the classifier on then training data, then of course we expect mu and beta power to be predictive, because the classifier is almost certainly using mu and beta power to discriminate between motor imagery and rest. Using CSP and LDA is a very common recipe for EEG-based BCIs, and it tends very often to converge on mu and beta desynchronization. This brings up another point which is that you should specify in your methods section which were the features that the classifier used. You don’t have to go into all of the details of the classifier, but at a minimum you should state what the features were. Did the classifier operate on the raw time series, or (more likely) on a time-frequency representation of the data?

→ R8. Thank you for this comment. We have now added more information about the classification performance, as follows, on page 11: “The feature extraction that followed each training session was carried out with a Variant of Common Spatial Pattern (CSP), and the classifier used  Linear Discriminant Analysis (LDA, see Kothe & Makeig, 2013). We used the Filter Bank Common Spatial Pattern (FBCSP) variant, which first separates the signal into different frequency bands before applying CSP on each band. Results in each frequency band are then automatically weighted in such a way that the two conditions are optimally discriminated. Two frequency bands were selected: 8-12Hz and 13-30Hz. FBCSP and LDA were applied on data collected within a 1s a time window. The position of this time window inside the - 3s of rest or right hand motor imagery was automatically optimized for each participant by sliding the time window, training the classifier, comparing achieved accuracy by cross-validation, and keeping the classifier giving the best classification accuracy. ”

C9. p. 14, label says “FIGURE 3” but I think this is figure 2.

 → R9. We have now corrected this typo. 

C10. “Location” = electrode location? This was not always clear. Maybe just call it “electrode”. What does the word “location” refer to in the section on "Perceived control and appropriation of the robotic hand”? Are you using the word “location” in two different ways. I know that one is the electrode location, but then it seems you are using it for something else as well. This was confusing.

→ R10. We agree with this suggestion and we have now used ‘electrode’ instead of ‘location’ when necessary, since location could also refer to the ‘location’ score from the rubber hand illusion questionnaire measuring the degree of appropriation of the robotic hand.  ‘Location’ when referring to the appropriation of the robotic hand was left as before since this is in accordance with the wording used in other papers on the RHI. 

C11. For spectral power calculations on the EEG data, did you normalize, i.e. divide by the standard deviation of the baseline? I realize that you did your power spectral analyses without baseline correction because the time leading up to the action was too heterogenous. But couldn’t you use a time window before even the start of the trial as your baseline, or perhaps during the two-second waiting period? Because power is typically higher at lower frequencies because of 1/f scaling, so it is more meaningful to express your power estimates as a Z score before comparing. It is not really valid to compare power at in different frequency bands. I.e. you comparing power in the mu band to power in the beta band in kind of meaningless. Mu band will always win, just because there is always more power at lower frequencies.

→ R11. Thank you for this insightful comment.Those two options are unfortunately not reliable. Regarding the 2s waiting period, even though we asked participants to relax, we did not specify exactly when the waiting period finished (e.g. with a sentence appearing on the screen for instance, explicitly instructing them to relax). In the BMI-generated action condition, we cannot be entirely sure of when participants start to use motor imagery. Some participants could for instance start trying to use motor imagery to make the robotic hand move after 1s or 1.5 s. Since we don’t have this information, we doubt we can use it  as a baseline. Regarding the start of the trial option, participants finished the former trial by validating their answer by pressing the ENTER keypress. They did not have to wait for a fixed period of time before starting the next trial by pressing the ENTER key again. Most of the participants started the next trial right away. Therefore, this cannot constitute a proper baseline period either. But we agree that for future studies we should integrate a fixed and more clear time period to obtain a proper baseline correction. 

We have followed the suggestion to normalize the data by transforming  the data associated with each frequency band into z scores. We carried all analyses again on the normalized data, and reported these analyses in our revision. 

C12. Figure 2 A and C: what are the units???

 → R12. This has now been added. 

C13. p. 18, line 16: What are the two variables precisely? One is the difference in interval estimates between the two tasks, but what is the other one? You just refer to the “difference between the real hand key press and the robotic hand key press”, but the difference in what? Spectral power? And what is each data point? A subject (I presume)? You just need to be more clear and specific.

→ R13. We apologize for the confusion. We have now modified the sentence as follows: “We additionally investigated, with Pearson correlations, whether or not the difference in power in the mu- and the beta-bands between the real hand keypress and the robotic hand keypress on C3, Cz and C4 for each participant could predict the difference in interval estimates between the same two tasks.”

C14. p. 20, line 10: This is confusing. You write “we observed a greater desynchronization in the beta band than in the mu band.” Do you mean for real hand movements? I thought that you had observed greater desynchronization in the mu band compared to the beta band (see p. 19 line 15).

→ R14. Those results have now changed because we have normalized the data within each frequency band. The new results and the adapted discussion are now reported in the revised version of the manuscript. Overall, power in the mu and the beta-band now do not significantly differ between each other. 

C15. You state that there was no difference in the interval estimates for real button press versus motor imagery, but did you verify that you even obtained a canonical intentional binding effect in the first place? I did not see this anywhere. Maybe there was an IB effect for both real-hand and robotic-hand keypresses, in which case you might not see any difference. Or maybe neither of them had the effect. Don’t you need a passive viewing or involuntary-motor task to control for that?

→ R15. We agree with this comment, which is similar to comment C31 from reviewer 2. In the classic IB literature, a comparison between two experimental conditions is necessary, generally an active and a passive condition, in order to detect changes in the sense of agency. In our study, we did not introduce a passive condition for the following reason: the experiment was already quite long and exhausting for participants. Adding additional experimental conditions would have led to even higher cognitive fatigue, which could have been detrimental for the interval estimate task. Our main aim was to investigate if agency would differ between the body-generated action condition and the BMI-generated action condition, and hence we did not introduce a control, passive condition which would have been critical to know if agency was ‘high’ in both conditions or ‘low’ in both conditions. Given the extensive existing literature on body-generated action and interval estimates, we considered that, like in the existing literature, this condition reflected a sort of ‘high agency condition’, since movements were voluntarily produced with participants’ own keypress. Based on this, we thus extrapolated that agency was also high in the BMI-generated action condition. We fully agree that this extrapolation may not be sufficient, and we have now added this comment as a limitation in the general discussion of the present paper, as follows: “A limitation of the present study is that we did not use a control, passive condition similarly to previous studies on interval estimates. Such passive conditions involve a lack of volition in the realization of the movement, for instance by using a TMS over the motor cortex (Haggard et al., 2002) or by having a motoric setup forcing the participant’s finger to press the key (e.g. Moore, Wegner & Haggard, 2009). Neither Study 1 nor Study 2, includes such passive conditions, thus precluding our conclusions on the extent to which a BMI-generated action condition does involve a high sense of agency. The main reason for not introducing such control conditions stemmed from the tiring nature of the task asked from participants (i.e. learning to control a robotic hand through a BMI) and its duration (i.e. 2 hours). Adding additional experimental conditions would have strongly increased the fatigue  of participants during the task. The existing literature on active, voluntary conditions in which participants use their own index finger to press a key whenever they want is very extensive and consistently points towards lower interval estimates in active conditions than in passive conditions, involving a higher sense of agency (see Moore & Obhi, 2012; Moore, 2016; Haggard, 2017 for reviews). We thus considered that the body-generated action condition used in the present study, which is entirely similar to the active conditions previously reported in the literature, would be the baseline associated with a high sense of agency. This condition was then compared to the BMI-generated action condition in order to evaluate if using a BMI would lead to a similarly high sense of agency than in the body-generated action condition. In Study 2, we assumed that results of Study 1 were replicated for the BMI-generated action condition, thus resulting in a sense of agency on both Days 1 and 2. However, to draw more reliable conclusions on the sense of agency for MBI-generated actions, a passive control condition should be added.”. 

C16. p. 24, line 6: You talk about the “difference between the imagery phases and the rest phases”, but the difference in what?? You should always specify what variable you are talking about, even if you think it is obvious. Did you mean the difference in spectral power? If so then how measured? I.e. over what time interval / frequency range? And how did you compute your power estimates? Wavelets? Band-pass filter plus Hilbert transform? In general you have not been precise enough in describing your analyses.

→ R16. We agree that we should have been more precise in the way we report our analyses. We have now added the following information: “We conducted a repeated measures ANOVA with Rhythm (Mu, Beta) and Electrode (C3, Cz, C4) as within-subject factors on the difference in spectral power in the mu- and beta-bands between the imagery phases and the rest phases during the training session. Data were normalized by subtracting the global mean from each data point and by dividing the result  by the global SD. ” Also, in the section entitled ‘electroencephalography recordings and processing’, we have now added more information on how the power estimates were computed. 

C17. Figure 5: units!!

 → R17. We have added the units.

C18. When you talk about the “mu rhythm” and “beta rhythm” what do you mean? Do you mean power in the mu-band and power in the beta band? If so then you should say that. Just saying “mu rhythm” or “beta rhythm” is not specific enough. It’s unclear.

→ R18. We have now changed the wording in the entire manuscript, as suggested. We hope that the manuscript is now clearer. 

C19. Figure 6A: I am not sure it is appropriate or optimal to use Pearson’s correlation coefficient on a discrete variable (like perceived control).

→ R19. We agree that there is an ongoing debate about the use of Pearson correlations for discrete variables that have 6 points or more or if the use of nonparametric tests is more appropriate. However, R1 is right that the use of Spearman’s Rho could be more adapted and we have now changed the manuscript where necessary by reporting results with Spearman’s Rho instead of Pearson r. 

C20. No detail was given about the robotic hand itself, and very little detail about the BMI that was controlling it. How long was the latency between the BMI decision and the movement of the robotic hand? This information was given in the discussion, but should appear in the methods section as well. How long did it take for the robotic hand to depress the key? Was it an abrupt movement, or did the robotic hand move slowly until the key was pressed? Or did it move at a speed that was determined by the degree of BMI control? Did the key produce an audible click sound when pressed? These might seem like trivial details, but they may play a role in determining the sense of agency over the robotic hand, and the subsequent sense of agency over the keypress made by the robotic hand.

→ We have now added an additional section in the method section regarding the robotic hand. We have mentioned its features, the publications that detail how to build it and the time to press the keys. The time the robotic hand takes to press the key is indeed very important regarding the time estimation that participants have to make. We set-up this timing to 900 ms in order to have the robotic hand redressing the finger only after the beep occured on each trial and so to avoid temporal distortion for interval estimates. 

C21. Does the theoretical accuracy predict perceived control? This was significant in both experiments and simply suggests that better performance on the part of the classifier is associated with a stronger feeling of control over the BCI (which it should). A prior study that is directly relevant here is Schurger et al, Brain & Cognition (2016) which also looked at judgements of control over a BMI.

→ R21. There was indeed a significant correlation between classification performance and perceived control. We have now added the requested reference in the discussion, since it was indeed relevant: “This result is in line with a former study, which showed that participants are good at evaluating their ability to control a BMI even if they do not receive an online feedback on their performance (Schurger, Gale, Gozel, & Blanke, 2017). ”

C22. p. 26, lines 9-11: This sentence is simply confusing. I can’t make any sense of it. Please clarify. And what do you mean by the “estimated interval estimate”? Isn’t it just either the “estimated interval” or the “interval estimate”? And what is "the reported difficulty to make the hand performing the movement”? Does this refer to the robotic hand? Do you perhaps mean “the reported difficulty making the robotic hand perform the movement”?

→ R22. We have now changed these sentences as follows: “We then performed non-parametric correlation with Spearman's rho (⍴) between the reported interval estimates, the perceived control over the movement of the robotic hand and the reported difficulty to make the robotic hand perform the movement, on a trial-by-trial basis. ”

C23. p. 26, lines 24-25: "Results indicated that the perceived control was a better predictor of interval estimates than the reported difficulty.” Isn’t there guaranteed to be some collinearity in this regression since we know already that perceived control and reported difficulty are correlated? And how did you determine that perceived control was the better predictor? If they are collinear then this would be difficult to ascertain. In part it could just be that this was not clearly expressed in writing. The writing definitely could stand to be improved.

 → R23. We have now reported the VIF that rules out a collinearity effect (all VIF are always =< 2).  However, we do agree that both factors (i.e. perceived control and reported difficulty) could play antagonist roles on the implicit sense of agency: while higher perceived control could boost the implicit sense of agency, higher difficulty could reduce it. 

C24. p. 27, line 5: “p = -2.360”???

 → R24. We have now corrected this typo. 

C25. Again, what are “location scores”? This was unclear to me given that you used the word “location” to refer to which electrode you were looking at.

→ R25. We have now used the word ‘electrode’ when it referred to the electrode location. The word ‘location’ now only refers to the location scores from the questionnaire assessing the appropriation of the robotic hand. 

------

Reviewer #2: In two studies the authors lay out the importance of one’s sense of agency (SoA) over one’s actions and in particular actions performed by assistive technology via a brain-computer interface. Furthermore, they explore how the performance of the BCI as well as one’s perceived control or fatigue affect one’s SoA for the observed actions and how this may affect one’s sense of ownership over the assistive technology.

• What are the main claims of the paper and how significant are they for the discipline?

The main claims of the paper are that i) the performance of the BCI has immediate influence over one’s explicit SoA (judgment of agency - JoA), even though the BCI actions do not produce reafferences; ii) similarly, the performance of the BCI predicts one’s implicit SoA (or feeling of Agency – FoA), as determined using intentional binding (IB) as dependent variable; iii) the perceived effort or difficulty of completing the task with the BCI had a negative effect on the FoA, and that iv) an increased SoA also aids embodiment or ownership of the device.

Generally, these claims are very interesting for the discipline, as they attempt to disentangle different aspects of one’s SoA by combining implicit and explicit judgments and linking them to movement-related cortical desynchronization.

Overall, I am not convinced the authors have controlled for all aspects of their study to fully support these claims.

→ Thanks for judging our work and claims as interesting. We hope our revision further supports these claims. 

• Are the claims properly placed in the context of the previous literature? Have the authors treated the literature fairly?

C27. The introduction of the manuscript provides an overview of relevant literature on the sense of agency, intentional binding, and BCIs. While the authors aim to disentangle these different concepts in order to motivate their study design, I disagree with some of their key arguments here. I would appreciate some clarification on these points, as they are important for the study design, the interpretation of their results, as well as their implications.

Intentional binding

C26. I agree that it is important to separate the FoA from the JoA, as the authors do. However, I am not convinced that the “pre-reflective experience of” the FoA can be assessed using Intentional Binding. Indeed, I would think that e.g. Moore and Obhi (2012 Consciousness & Cognition) or Ebert and Wegner (2010, also C&C) would argue that IB is rather compatible with the JoA. (I have more comments on this for the analysis.) The FoA describes an implicit, on-going aspect of performing an action or making a movement. Neither of these points is true for IB, where the action has already been completed with the press of the button. I understand that this is a general discussion point, not specific to this study, but I think it should be considered.

→ R26.This is indeed a fair point and a lot of explicit measures have also been criticized in terms of how much they relate to agency, since classically authors use different questions that are adapted to their experimental paradigm (i.e. “Do you feel in control of the action”, “Do you feel that you are the authors of the action”, “Do you feel that who caused the outcomes”? etc.). However, since the point is somewhat derivative with respect to our main goals,  we chose not to further develop this discussion point. 

C27.  Comparing the JoA and IB also seems difficult, as the former starts with one’s movement intention and ends with the press of the button (“the hand moved according to my will”), whereas the latter only reflects the cause-and-effect of the button-press followed by a tone (“the delay that occurred between the robotic hand keypress and the tone”). It could therefore be argued that the JoA and the FoA measured here concern different processes and should not directly be compared (without further justification).

→ R27. We agree with the reviewer’s comment and that’s why we do not directly compare them. We nonetheless evaluate the correlations between these two experiences of the self, when one is producing an action. Several earlier  papers have evaluated (through correlations) how these two experiences of the self, despite measuring different aspects, relate and that is the procedure we have used here. 

C28. With respect to further literature on the IB paradigm, Suzuki et al.’s findings (Psychological Science 2019) suggesting that IB may simply reflect “multisensory causal binding” would be relevant to include, particularly as the current paper does neither includes baseline estimations for action- or outcome-binding nor a passive control condition. The work by Rohde and Ernst (e.g. Behavioural Science, 2016) is also relevant.

→ R28. According to the study of Suzuki et al 2019, any study wishing to draw conclusions specific to voluntary action, or similar theoretical constructs, needs to be based on the comparison of carefully matched conditions that should ideally differ in just one critical aspect. Here, our conditions only differed in the sensorimotor information coming from the keypress; they were similar in terms of causality and volition. We are aware of the current debate over the effect of causality on IB. While some authors argued that causality plays an important role (e.g. Buehner & Humphrey, 2009) in explaining variability in IB between 2 experimental conditions, other authors have shown that when causality is controlled as well as other features between two experimental conditions, and that only intentionality remains, a difference in IB still occur (Caspar, Cleeremans & Haggard, 2018).  - study 2). Very recent studies come to the conclusion that even if several factors, such as causality, play a role in the modulation of binding, intentionality also plays a key role (e.g., Lorimer, … , Buehner, 2020). However, since this debate is not the aim of the present paper, we do not discuss this literature in the present paper.

Questionnaire Data

C29. Table 1 seems to be missing in my documents, so I am not entirely sure, my comments on these data are completely accurate. However, a general point to consider for the questionnaire is – what is the control condition and is there a control question? (also see my comment in the data analysis section.) Recently, the questionnaires used in the rubber hand illusion and comparable studies have come under (even more) scrutiny. It would be great, if the authors could include a brief discussion of their questionnaire results with respect to the points raised by Peter Lush (2020) “Demand Characteristics Confound the Rubber Hand Illusion” Collabra: Psychology.

→ R29. We apologize for the omission. Table 1 was indeed missing and has now been added in the manuscript.  As far as we understand it, the critical difference between our design and that of Lush (2020) is that we never mentioned anything to our participants regarding the ‘robotic’ hand illusion or even that they would be given a questionnaire at the end of the experiment evaluating their appropriation of the robotic hand. We even avoided putting the classical blanket between the robotic hand and the participant to cover the participant’s arm, in order to avoid them thinking that we would evaluate the robotic hand as part of their own body. Our participants thus had a limited room for creating expectancies regarding the questionnaires and the robotic hand. However, we agree that controlling for expectancies is relevant in the case of the rubber hand illusion. We have now integrated this discussion in the general discussion section, as follows: “Several articles have argued that demand characteristics (Orne, 1962) and expectancies can predict scores on the classical rubber hand illusion (RHI) questionnaires (Lush, 2020). In the present study, we limited as much as possible the effects of both demand characteristics and expectancies on participants’ answer to the questionnaire related to the appropriation of the robotic hand: (1) Participants were not told anything about the rubber hand illusion or the fact that some people may experience the robotic hand as a part of their body during the experiment; (2) We did not tell them in advance that they would have to fill in a questionnaire regarding their appropriation of the robotic hand at the end of the experiment; (3) We avoided creating a proper classical ‘rubber hand illusion’: the robotic hand was placed at a congruent place on the table regarding the participant’s arm, but we did not use the blanket to cover their real arm, neither we used a paintbrush to induce an illusion. This procedure limited the possibility for participants to guess what we would assess since in the classical rubber hand illusion, placing the blanket on the participant’s arm make them realize that we investigate to what extent they appropriate this hand in their body schema. However, we cannot fully rule out the influence of demand characteristics and expectancies, as participants perceiving a better control over the hand, even without knowing their own classification performance, could have intuitively indicated higher scores on the questionnaires. Control studies could thus be performed to manipulate demand characteristics in order to evaluate its effect on the appropriation of a robotic hand in a brain-machine interface procedure (Lush, 2020).”

• Do the data and analyses fully support the claims? If not, what other evidence is required?

C30. IB: Was there any evidence of an intentional binding effect? It would be interesting to see the distribution of the reported intervals – is it trimodal? (Cf. figures in Suzuki et al.) Was the interval actually reduced? As there is apparently no passive or control condition and no difference between human or robot hand movements, it is not clear if there was any effect at all. Perhaps this is something that can be explored in the existing data set. If there is no effect of binding, what does that mean with respect to the findings of study 2?

→ R30. With respect to Study 2, we cannot infer that the sense of agency was ‘strong’ or ‘weak’ on either training days, which we did not argue for in our discussion. Rather, we can say that having a good control over the BMI boosts sense of agency, as measured through the method of interval estimates. We had three intervals: we have run those analyses with delays as an additional within-subject factor and have observed that IE were longer for the BMI condition for the 100ms-interval than for the real hand condition, not statistically different for the 400ms-interval between the two conditions, and shorter for the BMI condition for the 700ms-interval than for the real hand condition. In our opinion, nothing reliable could be drawn from those analyses regarding binding, and we therefore do not mention these data in the paper. But we indeed agree that this is a critical point in our paper, which is now further elaborated on in the discussion section. 

C31.  Control condition: Moore and Obhi 2012 argue that “Out of all the factors that have been linked to intentional binding, it appears that the presence of efferent information is most central to the manifestation of intentional binding as, when efferent information is not present, such as in passive movement conditions, or passive observation of others, intentional binding is reduced or absent.” Should you not have included such a condition in order to verify an effect of IB?

→ R31. We agree with R2 that we should have included such control conditions. The main reason for not including them was the exhausting nature of the task (learning to control a BMI) in addition to the duration of the task (about 2 hours). We thus considered that the body-generated action condition, which was similar to a host of previous studies using temporal binding was our baseline condition, corresponding to a high sense of agency. We have now added this point as a limitation in our study in the discussion section: “A limitation of the present study is that we did not use a control, passive condition similarly to previous studies on interval estimates. Such passive conditions involve a lack of volition in the realization of the movement, for instance by using a TMS over the motor cortex (Haggard et al., 2002) or by having a motoric setup forcing the participant’s finger to press the key (e.g. Moore, Wegner & Haggard, 2009). Neither Study 1 nor Study 2, includes such passive conditions, thus precluding our conclusions on the extent to which a BMI-generated action condition does involve a high sense of agency. The main reason for not introducing such control conditions stemmed from the tiring nature of the task asked from participants (i.e. learning to control a robotic hand through a BMI) and its duration (i.e. 2 hours). Adding additional experimental conditions would have strongly increased the fatigue  of participants during the task. The existing literature on active, voluntary conditions in which participants use their own index finger to press a key whenever they want is very extensive and consistently points towards lower interval estimates in active conditions than in passive conditions, involving a higher sense of agency (see Moore & Obhi, 2012; Moore, 2016; Haggard, 2017 for reviews). We thus considered that the body-generated action condition used in the present study, which is entirely similar to the active conditions previously reported in the literature, would be the baseline associated with a high sense of agency. This condition was then compared to the BMI-generated action condition in order to evaluate if using a BMI would lead to a similarly high sense of agency than in the body-generated action condition. In Study 2, we assumed that results of Study 1 were replicated for the BMI-generated action condition, thus resulting in a sense of agency on both Days 1 and 2. However, to draw more reliable conclusions on the sense of agency for MBI-generated actions, a passive control condition should be added.”

C32. Agency Question: Were there any false positives or catch trials, in which the robotic hand moved without the user’s intention? If the hand only ever moves when participants “intend” to move, then the hand can never actually “move by its own”. Furthermore, if the question(s) are only asked after a successful triggering of the hand movement, is the latter answer not ~unattainable?

→ R32. There were no catch trials per se, in the sense that we did not introduce trials in which the robotic hand moved by its own in the experimental design. There were however false positives, induced by the participants’ lack of control over the robotic hand. The explicit question about control was developed to find out when those false positives occurred: With this question, participants could tell us if each trial was a false positive (i.e. the robotic hand moved by ‘its own’) or an intended trial. We thus indeed rely on participant’s perception to know if the trial was a ‘false positive’ or not. 

C33.  Ownership Question: Is there a way to distinguish between ownership and agency in the current paradigm? Are the scores highly correlated? Can you exclude a counterargument such as demand characteristics being responsible for the ownership scores?

→ R33. Agency and ownership scores were indeed correlated, such as in several previous studies. We can’t really ensure that demand characteristics played a role in the ownership scores, but we tried,  however, to limit its impact more than in former studies (see also R29). This has been added in the discussion section.

C34.  Theoretical accuracy: The EEG (and EMG) methods are clearly described. The results with respect to mu- and beta-band suppression and electrode location are in-line with prior findings and it is good to see this BCI approach being applied to agency and ownership questions. I have a couple of questions which you can probably quite easily clarify. How does “theoretical accuracy” differ from (actual) classifier performance? Could you also briefly explain why you do not use cross-validation to measure the performance?

→ R34. In fact the theoretical accuracy score was the name that we gave to the classifier performance. We agree that this may have been confusing and, in line with Reviewer 1, we have now replaced the word ‘theoretical accuracy’ by ‘classification performance throughout the entire manuscript for the sake of clarity. More information has now been added in the manuscript regarding how the classification performance was calculated and regarding cross-validation: “We used the Filter Bank Common Spatial Pattern (FBCSP) variant, which first separates the signal into different frequency bands before applying CSP on each band. Results in each frequency band are then automatically weighted in such a way that the two conditions are optimally discriminated. Two frequency bands were selected: 8-12Hz and 13-30Hz. FBCSP and LDA were applied on data collected within a 1s a time window. The position of this time window inside the - 3s of rest or right hand motor imagery was automatically optimized for each participant by sliding the time window, training the classifier, comparing achieved accuracy by cross-validation, and keeping the classifier giving the best classification accuracy. ”  and “These training sessions were performed at least twice. The first training session was used to build an initial data set containing the EEG data of the subject and the known associated condition: “motor imagery of the right hand” or “being at rest”. This data set was then used to train the classifier. During the training of the classifier, a first theoretical indication of performance was obtained by cross-validation, which consisted in training the classifier using a subset of the data set, and of testing the obtained classifier on the remaining data. The second training session was used to properly evaluate the classification performance of the classifier. When the participant was asked to perform the training session a second time, the classifier was predicting in real time whether the participant was at rest or in right hand motor imagery condition. If classification performance was still low (around 50-60%), a new model was trained based on the additional training session, and combined with all previous training sessions. At most, six training sessions were performed. If the level of classification performance failed to reach chance level (i.e. 50%) after six sessions, the experiment was then terminated. ”

C35.  On the same point, can you calculate the classifier performance during the actual experimental block? Does this match the training performance and is this a better or poorer indicator of perceived control?

→ R35. The classifier performance was calculated based on the discrepancy/predictability between the model and participant’s performance during the training session. It means that in the training session, the classifier could know what was the expected fit because we alternated a cross appearing on the screen (corresponding to the rest phase) and an arrow appearing on that cross (corresponding to the motor imagery phase). This is what is represented on Figure 1A and Figure 1B. In the experimental block, participants could manipulate the robotic hand as they wanted, meaning that there was no expected match between a specific stimulus appearing on the screen and participants’ willingness to make the robotic hand move or not. It was thus not possible to calculate a score of classification performance during the experimental block, since we could not ask participants to report continuously when they were imagining the ‘rest phase’ to avoid making the robotic hand moving. 

C36. Theoretical accuracy and correlation to mu/beta oscillations: If I understand correctly, the classifier is trained to detect desynchronization in the mu/beta frequency bands. Performance is then quantified as theoretical accuracy. What does the correlation between theoretical accuracy and mu/beta oscillation actually tell us? Is this simply the confirmation that these are the trained criteria? (Apologies, if I simply missed the point here.)

→ R36. As a reminder, we have now changed the word ‘theoretical accuracy’ into ‘classification performance’ for more clarity. In fact, the feature extraction with CSP and LDA is not directly linked to mu and beta oscillations. However, since we filter the data based on the mu and the beta-bands, this means the classification performance score could be related to oscillations in the mu- and beta- bands. The correlation thus suggests that CSP+LDA correctly identifies the changes in mu and beta oscillations. According to the comment of Reviewer 1 (see R7 and R8), we have now added more information regarding the classification performance in the manuscript. We hope that this section is now clearer. 

- C37. Sample size: The justification of the sample size is not very clear. Would it not be better to either calculate it based on prior or expected effect sizes and add a percentage of participants in case of BCI illiteracy? Alternatively, you could use a Bayesian approach and determine a cut-off criterion.

→ R37. We agree with R2 but we  did not have  a strong prior regarding the expected effect sizes since no previous studies used a similar method and since we had no idea how many participants would be lower or higher than the chance level with our BMI. We did not have to exclude participants with a classification performance score lower than 50% and thus did not have to remove participants. Future studies using a similar approach will include a-priori computation of the sample size.

Dear Editor,

We would like to thank you for your time on this manuscript. We have revised the manuscript according to the comments received from the two reviewers. You will find attached a point-by-point response to the reviewers’ comments. We hope that the manuscript has been improved and will be suitable for publication in Plos One. 

Sincerely,

Emilie A. Caspar, Albert De Beir, Gil Lauwers, Axel Cleeremans and Bram Vanderborght

Reviewer #1: PONE-D-20-06849

This study looked at the sense of agency for actions performed using an EEG-based brain-machine interface (BMI). The intentional binding effect was used as the primary dependent measure, where subjects directly estimate the interval between the cause (button press) and effect (auditory click). In two experiments there were no significant differences in interval estimates between motor actions and BMI-mediated actions, suggesting no difference in agency. The authors did, however, find relationships between other variables in the experiment, such as the sense of control, experienced difficulty, ownership of the robotic hand, and agency (as indexed by the interval estimates).

The paper is not very clearly written, and it was very difficult to make sense of it in places, often because crucial information was left out or not made clear, or jargon was undefined. It really reads like a first or second draft of a paper, rather than a paper that has been through a few rounds of internal review before being deemed ready to submit to a journal for peer review. Many of my comments are thus focused on the writing and on making things more clear. The paper does not make any strong claims, and the conclusions that are drawn seem reasonable given the data. The methods involved in controlling the robotic hand and the robotic hand itself were not given in sufficient detail (for example we do not know anything about the features used to discriminate motor imagery from rest conditions, even though this might be relevant to some of the results that involved correlation with the “theoretical accuracy”). The statistics appear to be appropriate and properly reported, except that for correlations with ordinal data (as in figure 6A) it can be better to use a non-parametric statistic like Spearman’s rho.

 → We thank the referee for this frank but sympathetic assessment and apologize for the quality of the language. We have revised the manuscript extensively, improving both language quality and presentation. We have also added information about the decoding algorithms we used and now hope the manuscript feels more straightforward. We reply to the referee’s specific comments below: 

Specific comments (in no particular order):

C1. p. 4, line 3: what is a “skin-based input”?

→ R1. In the paper of Coyle and colleagues, the authors used a system that was detecting when participants were tapping on their arm to produce an outcome (i.e. a tone). We have now added this description in the relevant passage, as follows: For instance, Coyle, Moore, Kristensson, Fletcher and Blackwell (2012) showed that intentional binding was stronger in a condition in which participants used a skin-based input system that detected when participants were tapping on their arm to produce a resulting tone rather than in a condition in which they were tapping on a button to produce a similar tone

C2. p. 6, line 21”main age” should be “mean age”

→ R2. Corrected.

C3. p. 7, bottom and p. 12, line 22, and figure 3

So they used bi-polar electrodes to record EMG over the flexor digitorum superficialis, but it seems that they just monitored it on-line or on a single-trial basis (says that they used the EMG traces off-line to remove trials with muscle activity). But they did not, it seems, check for sub-threshold muscle activity during motor imagery by looking at the average EMG power over all trials. To do this you have to take the sqrt of the sum of squares in a sliding window for each trial, before you average the trials together. You have to do this if you want to rule out any muscle activity, including sub-threshold muscle activity.

 Also it says that EMG data were high-pass filtered at 10 Hz, but then nothing else. You can’t really see muscle activity without at least rectifying the signal, or (better) converting to power.

→ R3. We agree with this comment. An important aspect of the methods that we did not mention in the manuscript was that there was a baseline correction for the analysis of the EMG. This information has now been added in the relevant section. However, even using a different approach to analyse the EMG data, we cannot entirely rule out the presence of muscular activity. The flexor digitorum superficialis does not control for the movement of the thumb for instance - a movement that we could only check with visual control for overts mouvements. There were also no electrodes placed on the left arm for instance. We agree that these shortcomings constitute a limitation and have now added a discussion paragraph regarding this aspect, as follows (page 42): “Finally, another limitation is that we could not entirely rule out the presence of sub-thresholded muscular activity that could have triggered the BMI during the task. The flexor digitorum superficialis where the electrodes were placed does not allow to record activity associated with the movement of the thumb. In addition, no electrodes were placed on the left arm. Those movements were only controled visually by the experimenter during the task.”. 

C4. p. 9, line 4: They checked to see if, after six training sessions, accuracy of the classifier was below chance. But that’s not what you want to do. You want to check to see if accuracy of the classifier is significantly *different* from chance. Performance of 51% correct might still be chance! You should always say “not different from chance” rather than “below chance”.

→ R4. We were not interested in including only participants that were above chance level, as is indeed usually done in the classic literature regarding BCI. Our aim was to include even people at the chance level in order to have variability in our sample, that is a distribution ranging from people only able to control the BCI at chance level to people having a good control of the BCI. This variability was of interest to us in order to evaluate how different degrees of control over the BCI result in a modulation of the sense of agency as well as in embodiment of the robotic hand. We further clarified this choice on page 12 as follows: “We chose to also accept participants who were at the chance level in order to create variability in the control of the BMI to be able to study how this variability influences other factors.” Our only criterion was excluding participants if they were lower than 50%. We agreed that performance should be significantly lower than 50% to truly consider that our exclusion criteria were respected, since indeed a score of 48 or 49% could still be considered as the chance level. We have now modified the sentence on page 12 as follows: “At most, six training sessions were performed. If the level of classification performance failed to reach chance level (i.e. 50%) after six sessions, the experiment was then terminated. ”

C5. p. 10, line 5: "red cross” should be “red arrow”

 → R5. Thank you for noticing this typo, we have now corrected it.

C6. It would be helpful and useful if you computed AUC for control of the robotic hand, just to give an idea of what level of control your subjects managed to achieve.

 → R6. We thank the referee for this suggestion. Computing the area under the curve indeed makes it possible to better describe a binary classifier. Although often used in data science, we found very few examples of this metric in our field. As the goal of this paper was to use the BCI merely as a method in behavioural research  rather than to develop a better BCI, we decided not to include the suggested AUC analyses.

C7. What do you mean by the “theoretical” accuracy score? This was not clear enough. On what data set was this theoretical accuracy determined? You should have a test set and a training set, but you did not specify what you used as training data and what you used as test data. Please clarify.

→ R7. Thank you for this comment. We agree that the theoretical accuracy score could be confusing, such as pointed out by Reviewer 2 as well. We have now decided to use the terminology “classification performance” for more clarity. We have also added more information about the training session, as follows, on page 12: “These training sessions were performed at least twice. The first training session was used to build an initial data set containing the EEG data of the subject and the known associated condition: “motor imagery of the right hand” or “being at rest”. This data set was then used to train the classifier. During the training of the classifier, a first theoretical indication of performance was obtained by cross-validation, which consisted in training the classifier using a subset of the data set, and of testing the obtained classifier on the remaining data. The second training session was used to properly evaluate the classification performance of the classifier. When the participant was asked to perform the training session a second time, the classifier was predicting in real time whether the participant was at rest or in right hand motor imagery condition. If classification performance was still low (around 50-60%), a new model was trained based on the additional training session, and combined with all previous training sessions. At most, six training sessions were performed.”

C8. If theoretical accuracy refers to the predicted accuracy on test data given the accuracy of the classifier on then training data, then of course we expect mu and beta power to be predictive, because the classifier is almost certainly using mu and beta power to discriminate between motor imagery and rest. Using CSP and LDA is a very common recipe for EEG-based BCIs, and it tends very often to converge on mu and beta desynchronization. This brings up another point which is that you should specify in your methods section which were the features that the classifier used. You don’t have to go into all of the details of the classifier, but at a minimum you should state what the features were. Did the classifier operate on the raw time series, or (more likely) on a time-frequency representation of the data?

→ R8. Thank you for this comment. We have now added more information about the classification performance, as follows, on page 11: “The feature extraction that followed each training session was carried out with a Variant of Common Spatial Pattern (CSP), and the classifier used  Linear Discriminant Analysis (LDA, see Kothe & Makeig, 2013). We used the Filter Bank Common Spatial Pattern (FBCSP) variant, which first separates the signal into different frequency bands before applying CSP on each band. Results in each frequency band are then automatically weighted in such a way that the two conditions are optimally discriminated. Two frequency bands were selected: 8-12Hz and 13-30Hz. FBCSP and LDA were applied on data collected within a 1s a time window. The position of this time window inside the - 3s of rest or right hand motor imagery was automatically optimized for each participant by sliding the time window, training the classifier, comparing achieved accuracy by cross-validation, and keeping the classifier giving the best classification accuracy. ”

C9. p. 14, label says “FIGURE 3” but I think this is figure 2.

 → R9. We have now corrected this typo. 

C10. “Location” = electrode location? This was not always clear. Maybe just call it “electrode”. What does the word “location” refer to in the section on "Perceived control and appropriation of the robotic hand”? Are you using the word “location” in two different ways. I know that one is the electrode location, but then it seems you are using it for something else as well. This was confusing.

→ R10. We agree with this suggestion and we have now used ‘electrode’ instead of ‘location’ when necessary, since location could also refer to the ‘location’ score from the rubber hand illusion questionnaire measuring the degree of appropriation of the robotic hand.  ‘Location’ when referring to the appropriation of the robotic hand was left as before since this is in accordance with the wording used in other papers on the RHI. 

C11. For spectral power calculations on the EEG data, did you normalize, i.e. divide by the standard deviation of the baseline? I realize that you did your power spectral analyses without baseline correction because the time leading up to the action was too heterogenous. But couldn’t you use a time window before even the start of the trial as your baseline, or perhaps during the two-second waiting period? Because power is typically higher at lower frequencies because of 1/f scaling, so it is more meaningful to express your power estimates as a Z score before comparing. It is not really valid to compare power at in different frequency bands. I.e. you comparing power in the mu band to power in the beta band in kind of meaningless. Mu band will always win, just because there is always more power at lower frequencies.

→ R11. Thank you for this insightful comment.Those two options are unfortunately not reliable. Regarding the 2s waiting period, even though we asked participants to relax, we did not specify exactly when the waiting period finished (e.g. with a sentence appearing on the screen for instance, explicitly instructing them to relax). In the BMI-generated action condition, we cannot be entirely sure of when participants start to use motor imagery. Some participants could for instance start trying to use motor imagery to make the robotic hand move after 1s or 1.5 s. Since we don’t have this information, we doubt we can use it  as a baseline. Regarding the start of the trial option, participants finished the former trial by validating their answer by pressing the ENTER keypress. They did not have to wait for a fixed period of time before starting the next trial by pressing the ENTER key again. Most of the participants started the next trial right away. Therefore, this cannot constitute a proper baseline period either. But we agree that for future studies we should integrate a fixed and more clear time period to obtain a proper baseline correction. 

We have followed the suggestion to normalize the data by transforming  the data associated with each frequency band into z scores. We carried all analyses again on the normalized data, and reported these analyses in our revision. 

C12. Figure 2 A and C: what are the units???

 → R12. This has now been added. 

C13. p. 18, line 16: What are the two variables precisely? One is the difference in interval estimates between the two tasks, but what is the other one? You just refer to the “difference between the real hand key press and the robotic hand key press”, but the difference in what? Spectral power? And what is each data point? A subject (I presume)? You just need to be more clear and specific.

→ R13. We apologize for the confusion. We have now modified the sentence as follows: “We additionally investigated, with Pearson correlations, whether or not the difference in power in the mu- and the beta-bands between the real hand keypress and the robotic hand keypress on C3, Cz and C4 for each participant could predict the difference in interval estimates between the same two tasks.”

C14. p. 20, line 10: This is confusing. You write “we observed a greater desynchronization in the beta band than in the mu band.” Do you mean for real hand movements? I thought that you had observed greater desynchronization in the mu band compared to the beta band (see p. 19 line 15).

→ R14. Those results have now changed because we have normalized the data within each frequency band. The new results and the adapted discussion are now reported in the revised version of the manuscript. Overall, power in the mu and the beta-band now do not significantly differ between each other. 

C15. You state that there was no difference in the interval estimates for real button press versus motor imagery, but did you verify that you even obtained a canonical intentional binding effect in the first place? I did not see this anywhere. Maybe there was an IB effect for both real-hand and robotic-hand keypresses, in which case you might not see any difference. Or maybe neither of them had the effect. Don’t you need a passive viewing or involuntary-motor task to control for that?

→ R15. We agree with this comment, which is similar to comment C31 from reviewer 2. In the classic IB literature, a comparison between two experimental conditions is necessary, generally an active and a passive condition, in order to detect changes in the sense of agency. In our study, we did not introduce a passive condition for the following reason: the experiment was already quite long and exhausting for participants. Adding additional experimental conditions would have led to even higher cognitive fatigue, which could have been detrimental for the interval estimate task. Our main aim was to investigate if agency would differ between the body-generated action condition and the BMI-generated action condition, and hence we did not introduce a control, passive condition which would have been critical to know if agency was ‘high’ in both conditions or ‘low’ in both conditions. Given the extensive existing literature on body-generated action and interval estimates, we considered that, like in the existing literature, this condition reflected a sort of ‘high agency condition’, since movements were voluntarily produced with participants’ own keypress. Based on this, we thus extrapolated that agency was also high in the BMI-generated action condition. We fully agree that this extrapolation may not be sufficient, and we have now added this comment as a limitation in the general discussion of the present paper, as follows: “A limitation of the present study is that we did not use a control, passive condition similarly to previous studies on interval estimates. Such passive conditions involve a lack of volition in the realization of the movement, for instance by using a TMS over the motor cortex (Haggard et al., 2002) or by having a motoric setup forcing the participant’s finger to press the key (e.g. Moore, Wegner & Haggard, 2009). Neither Study 1 nor Study 2, includes such passive conditions, thus precluding our conclusions on the extent to which a BMI-generated action condition does involve a high sense of agency. The main reason for not introducing such control conditions stemmed from the tiring nature of the task asked from participants (i.e. learning to control a robotic hand through a BMI) and its duration (i.e. 2 hours). Adding additional experimental conditions would have strongly increased the fatigue  of participants during the task. The existing literature on active, voluntary conditions in which participants use their own index finger to press a key whenever they want is very extensive and consistently points towards lower interval estimates in active conditions than in passive conditions, involving a higher sense of agency (see Moore & Obhi, 2012; Moore, 2016; Haggard, 2017 for reviews). We thus considered that the body-generated action condition used in the present study, which is entirely similar to the active conditions previously reported in the literature, would be the baseline associated with a high sense of agency. This condition was then compared to the BMI-generated action condition in order to evaluate if using a BMI would lead to a similarly high sense of agency than in the body-generated action condition. In Study 2, we assumed that results of Study 1 were replicated for the BMI-generated action condition, thus resulting in a sense of agency on both Days 1 and 2. However, to draw more reliable conclusions on the sense of agency for MBI-generated actions, a passive control condition should be added.”. 

C16. p. 24, line 6: You talk about the “difference between the imagery phases and the rest phases”, but the difference in what?? You should always specify what variable you are talking about, even if you think it is obvious. Did you mean the difference in spectral power? If so then how measured? I.e. over what time interval / frequency range? And how did you compute your power estimates? Wavelets? Band-pass filter plus Hilbert transform? In general you have not been precise enough in describing your analyses.

→ R16. We agree that we should have been more precise in the way we report our analyses. We have now added the following information: “We conducted a repeated measures ANOVA with Rhythm (Mu, Beta) and Electrode (C3, Cz, C4) as within-subject factors on the difference in spectral power in the mu- and beta-bands between the imagery phases and the rest phases during the training session. Data were normalized by subtracting the global mean from each data point and by dividing the result  by the global SD. ” Also, in the section entitled ‘electroencephalography recordings and processing’, we have now added more information on how the power estimates were computed. 

C17. Figure 5: units!!

 → R17. We have added the units.

C18. When you talk about the “mu rhythm” and “beta rhythm” what do you mean? Do you mean power in the mu-band and power in the beta band? If so then you should say that. Just saying “mu rhythm” or “beta rhythm” is not specific enough. It’s unclear.

→ R18. We have now changed the wording in the entire manuscript, as suggested. We hope that the manuscript is now clearer. 

C19. Figure 6A: I am not sure it is appropriate or optimal to use Pearson’s correlation coefficient on a discrete variable (like perceived control).

→ R19. We agree that there is an ongoing debate about the use of Pearson correlations for discrete variables that have 6 points or more or if the use of nonparametric tests is more appropriate. However, R1 is right that the use of Spearman’s Rho could be more adapted and we have now changed the manuscript where necessary by reporting results with Spearman’s Rho instead of Pearson r. 

C20. No detail was given about the robotic hand itself, and very little detail about the BMI that was controlling it. How long was the latency between the BMI decision and the movement of the robotic hand? This information was given in the discussion, but should appear in the methods section as well. How long did it take for the robotic hand to depress the key? Was it an abrupt movement, or did the robotic hand move slowly until the key was pressed? Or did it move at a speed that was determined by the degree of BMI control? Did the key produce an audible click sound when pressed? These might seem like trivial details, but they may play a role in determining the sense of agency over the robotic hand, and the subsequent sense of agency over the keypress made by the robotic hand.

→ We have now added an additional section in the method section regarding the robotic hand. We have mentioned its features, the publications that detail how to build it and the time to press the keys. The time the robotic hand takes to press the key is indeed very important regarding the time estimation that participants have to make. We set-up this timing to 900 ms in order to have the robotic hand redressing the finger only after the beep occured on each trial and so to avoid temporal distortion for interval estimates. 

C21. Does the theoretical accuracy predict perceived control? This was significant in both experiments and simply suggests that better performance on the part of the classifier is associated with a stronger feeling of control over the BCI (which it should). A prior study that is directly relevant here is Schurger et al, Brain & Cognition (2016) which also looked at judgements of control over a BMI.

→ R21. There was indeed a significant correlation between classification performance and perceived control. We have now added the requested reference in the discussion, since it was indeed relevant: “This result is in line with a former study, which showed that participants are good at evaluating their ability to control a BMI even if they do not receive an online feedback on their performance (Schurger, Gale, Gozel, & Blanke, 2017). ”

C22. p. 26, lines 9-11: This sentence is simply confusing. I can’t make any sense of it. Please clarify. And what do you mean by the “estimated interval estimate”? Isn’t it just either the “estimated interval” or the “interval estimate”? And what is "the reported difficulty to make the hand performing the movement”? Does this refer to the robotic hand? Do you perhaps mean “the reported difficulty making the robotic hand perform the movement”?

→ R22. We have now changed these sentences as follows: “We then performed non-parametric correlation with Spearman's rho (⍴) between the reported interval estimates, the perceived control over the movement of the robotic hand and the reported difficulty to make the robotic hand perform the movement, on a trial-by-trial basis. ”

C23. p. 26, lines 24-25: "Results indicated that the perceived control was a better predictor of interval estimates than the reported difficulty.” Isn’t there guaranteed to be some collinearity in this regression since we know already that perceived control and reported difficulty are correlated? And how did you determine that perceived control was the better predictor? If they are collinear then this would be difficult to ascertain. In part it could just be that this was not clearly expressed in writing. The writing definitely could stand to be improved.

 → R23. We have now reported the VIF that rules out a collinearity effect (all VIF are always =< 2).  However, we do agree that both factors (i.e. perceived control and reported difficulty) could play antagonist roles on the implicit sense of agency: while higher perceived control could boost the implicit sense of agency, higher difficulty could reduce it. 

C24. p. 27, line 5: “p = -2.360”???

 → R24. We have now corrected this typo. 

C25. Again, what are “location scores”? This was unclear to me given that you used the word “location” to refer to which electrode you were looking at.

→ R25. We have now used the word ‘electrode’ when it referred to the electrode location. The word ‘location’ now only refers to the location scores from the questionnaire assessing the appropriation of the robotic hand. 

------

Reviewer #2: In two studies the authors lay out the importance of one’s sense of agency (SoA) over one’s actions and in particular actions performed by assistive technology via a brain-computer interface. Furthermore, they explore how the performance of the BCI as well as one’s perceived control or fatigue affect one’s SoA for the observed actions and how this may affect one’s sense of ownership over the assistive technology.

• What are the main claims of the paper and how significant are they for the discipline?

The main claims of the paper are that i) the performance of the BCI has immediate influence over one’s explicit SoA (judgment of agency - JoA), even though the BCI actions do not produce reafferences; ii) similarly, the performance of the BCI predicts one’s implicit SoA (or feeling of Agency – FoA), as determined using intentional binding (IB) as dependent variable; iii) the perceived effort or difficulty of completing the task with the BCI had a negative effect on the FoA, and that iv) an increased SoA also aids embodiment or ownership of the device.

Generally, these claims are very interesting for the discipline, as they attempt to disentangle different aspects of one’s SoA by combining implicit and explicit judgments and linking them to movement-related cortical desynchronization.

Overall, I am not convinced the authors have controlled for all aspects of their study to fully support these claims.

→ Thanks for judging our work and claims as interesting. We hope our revision further supports these claims. 

• Are the claims properly placed in the context of the previous literature? Have the authors treated the literature fairly?

C27. The introduction of the manuscript provides an overview of relevant literature on the sense of agency, intentional binding, and BCIs. While the authors aim to disentangle these different concepts in order to motivate their study design, I disagree with some of their key arguments here. I would appreciate some clarification on these points, as they are important for the study design, the interpretation of their results, as well as their implications.

Intentional binding

C26. I agree that it is important to separate the FoA from the JoA, as the authors do. However, I am not convinced that the “pre-reflective experience of” the FoA can be assessed using Intentional Binding. Indeed, I would think that e.g. Moore and Obhi (2012 Consciousness & Cognition) or Ebert and Wegner (2010, also C&C) would argue that IB is rather compatible with the JoA. (I have more comments on this for the analysis.) The FoA describes an implicit, on-going aspect of performing an action or making a movement. Neither of these points is true for IB, where the action has already been completed with the press of the button. I understand that this is a general discussion point, not specific to this study, but I think it should be considered.

→ R26.This is indeed a fair point and a lot of explicit measures have also been criticized in terms of how much they relate to agency, since classically authors use different questions that are adapted to their experimental paradigm (i.e. “Do you feel in control of the action”, “Do you feel that you are the authors of the action”, “Do you feel that who caused the outcomes”? etc.). However, since the point is somewhat derivative with respect to our main goals,  we chose not to further develop this discussion point. 

C27.  Comparing the JoA and IB also seems difficult, as the former starts with one’s movement intention and ends with the press of the button (“the hand moved according to my will”), whereas the latter only reflects the cause-and-effect of the button-press followed by a tone (“the delay that occurred between the robotic hand keypress and the tone”). It could therefore be argued that the JoA and the FoA measured here concern different processes and should not directly be compared (without further justification).

→ R27. We agree with the reviewer’s comment and that’s why we do not directly compare them. We nonetheless evaluate the correlations between these two experiences of the self, when one is producing an action. Several earlier  papers have evaluated (through correlations) how these two experiences of the self, despite measuring different aspects, relate and that is the procedure we have used here. 

C28. With respect to further literature on the IB paradigm, Suzuki et al.’s findings (Psychological Science 2019) suggesting that IB may simply reflect “multisensory causal binding” would be relevant to include, particularly as the current paper does neither includes baseline estimations for action- or outcome-binding nor a passive control condition. The work by Rohde and Ernst (e.g. Behavioural Science, 2016) is also relevant.

→ R28. According to the study of Suzuki et al 2019, any study wishing to draw conclusions specific to voluntary action, or similar theoretical constructs, needs to be based on the comparison of carefully matched conditions that should ideally differ in just one critical aspect. Here, our conditions only differed in the sensorimotor information coming from the keypress; they were similar in terms of causality and volition. We are aware of the current debate over the effect of causality on IB. While some authors argued that causality plays an important role (e.g. Buehner & Humphrey, 2009) in explaining variability in IB between 2 experimental conditions, other authors have shown that when causality is controlled as well as other features between two experimental conditions, and that only intentionality remains, a difference in IB still occur (Caspar, Cleeremans & Haggard, 2018).  - study 2). Very recent studies come to the conclusion that even if several factors, such as causality, play a role in the modulation of binding, intentionality also plays a key role (e.g., Lorimer, … , Buehner, 2020). However, since this debate is not the aim of the present paper, we do not discuss this literature in the present paper.

Questionnaire Data

C29. Table 1 seems to be missing in my documents, so I am not entirely sure, my comments on these data are completely accurate. However, a general point to consider for the questionnaire is – what is the control condition and is there a control question? (also see my comment in the data analysis section.) Recently, the questionnaires used in the rubber hand illusion and comparable studies have come under (even more) scrutiny. It would be great, if the authors could include a brief discussion of their questionnaire results with respect to the points raised by Peter Lush (2020) “Demand Characteristics Confound the Rubber Hand Illusion” Collabra: Psychology.

→ R29. We apologize for the omission. Table 1 was indeed missing and has now been added in the manuscript.  As far as we understand it, the critical difference between our design and that of Lush (2020) is that we never mentioned anything to our participants regarding the ‘robotic’ hand illusion or even that they would be given a questionnaire at the end of the experiment evaluating their appropriation of the robotic hand. We even avoided putting the classical blanket between the robotic hand and the participant to cover the participant’s arm, in order to avoid them thinking that we would evaluate the robotic hand as part of their own body. Our participants thus had a limited room for creating expectancies regarding the questionnaires and the robotic hand. However, we agree that controlling for expectancies is relevant in the case of the rubber hand illusion. We have now integrated this discussion in the general discussion section, as follows: “Several articles have argued that demand characteristics (Orne, 1962) and expectancies can predict scores on the classical rubber hand illusion (RHI) questionnaires (Lush, 2020). In the present study, we limited as much as possible the effects of both demand characteristics and expectancies on participants’ answer to the questionnaire related to the appropriation of the robotic hand: (1) Participants were not told anything about the rubber hand illusion or the fact that some people may experience the robotic hand as a part of their body during the experiment; (2) We did not tell them in advance that they would have to fill in a questionnaire regarding their appropriation of the robotic hand at the end of the experiment; (3) We avoided creating a proper classical ‘rubber hand illusion’: the robotic hand was placed at a congruent place on the table regarding the participant’s arm, but we did not use the blanket to cover their real arm, neither we used a paintbrush to induce an illusion. This procedure limited the possibility for participants to guess what we would assess since in the classical rubber hand illusion, placing the blanket on the participant’s arm make them realize that we investigate to what extent they appropriate this hand in their body schema. However, we cannot fully rule out the influence of demand characteristics and expectancies, as participants perceiving a better control over the hand, even without knowing their own classification performance, could have intuitively indicated higher scores on the questionnaires. Control studies could thus be performed to manipulate demand characteristics in order to evaluate its effect on the appropriation of a robotic hand in a brain-machine interface procedure (Lush, 2020).”

• Do the data and analyses fully support the claims? If not, what other evidence is required?

C30. IB: Was there any evidence of an intentional binding effect? It would be interesting to see the distribution of the reported intervals – is it trimodal? (Cf. figures in Suzuki et al.) Was the interval actually reduced? As there is apparently no passive or control condition and no difference between human or robot hand movements, it is not clear if there was any effect at all. Perhaps this is something that can be explored in the existing data set. If there is no effect of binding, what does that mean with respect to the findings of study 2?

→ R30. With respect to Study 2, we cannot infer that the sense of agency was ‘strong’ or ‘weak’ on either training days, which we did not argue for in our discussion. Rather, we can say that having a good control over the BMI boosts sense of agency, as measured through the method of interval estimates. We had three intervals: we have run those analyses with delays as an additional within-subject factor and have observed that IE were longer for the BMI condition for the 100ms-interval than for the real hand condition, not statistically different for the 400ms-interval between the two conditions, and shorter for the BMI condition for the 700ms-interval than for the real hand condition. In our opinion, nothing reliable could be drawn from those analyses regarding binding, and we therefore do not mention these data in the paper. But we indeed agree that this is a critical point in our paper, which is now further elaborated on in the discussion section. 

C31.  Control condition: Moore and Obhi 2012 argue that “Out of all the factors that have been linked to intentional binding, it appears that the presence of efferent information is most central to the manifestation of intentional binding as, when efferent information is not present, such as in passive movement conditions, or passive observation of others, intentional binding is reduced or absent.” Should you not have included such a condition in order to verify an effect of IB?

→ R31. We agree with R2 that we should have included such control conditions. The main reason for not including them was the exhausting nature of the task (learning to control a BMI) in addition to the duration of the task (about 2 hours). We thus considered that the body-generated action condition, which was similar to a host of previous studies using temporal binding was our baseline condition, corresponding to a high sense of agency. We have now added this point as a limitation in our study in the discussion section: “A limitation of the present study is that we did not use a control, passive condition similarly to previous studies on interval estimates. Such passive conditions involve a lack of volition in the realization of the movement, for instance by using a TMS over the motor cortex (Haggard et al., 2002) or by having a motoric setup forcing the participant’s finger to press the key (e.g. Moore, Wegner & Haggard, 2009). Neither Study 1 nor Study 2, includes such passive conditions, thus precluding our conclusions on the extent to which a BMI-generated action condition does involve a high sense of agency. The main reason for not introducing such control conditions stemmed from the tiring nature of the task asked from participants (i.e. learning to control a robotic hand through a BMI) and its duration (i.e. 2 hours). Adding additional experimental conditions would have strongly increased the fatigue  of participants during the task. The existing literature on active, voluntary conditions in which participants use their own index finger to press a key whenever they want is very extensive and consistently points towards lower interval estimates in active conditions than in passive conditions, involving a higher sense of agency (see Moore & Obhi, 2012; Moore, 2016; Haggard, 2017 for reviews). We thus considered that the body-generated action condition used in the present study, which is entirely similar to the active conditions previously reported in the literature, would be the baseline associated with a high sense of agency. This condition was then compared to the BMI-generated action condition in order to evaluate if using a BMI would lead to a similarly high sense of agency than in the body-generated action condition. In Study 2, we assumed that results of Study 1 were replicated for the BMI-generated action condition, thus resulting in a sense of agency on both Days 1 and 2. However, to draw more reliable conclusions on the sense of agency for MBI-generated actions, a passive control condition should be added.”

C32. Agency Question: Were there any false positives or catch trials, in which the robotic hand moved without the user’s intention? If the hand only ever moves when participants “intend” to move, then the hand can never actually “move by its own”. Furthermore, if the question(s) are only asked after a successful triggering of the hand movement, is the latter answer not ~unattainable?

→ R32. There were no catch trials per se, in the sense that we did not introduce trials in which the robotic hand moved by its own in the experimental design. There were however false positives, induced by the participants’ lack of control over the robotic hand. The explicit question about control was developed to find out when those false positives occurred: With this question, participants could tell us if each trial was a false positive (i.e. the robotic hand moved by ‘its own’) or an intended trial. We thus indeed rely on participant’s perception to know if the trial was a ‘false positive’ or not. 

C33.  Ownership Question: Is there a way to distinguish between ownership and agency in the current paradigm? Are the scores highly correlated? Can you exclude a counterargument such as demand characteristics being responsible for the ownership scores?

→ R33. Agency and ownership scores were indeed correlated, such as in several previous studies. We can’t really ensure that demand characteristics played a role in the ownership scores, but we tried,  however, to limit its impact more than in former studies (see also R29). This has been added in the discussion section.

C34.  Theoretical accuracy: The EEG (and EMG) methods are clearly described. The results with respect to mu- and beta-band suppression and electrode location are in-line with prior findings and it is good to see this BCI approach being applied to agency and ownership questions. I have a couple of questions which you can probably quite easily clarify. How does “theoretical accuracy” differ from (actual) classifier performance? Could you also briefly explain why you do not use cross-validation to measure the performance?

→ R34. In fact the theoretical accuracy score was the name that we gave to the classifier performance. We agree that this may have been confusing and, in line with Reviewer 1, we have now replaced the word ‘theoretical accuracy’ by ‘classification performance throughout the entire manuscript for the sake of clarity. More information has now been added in the manuscript regarding how the classification performance was calculated and regarding cross-validation: “We used the Filter Bank Common Spatial Pattern (FBCSP) variant, which first separates the signal into different frequency bands before applying CSP on each band. Results in each frequency band are then automatically weighted in such a way that the two conditions are optimally discriminated. Two frequency bands were selected: 8-12Hz and 13-30Hz. FBCSP and LDA were applied on data collected within a 1s a time window. The position of this time window inside the - 3s of rest or right hand motor imagery was automatically optimized for each participant by sliding the time window, training the classifier, comparing achieved accuracy by cross-validation, and keeping the classifier giving the best classification accuracy. ”  and “These training sessions were performed at least twice. The first training session was used to build an initial data set containing the EEG data of the subject and the known associated condition: “motor imagery of the right hand” or “being at rest”. This data set was then used to train the classifier. During the training of the classifier, a first theoretical indication of performance was obtained by cross-validation, which consisted in training the classifier using a subset of the data set, and of testing the obtained classifier on the remaining data. The second training session was used to properly evaluate the classification performance of the classifier. When the participant was asked to perform the training session a second time, the classifier was predicting in real time whether the participant was at rest or in right hand motor imagery condition. If classification performance was still low (around 50-60%), a new model was trained based on the additional training session, and combined with all previous training sessions. At most, six training sessions were performed. If the level of classification performance failed to reach chance level (i.e. 50%) after six sessions, the experiment was then terminated. ”

C35.  On the same point, can you calculate the classifier performance during the actual experimental block? Does this match the training performance and is this a better or poorer indicator of perceived control?

→ R35. The classifier performance was calculated based on the discrepancy/predictability between the model and participant’s performance during the training session. It means that in the training session, the classifier could know what was the expected fit because we alternated a cross appearing on the screen (corresponding to the rest phase) and an arrow appearing on that cross (corresponding to the motor imagery phase). This is what is represented on Figure 1A and Figure 1B. In the experimental block, participants could manipulate the robotic hand as they wanted, meaning that there was no expected match between a specific stimulus appearing on the screen and participants’ willingness to make the robotic hand move or not. It was thus not possible to calculate a score of classification performance during the experimental block, since we could not ask participants to report continuously when they were imagining the ‘rest phase’ to avoid making the robotic hand moving. 

C36. Theoretical accuracy and correlation to mu/beta oscillations: If I understand correctly, the classifier is trained to detect desynchronization in the mu/beta frequency bands. Performance is then quantified as theoretical accuracy. What does the correlation between theoretical accuracy and mu/beta oscillation actually tell us? Is this simply the confirmation that these are the trained criteria? (Apologies, if I simply missed the point here.)

→ R36. As a reminder, we have now changed the word ‘theoretical accuracy’ into ‘classification performance’ for more clarity. In fact, the feature extraction with CSP and LDA is not directly linked to mu and beta oscillations. However, since we filter the data based on the mu and the beta-bands, this means the classification performance score could be related to oscillations in the mu- and beta- bands. The correlation thus suggests that CSP+LDA correctly identifies the changes in mu and beta oscillations. According to the comment of Reviewer 1 (see R7 and R8), we have now added more information regarding the classification performance in the manuscript. We hope that this section is now clearer. 

- C37. Sample size: The justification of the sample size is not very clear. Would it not be better to either calculate it based on prior or expected effect sizes and add a percentage of participants in case of BCI illiteracy? Alternatively, you could use a Bayesian approach and determine a cut-off criterion.

→ R37. We agree with R2 but we  did not have  a strong prior regarding the expected effect sizes since no previous studies used a similar method and since we had no idea how many participants would be lower or higher than the chance level with our BMI. We did not have to exclude participants with a classification performance score lower than 50% and thus did not have to remove participants. Future studies using a similar approach will include a-priori computation of the sample size.

Dear Editor,

We would like to thank you for your time on this manuscript. We have revised the manuscript according to the comments received from the two reviewers. You will find attached a point-by-point response to the reviewers’ comments. We hope that the manuscript has been improved and will be suitable for publication in Plos One. 

Sincerely,

Emilie A. Caspar, Albert De Beir, Gil Lauwers, Axel Cleeremans and Bram Vanderborght

Reviewer #1: PONE-D-20-06849

This study looked at the sense of agency for actions performed using an EEG-based brain-machine interface (BMI). The intentional binding effect was used as the primary dependent measure, where subjects directly estimate the interval between the cause (button press) and effect (auditory click). In two experiments there were no significant differences in interval estimates between motor actions and BMI-mediated actions, suggesting no difference in agency. The authors did, however, find relationships between other variables in the experiment, such as the sense of control, experienced difficulty, ownership of the robotic hand, and agency (as indexed by the interval estimates).

The paper is not very clearly written, and it was very difficult to make sense of it in places, often because crucial information was left out or not made clear, or jargon was undefined. It really reads like a first or second draft of a paper, rather than a paper that has been through a few rounds of internal review before being deemed ready to submit to a journal for peer review. Many of my comments are thus focused on the writing and on making things more clear. The paper does not make any strong claims, and the conclusions that are drawn seem reasonable given the data. The methods involved in controlling the robotic hand and the robotic hand itself were not given in sufficient detail (for example we do not know anything about the features used to discriminate motor imagery from rest conditions, even though this might be relevant to some of the results that involved correlation with the “theoretical accuracy”). The statistics appear to be appropriate and properly reported, except that for correlations with ordinal data (as in figure 6A) it can be better to use a non-parametric statistic like Spearman’s rho.

 → We thank the referee for this frank but sympathetic assessment and apologize for the quality of the language. We have revised the manuscript extensively, improving both language quality and presentation. We have also added information about the decoding algorithms we used and now hope the manuscript feels more straightforward. We reply to the referee’s specific comments below: 

Specific comments (in no particular order):

C1. p. 4, line 3: what is a “skin-based input”?

→ R1. In the paper of Coyle and colleagues, the authors used a system that was detecting when participants were tapping on their arm to produce an outcome (i.e. a tone). We have now added this description in the relevant passage, as follows: For instance, Coyle, Moore, Kristensson, Fletcher and Blackwell (2012) showed that intentional binding was stronger in a condition in which participants used a skin-based input system that detected when participants were tapping on their arm to produce a resulting tone rather than in a condition in which they were tapping on a button to produce a similar tone

C2. p. 6, line 21”main age” should be “mean age”

→ R2. Corrected.

C3. p. 7, bottom and p. 12, line 22, and figure 3

So they used bi-polar electrodes to record EMG over the flexor digitorum superficialis, but it seems that they just monitored it on-line or on a single-trial basis (says that they used the EMG traces off-line to remove trials with muscle activity). But they did not, it seems, check for sub-threshold muscle activity during motor imagery by looking at the average EMG power over all trials. To do this you have to take the sqrt of the sum of squares in a sliding window for each trial, before you average the trials together. You have to do this if you want to rule out any muscle activity, including sub-threshold muscle activity.

 Also it says that EMG data were high-pass filtered at 10 Hz, but then nothing else. You can’t really see muscle activity without at least rectifying the signal, or (better) converting to power.

→ R3. We agree with this comment. An important aspect of the methods that we did not mention in the manuscript was that there was a baseline correction for the analysis of the EMG. This information has now been added in the relevant section. However, even using a different approach to analyse the EMG data, we cannot entirely rule out the presence of muscular activity. The flexor digitorum superficialis does not control for the movement of the thumb for instance - a movement that we could only check with visual control for overts mouvements. There were also no electrodes placed on the left arm for instance. We agree that these shortcomings constitute a limitation and have now added a discussion paragraph regarding this aspect, as follows (page 42): “Finally, another limitation is that we could not entirely rule out the presence of sub-thresholded muscular activity that could have triggered the BMI during the task. The flexor digitorum superficialis where the electrodes were placed does not allow to record activity associated with the movement of the thumb. In addition, no electrodes were placed on the left arm. Those movements were only controled visually by the experimenter during the task.”. 

C4. p. 9, line 4: They checked to see if, after six training sessions, accuracy of the classifier was below chance. But that’s not what you want to do. You want to check to see if accuracy of the classifier is significantly *different* from chance. Performance of 51% correct might still be chance! You should always say “not different from chance” rather than “below chance”.

→ R4. We were not interested in including only participants that were above chance level, as is indeed usually done in the classic literature regarding BCI. Our aim was to include even people at the chance level in order to have variability in our sample, that is a distribution ranging from people only able to control the BCI at chance level to people having a good control of the BCI. This variability was of interest to us in order to evaluate how different degrees of control over the BCI result in a modulation of the sense of agency as well as in embodiment of the robotic hand. We further clarified this choice on page 12 as follows: “We chose to also accept participants who were at the chance level in order to create variability in the control of the BMI to be able to study how this variability influences other factors.” Our only criterion was excluding participants if they were lower than 50%. We agreed that performance should be significantly lower than 50% to truly consider that our exclusion criteria were respected, since indeed a score of 48 or 49% could still be considered as the chance level. We have now modified the sentence on page 12 as follows: “At most, six training sessions were performed. If the level of classification performance failed to reach chance level (i.e. 50%) after six sessions, the experiment was then terminated. ”

C5. p. 10, line 5: "red cross” should be “red arrow”

 → R5. Thank you for noticing this typo, we have now corrected it.

C6. It would be helpful and useful if you computed AUC for control of the robotic hand, just to give an idea of what level of control your subjects managed to achieve.

 → R6. We thank the referee for this suggestion. Computing the area under the curve indeed makes it possible to better describe a binary classifier. Although often used in data science, we found very few examples of this metric in our field. As the goal of this paper was to use the BCI merely as a method in behavioural research  rather than to develop a better BCI, we decided not to include the suggested AUC analyses.

C7. What do you mean by the “theoretical” accuracy score? This was not clear enough. On what data set was this theoretical accuracy determined? You should have a test set and a training set, but you did not specify what you used as training data and what you used as test data. Please clarify.

→ R7. Thank you for this comment. We agree that the theoretical accuracy score could be confusing, such as pointed out by Reviewer 2 as well. We have now decided to use the terminology “classification performance” for more clarity. We have also added more information about the training session, as follows, on page 12: “These training sessions were performed at least twice. The first training session was used to build an initial data set containing the EEG data of the subject and the known associated condition: “motor imagery of the right hand” or “being at rest”. This data set was then used to train the classifier. During the training of the classifier, a first theoretical indication of performance was obtained by cross-validation, which consisted in training the classifier using a subset of the data set, and of testing the obtained classifier on the remaining data. The second training session was used to properly evaluate the classification performance of the classifier. When the participant was asked to perform the training session a second time, the classifier was predicting in real time whether the participant was at rest or in right hand motor imagery condition. If classification performance was still low (around 50-60%), a new model was trained based on the additional training session, and combined with all previous training sessions. At most, six training sessions were performed.”

C8. If theoretical accuracy refers to the predicted accuracy on test data given the accuracy of the classifier on then training data, then of course we expect mu and beta power to be predictive, because the classifier is almost certainly using mu and beta power to discriminate between motor imagery and rest. Using CSP and LDA is a very common recipe for EEG-based BCIs, and it tends very often to converge on mu and beta desynchronization. This brings up another point which is that you should specify in your methods section which were the features that the classifier used. You don’t have to go into all of the details of the classifier, but at a minimum you should state what the features were. Did the classifier operate on the raw time series, or (more likely) on a time-frequency representation of the data?

→ R8. Thank you for this comment. We have now added more information about the classification performance, as follows, on page 11: “The feature extraction that followed each training session was carried out with a Variant of Common Spatial Pattern (CSP), and the classifier used  Linear Discriminant Analysis (LDA, see Kothe & Makeig, 2013). We used the Filter Bank Common Spatial Pattern (FBCSP) variant, which first separates the signal into different frequency bands before applying CSP on each band. Results in each frequency band are then automatically weighted in such a way that the two conditions are optimally discriminated. Two frequency bands were selected: 8-12Hz and 13-30Hz. FBCSP and LDA were applied on data collected within a 1s a time window. The position of this time window inside the - 3s of rest or right hand motor imagery was automatically optimized for each participant by sliding the time window, training the classifier, comparing achieved accuracy by cross-validation, and keeping the classifier giving the best classification accuracy. ”

C9. p. 14, label says “FIGURE 3” but I think this is figure 2.

 → R9. We have now corrected this typo. 

C10. “Location” = electrode location? This was not always clear. Maybe just call it “electrode”. What does the word “location” refer to in the section on "Perceived control and appropriation of the robotic hand”? Are you using the word “location” in two different ways. I know that one is the electrode location, but then it seems you are using it for something else as well. This was confusing.

→ R10. We agree with this suggestion and we have now used ‘electrode’ instead of ‘location’ when necessary, since location could also refer to the ‘location’ score from the rubber hand illusion questionnaire measuring the degree of appropriation of the robotic hand.  ‘Location’ when referring to the appropriation of the robotic hand was left as before since this is in accordance with the wording used in other papers on the RHI. 

C11. For spectral power calculations on the EEG data, did you normalize, i.e. divide by the standard deviation of the baseline? I realize that you did your power spectral analyses without baseline correction because the time leading up to the action was too heterogenous. But couldn’t you use a time window before even the start of the trial as your baseline, or perhaps during the two-second waiting period? Because power is typically higher at lower frequencies because of 1/f scaling, so it is more meaningful to express your power estimates as a Z score before comparing. It is not really valid to compare power at in different frequency bands. I.e. you comparing power in the mu band to power in the beta band in kind of meaningless. Mu band will always win, just because there is always more power at lower frequencies.

→ R11. Thank you for this insightful comment.Those two options are unfortunately not reliable. Regarding the 2s waiting period, even though we asked participants to relax, we did not specify exactly when the waiting period finished (e.g. with a sentence appearing on the screen for instance, explicitly instructing them to relax). In the BMI-generated action condition, we cannot be entirely sure of when participants start to use motor imagery. Some participants could for instance start trying to use motor imagery to make the robotic hand move after 1s or 1.5 s. Since we don’t have this information, we doubt we can use it  as a baseline. Regarding the start of the trial option, participants finished the former trial by validating their answer by pressing the ENTER keypress. They did not have to wait for a fixed period of time before starting the next trial by pressing the ENTER key again. Most of the participants started the next trial right away. Therefore, this cannot constitute a proper baseline period either. But we agree that for future studies we should integrate a fixed and more clear time period to obtain a proper baseline correction. 

We have followed the suggestion to normalize the data by transforming  the data associated with each frequency band into z scores. We carried all analyses again on the normalized data, and reported these analyses in our revision. 

C12. Figure 2 A and C: what are the units???

 → R12. This has now been added. 

C13. p. 18, line 16: What are the two variables precisely? One is the difference in interval estimates between the two tasks, but what is the other one? You just refer to the “difference between the real hand key press and the robotic hand key press”, but the difference in what? Spectral power? And what is each data point? A subject (I presume)? You just need to be more clear and specific.

→ R13. We apologize for the confusion. We have now modified the sentence as follows: “We additionally investigated, with Pearson correlations, whether or not the difference in power in the mu- and the beta-bands between the real hand keypress and the robotic hand keypress on C3, Cz and C4 for each participant could predict the difference in interval estimates between the same two tasks.”

C14. p. 20, line 10: This is confusing. You write “we observed a greater desynchronization in the beta band than in the mu band.” Do you mean for real hand movements? I thought that you had observed greater desynchronization in the mu band compared to the beta band (see p. 19 line 15).

→ R14. Those results have now changed because we have normalized the data within each frequency band. The new results and the adapted discussion are now reported in the revised version of the manuscript. Overall, power in the mu and the beta-band now do not significantly differ between each other. 

C15. You state that there was no difference in the interval estimates for real button press versus motor imagery, but did you verify that you even obtained a canonical intentional binding effect in the first place? I did not see this anywhere. Maybe there was an IB effect for both real-hand and robotic-hand keypresses, in which case you might not see any difference. Or maybe neither of them had the effect. Don’t you need a passive viewing or involuntary-motor task to control for that?

→ R15. We agree with this comment, which is similar to comment C31 from reviewer 2. In the classic IB literature, a comparison between two experimental conditions is necessary, generally an active and a passive condition, in order to detect changes in the sense of agency. In our study, we did not introduce a passive condition for the following reason: the experiment was already quite long and exhausting for participants. Adding additional experimental conditions would have led to even higher cognitive fatigue, which could have been detrimental for the interval estimate task. Our main aim was to investigate if agency would differ between the body-generated action condition and the BMI-generated action condition, and hence we did not introduce a control, passive condition which would have been critical to know if agency was ‘high’ in both conditions or ‘low’ in both conditions. Given the extensive existing literature on body-generated action and interval estimates, we considered that, like in the existing literature, this condition reflected a sort of ‘high agency condition’, since movements were voluntarily produced with participants’ own keypress. Based on this, we thus extrapolated that agency was also high in the BMI-generated action condition. We fully agree that this extrapolation may not be sufficient, and we have now added this comment as a limitation in the general discussion of the present paper, as follows: “A limitation of the present study is that we did not use a control, passive condition similarly to previous studies on interval estimates. Such passive conditions involve a lack of volition in the realization of the movement, for instance by using a TMS over the motor cortex (Haggard et al., 2002) or by having a motoric setup forcing the participant’s finger to press the key (e.g. Moore, Wegner & Haggard, 2009). Neither Study 1 nor Study 2, includes such passive conditions, thus precluding our conclusions on the extent to which a BMI-generated action condition does involve a high sense of agency. The main reason for not introducing such control conditions stemmed from the tiring nature of the task asked from participants (i.e. learning to control a robotic hand through a BMI) and its duration (i.e. 2 hours). Adding additional experimental conditions would have strongly increased the fatigue  of participants during the task. The existing literature on active, voluntary conditions in which participants use their own index finger to press a key whenever they want is very extensive and consistently points towards lower interval estimates in active conditions than in passive conditions, involving a higher sense of agency (see Moore & Obhi, 2012; Moore, 2016; Haggard, 2017 for reviews). We thus considered that the body-generated action condition used in the present study, which is entirely similar to the active conditions previously reported in the literature, would be the baseline associated with a high sense of agency. This condition was then compared to the BMI-generated action condition in order to evaluate if using a BMI would lead to a similarly high sense of agency than in the body-generated action condition. In Study 2, we assumed that results of Study 1 were replicated for the BMI-generated action condition, thus resulting in a sense of agency on both Days 1 and 2. However, to draw more reliable conclusions on the sense of agency for MBI-generated actions, a passive control condition should be added.”. 

C16. p. 24, line 6: You talk about the “difference between the imagery phases and the rest phases”, but the difference in what?? You should always specify what variable you are talking about, even if you think it is obvious. Did you mean the difference in spectral power? If so then how measured? I.e. over what time interval / frequency range? And how did you compute your power estimates? Wavelets? Band-pass filter plus Hilbert transform? In general you have not been precise enough in describing your analyses.

→ R16. We agree that we should have been more precise in the way we report our analyses. We have now added the following information: “We conducted a repeated measures ANOVA with Rhythm (Mu, Beta) and Electrode (C3, Cz, C4) as within-subject factors on the difference in spectral power in the mu- and beta-bands between the imagery phases and the rest phases during the training session. Data were normalized by subtracting the global mean from each data point and by dividing the result  by the global SD. ” Also, in the section entitled ‘electroencephalography recordings and processing’, we have now added more information on how the power estimates were computed. 

C17. Figure 5: units!!

 → R17. We have added the units.

C18. When you talk about the “mu rhythm” and “beta rhythm” what do you mean? Do you mean power in the mu-band and power in the beta band? If so then you should say that. Just saying “mu rhythm” or “beta rhythm” is not specific enough. It’s unclear.

→ R18. We have now changed the wording in the entire manuscript, as suggested. We hope that the manuscript is now clearer. 

C19. Figure 6A: I am not sure it is appropriate or optimal to use Pearson’s correlation coefficient on a discrete variable (like perceived control).

→ R19. We agree that there is an ongoing debate about the use of Pearson correlations for discrete variables that have 6 points or more or if the use of nonparametric tests is more appropriate. However, R1 is right that the use of Spearman’s Rho could be more adapted and we have now changed the manuscript where necessary by reporting results with Spearman’s Rho instead of Pearson r. 

C20. No detail was given about the robotic hand itself, and very little detail about the BMI that was controlling it. How long was the latency between the BMI decision and the movement of the robotic hand? This information was given in the discussion, but should appear in the methods section as well. How long did it take for the robotic hand to depress the key? Was it an abrupt movement, or did the robotic hand move slowly until the key was pressed? Or did it move at a speed that was determined by the degree of BMI control? Did the key produce an audible click sound when pressed? These might seem like trivial details, but they may play a role in determining the sense of agency over the robotic hand, and the subsequent sense of agency over the keypress made by the robotic hand.

→ We have now added an additional section in the method section regarding the robotic hand. We have mentioned its features, the publications that detail how to build it and the time to press the keys. The time the robotic hand takes to press the key is indeed very important regarding the time estimation that participants have to make. We set-up this timing to 900 ms in order to have the robotic hand redressing the finger only after the beep occured on each trial and so to avoid temporal distortion for interval estimates. 

C21. Does the theoretical accuracy predict perceived control? This was significant in both experiments and simply suggests that better performance on the part of the classifier is associated with a stronger feeling of control over the BCI (which it should). A prior study that is directly relevant here is Schurger et al, Brain & Cognition (2016) which also looked at judgements of control over a BMI.

→ R21. There was indeed a significant correlation between classification performance and perceived control. We have now added the requested reference in the discussion, since it was indeed relevant: “This result is in line with a former study, which showed that participants are good at evaluating their ability to control a BMI even if they do not receive an online feedback on their performance (Schurger, Gale, Gozel, & Blanke, 2017). ”

C22. p. 26, lines 9-11: This sentence is simply confusing. I can’t make any sense of it. Please clarify. And what do you mean by the “estimated interval estimate”? Isn’t it just either the “estimated interval” or the “interval estimate”? And what is "the reported difficulty to make the hand performing the movement”? Does this refer to the robotic hand? Do you perhaps mean “the reported difficulty making the robotic hand perform the movement”?

→ R22. We have now changed these sentences as follows: “We then performed non-parametric correlation with Spearman's rho (⍴) between the reported interval estimates, the perceived control over the movement of the robotic hand and the reported difficulty to make the robotic hand perform the movement, on a trial-by-trial basis. ”

C23. p. 26, lines 24-25: "Results indicated that the perceived control was a better predictor of interval estimates than the reported difficulty.” Isn’t there guaranteed to be some collinearity in this regression since we know already that perceived control and reported difficulty are correlated? And how did you determine that perceived control was the better predictor? If they are collinear then this would be difficult to ascertain. In part it could just be that this was not clearly expressed in writing. The writing definitely could stand to be improved.

 → R23. We have now reported the VIF that rules out a collinearity effect (all VIF are always =< 2).  However, we do agree that both factors (i.e. perceived control and reported difficulty) could play antagonist roles on the implicit sense of agency: while higher perceived control could boost the implicit sense of agency, higher difficulty could reduce it. 

C24. p. 27, line 5: “p = -2.360”???

 → R24. We have now corrected this typo. 

C25. Again, what are “location scores”? This was unclear to me given that you used the word “location” to refer to which electrode you were looking at.

→ R25. We have now used the word ‘electrode’ when it referred to the electrode location. The word ‘location’ now only refers to the location scores from the questionnaire assessing the appropriation of the robotic hand. 

------

Reviewer #2: In two studies the authors lay out the importance of one’s sense of agency (SoA) over one’s actions and in particular actions performed by assistive technology via a brain-computer interface. Furthermore, they explore how the performance of the BCI as well as one’s perceived control or fatigue affect one’s SoA for the observed actions and how this may affect one’s sense of ownership over the assistive technology.

• What are the main claims of the paper and how significant are they for the discipline?

The main claims of the paper are that i) the performance of the BCI has immediate influence over one’s explicit SoA (judgment of agency - JoA), even though the BCI actions do not produce reafferences; ii) similarly, the performance of the BCI predicts one’s implicit SoA (or feeling of Agency – FoA), as determined using intentional binding (IB) as dependent variable; iii) the perceived effort or difficulty of completing the task with the BCI had a negative effect on the FoA, and that iv) an increased SoA also aids embodiment or ownership of the device.

Generally, these claims are very interesting for the discipline, as they attempt to disentangle different aspects of one’s SoA by combining implicit and explicit judgments and linking them to movement-related cortical desynchronization.

Overall, I am not convinced the authors have controlled for all aspects of their study to fully support these claims.

→ Thanks for judging our work and claims as interesting. We hope our revision further supports these claims. 

• Are the claims properly placed in the context of the previous literature? Have the authors treated the literature fairly?

C27. The introduction of the manuscript provides an overview of relevant literature on the sense of agency, intentional binding, and BCIs. While the authors aim to disentangle these different concepts in order to motivate their study design, I disagree with some of their key arguments here. I would appreciate some clarification on these points, as they are important for the study design, the interpretation of their results, as well as their implications.

Intentional binding

C26. I agree that it is important to separate the FoA from the JoA, as the authors do. However, I am not convinced that the “pre-reflective experience of” the FoA can be assessed using Intentional Binding. Indeed, I would think that e.g. Moore and Obhi (2012 Consciousness & Cognition) or Ebert and Wegner (2010, also C&C) would argue that IB is rather compatible with the JoA. (I have more comments on this for the analysis.) The FoA describes an implicit, on-going aspect of performing an action or making a movement. Neither of these points is true for IB, where the action has already been completed with the press of the button. I understand that this is a general discussion point, not specific to this study, but I think it should be considered.

→ R26.This is indeed a fair point and a lot of explicit measures have also been criticized in terms of how much they relate to agency, since classically authors use different questions that are adapted to their experimental paradigm (i.e. “Do you feel in control of the action”, “Do you feel that you are the authors of the action”, “Do you feel that who caused the outcomes”? etc.). However, since the point is somewhat derivative with respect to our main goals,  we chose not to further develop this discussion point. 

C27.  Comparing the JoA and IB also seems difficult, as the former starts with one’s movement intention and ends with the press of the button (“the hand moved according to my will”), whereas the latter only reflects the cause-and-effect of the button-press followed by a tone (“the delay that occurred between the robotic hand keypress and the tone”). It could therefore be argued that the JoA and the FoA measured here concern different processes and should not directly be compared (without further justification).

→ R27. We agree with the reviewer’s comment and that’s why we do not directly compare them. We nonetheless evaluate the correlations between these two experiences of the self, when one is producing an action. Several earlier  papers have evaluated (through correlations) how these two experiences of the self, despite measuring different aspects, relate and that is the procedure we have used here. 

C28. With respect to further literature on the IB paradigm, Suzuki et al.’s findings (Psychological Science 2019) suggesting that IB may simply reflect “multisensory causal binding” would be relevant to include, particularly as the current paper does neither includes baseline estimations for action- or outcome-binding nor a passive control condition. The work by Rohde and Ernst (e.g. Behavioural Science, 2016) is also relevant.

→ R28. According to the study of Suzuki et al 2019, any study wishing to draw conclusions specific to voluntary action, or similar theoretical constructs, needs to be based on the comparison of carefully matched conditions that should ideally differ in just one critical aspect. Here, our conditions only differed in the sensorimotor information coming from the keypress; they were similar in terms of causality and volition. We are aware of the current debate over the effect of causality on IB. While some authors argued that causality plays an important role (e.g. Buehner & Humphrey, 2009) in explaining variability in IB between 2 experimental conditions, other authors have shown that when causality is controlled as well as other features between two experimental conditions, and that only intentionality remains, a difference in IB still occur (Caspar, Cleeremans & Haggard, 2018).  - study 2). Very recent studies come to the conclusion that even if several factors, such as causality, play a role in the modulation of binding, intentionality also plays a key role (e.g., Lorimer, … , Buehner, 2020). However, since this debate is not the aim of the present paper, we do not discuss this literature in the present paper.

Questionnaire Data

C29. Table 1 seems to be missing in my documents, so I am not entirely sure, my comments on these data are completely accurate. However, a general point to consider for the questionnaire is – what is the control condition and is there a control question? (also see my comment in the data analysis section.) Recently, the questionnaires used in the rubber hand illusion and comparable studies have come under (even more) scrutiny. It would be great, if the authors could include a brief discussion of their questionnaire results with respect to the points raised by Peter Lush (2020) “Demand Characteristics Confound the Rubber Hand Illusion” Collabra: Psychology.

→ R29. We apologize for the omission. Table 1 was indeed missing and has now been added in the manuscript.  As far as we understand it, the critical difference between our design and that of Lush (2020) is that we never mentioned anything to our participants regarding the ‘robotic’ hand illusion or even that they would be given a questionnaire at the end of the experiment evaluating their appropriation of the robotic hand. We even avoided putting the classical blanket between the robotic hand and the participant to cover the participant’s arm, in order to avoid them thinking that we would evaluate the robotic hand as part of their own body. Our participants thus had a limited room for creating expectancies regarding the questionnaires and the robotic hand. However, we agree that controlling for expectancies is relevant in the case of the rubber hand illusion. We have now integrated this discussion in the general discussion section, as follows: “Several articles have argued that demand characteristics (Orne, 1962) and expectancies can predict scores on the classical rubber hand illusion (RHI) questionnaires (Lush, 2020). In the present study, we limited as much as possible the effects of both demand characteristics and expectancies on participants’ answer to the questionnaire related to the appropriation of the robotic hand: (1) Participants were not told anything about the rubber hand illusion or the fact that some people may experience the robotic hand as a part of their body during the experiment; (2) We did not tell them in advance that they would have to fill in a questionnaire regarding their appropriation of the robotic hand at the end of the experiment; (3) We avoided creating a proper classical ‘rubber hand illusion’: the robotic hand was placed at a congruent place on the table regarding the participant’s arm, but we did not use the blanket to cover their real arm, neither we used a paintbrush to induce an illusion. This procedure limited the possibility for participants to guess what we would assess since in the classical rubber hand illusion, placing the blanket on the participant’s arm make them realize that we investigate to what extent they appropriate this hand in their body schema. However, we cannot fully rule out the influence of demand characteristics and expectancies, as participants perceiving a better control over the hand, even without knowing their own classification performance, could have intuitively indicated higher scores on the questionnaires. Control studies could thus be performed to manipulate demand characteristics in order to evaluate its effect on the appropriation of a robotic hand in a brain-machine interface procedure (Lush, 2020).”

• Do the data and analyses fully support the claims? If not, what other evidence is required?

C30. IB: Was there any evidence of an intentional binding effect? It would be interesting to see the distribution of the reported intervals – is it trimodal? (Cf. figures in Suzuki et al.) Was the interval actually reduced? As there is apparently no passive or control condition and no difference between human or robot hand movements, it is not clear if there was any effect at all. Perhaps this is something that can be explored in the existing data set. If there is no effect of binding, what does that mean with respect to the findings of study 2?

→ R30. With respect to Study 2, we cannot infer that the sense of agency was ‘strong’ or ‘weak’ on either training days, which we did not argue for in our discussion. Rather, we can say that having a good control over the BMI boosts sense of agency, as measured through the method of interval estimates. We had three intervals: we have run those analyses with delays as an additional within-subject factor and have observed that IE were longer for the BMI condition for the 100ms-interval than for the real hand condition, not statistically different for the 400ms-interval between the two conditions, and shorter for the BMI condition for the 700ms-interval than for the real hand condition. In our opinion, nothing reliable could be drawn from those analyses regarding binding, and we therefore do not mention these data in the paper. But we indeed agree that this is a critical point in our paper, which is now further elaborated on in the discussion section. 

C31.  Control condition: Moore and Obhi 2012 argue that “Out of all the factors that have been linked to intentional binding, it appears that the presence of efferent information is most central to the manifestation of intentional binding as, when efferent information is not present, such as in passive movement conditions, or passive observation of others, intentional binding is reduced or absent.” Should you not have included such a condition in order to verify an effect of IB?

→ R31. We agree with R2 that we should have included such control conditions. The main reason for not including them was the exhausting nature of the task (learning to control a BMI) in addition to the duration of the task (about 2 hours). We thus considered that the body-generated action condition, which was similar to a host of previous studies using temporal binding was our baseline condition, corresponding to a high sense of agency. We have now added this point as a limitation in our study in the discussion section: “A limitation of the present study is that we did not use a control, passive condition similarly to previous studies on interval estimates. Such passive conditions involve a lack of volition in the realization of the movement, for instance by using a TMS over the motor cortex (Haggard et al., 2002) or by having a motoric setup forcing the participant’s finger to press the key (e.g. Moore, Wegner & Haggard, 2009). Neither Study 1 nor Study 2, includes such passive conditions, thus precluding our conclusions on the extent to which a BMI-generated action condition does involve a high sense of agency. The main reason for not introducing such control conditions stemmed from the tiring nature of the task asked from participants (i.e. learning to control a robotic hand through a BMI) and its duration (i.e. 2 hours). Adding additional experimental conditions would have strongly increased the fatigue  of participants during the task. The existing literature on active, voluntary conditions in which participants use their own index finger to press a key whenever they want is very extensive and consistently points towards lower interval estimates in active conditions than in passive conditions, involving a higher sense of agency (see Moore & Obhi, 2012; Moore, 2016; Haggard, 2017 for reviews). We thus considered that the body-generated action condition used in the present study, which is entirely similar to the active conditions previously reported in the literature, would be the baseline associated with a high sense of agency. This condition was then compared to the BMI-generated action condition in order to evaluate if using a BMI would lead to a similarly high sense of agency than in the body-generated action condition. In Study 2, we assumed that results of Study 1 were replicated for the BMI-generated action condition, thus resulting in a sense of agency on both Days 1 and 2. However, to draw more reliable conclusions on the sense of agency for MBI-generated actions, a passive control condition should be added.”

C32. Agency Question: Were there any false positives or catch trials, in which the robotic hand moved without the user’s intention? If the hand only ever moves when participants “intend” to move, then the hand can never actually “move by its own”. Furthermore, if the question(s) are only asked after a successful triggering of the hand movement, is the latter answer not ~unattainable?

→ R32. There were no catch trials per se, in the sense that we did not introduce trials in which the robotic hand moved by its own in the experimental design. There were however false positives, induced by the participants’ lack of control over the robotic hand. The explicit question about control was developed to find out when those false positives occurred: With this question, participants could tell us if each trial was a false positive (i.e. the robotic hand moved by ‘its own’) or an intended trial. We thus indeed rely on participant’s perception to know if the trial was a ‘false positive’ or not. 

C33.  Ownership Question: Is there a way to distinguish between ownership and agency in the current paradigm? Are the scores highly correlated? Can you exclude a counterargument such as demand characteristics being responsible for the ownership scores?

→ R33. Agency and ownership scores were indeed correlated, such as in several previous studies. We can’t really ensure that demand characteristics played a role in the ownership scores, but we tried,  however, to limit its impact more than in former studies (see also R29). This has been added in the discussion section.

C34.  Theoretical accuracy: The EEG (and EMG) methods are clearly described. The results with respect to mu- and beta-band suppression and electrode location are in-line with prior findings and it is good to see this BCI approach being applied to agency and ownership questions. I have a couple of questions which you can probably quite easily clarify. How does “theoretical accuracy” differ from (actual) classifier performance? Could you also briefly explain why you do not use cross-validation to measure the performance?

→ R34. In fact the theoretical accuracy score was the name that we gave to the classifier performance. We agree that this may have been confusing and, in line with Reviewer 1, we have now replaced the word ‘theoretical accuracy’ by ‘classification performance throughout the entire manuscript for the sake of clarity. More information has now been added in the manuscript regarding how the classification performance was calculated and regarding cross-validation: “We used the Filter Bank Common Spatial Pattern (FBCSP) variant, which first separates the signal into different frequency bands before applying CSP on each band. Results in each frequency band are then automatically weighted in such a way that the two conditions are optimally discriminated. Two frequency bands were selected: 8-12Hz and 13-30Hz. FBCSP and LDA were applied on data collected within a 1s a time window. The position of this time window inside the - 3s of rest or right hand motor imagery was automatically optimized for each participant by sliding the time window, training the classifier, comparing achieved accuracy by cross-validation, and keeping the classifier giving the best classification accuracy. ”  and “These training sessions were performed at least twice. The first training session was used to build an initial data set containing the EEG data of the subject and the known associated condition: “motor imagery of the right hand” or “being at rest”. This data set was then used to train the classifier. During the training of the classifier, a first theoretical indication of performance was obtained by cross-validation, which consisted in training the classifier using a subset of the data set, and of testing the obtained classifier on the remaining data. The second training session was used to properly evaluate the classification performance of the classifier. When the participant was asked to perform the training session a second time, the classifier was predicting in real time whether the participant was at rest or in right hand motor imagery condition. If classification performance was still low (around 50-60%), a new model was trained based on the additional training session, and combined with all previous training sessions. At most, six training sessions were performed. If the level of classification performance failed to reach chance level (i.e. 50%) after six sessions, the experiment was then terminated. ”

C35.  On the same point, can you calculate the classifier performance during the actual experimental block? Does this match the training performance and is this a better or poorer indicator of perceived control?

→ R35. The classifier performance was calculated based on the discrepancy/predictability between the model and participant’s performance during the training session. It means that in the training session, the classifier could know what was the expected fit because we alternated a cross appearing on the screen (corresponding to the rest phase) and an arrow appearing on that cross (corresponding to the motor imagery phase). This is what is represented on Figure 1A and Figure 1B. In the experimental block, participants could manipulate the robotic hand as they wanted, meaning that there was no expected match between a specific stimulus appearing on the screen and participants’ willingness to make the robotic hand move or not. It was thus not possible to calculate a score of classification performance during the experimental block, since we could not ask participants to report continuously when they were imagining the ‘rest phase’ to avoid making the robotic hand moving. 

C36. Theoretical accuracy and correlation to mu/beta oscillations: If I understand correctly, the classifier is trained to detect desynchronization in the mu/beta frequency bands. Performance is then quantified as theoretical accuracy. What does the correlation between theoretical accuracy and mu/beta oscillation actually tell us? Is this simply the confirmation that these are the trained criteria? (Apologies, if I simply missed the point here.)

→ R36. As a reminder, we have now changed the word ‘theoretical accuracy’ into ‘classification performance’ for more clarity. In fact, the feature extraction with CSP and LDA is not directly linked to mu and beta oscillations. However, since we filter the data based on the mu and the beta-bands, this means the classification performance score could be related to oscillations in the mu- and beta- bands. The correlation thus suggests that CSP+LDA correctly identifies the changes in mu and beta oscillations. According to the comment of Reviewer 1 (see R7 and R8), we have now added more information regarding the classification performance in the manuscript. We hope that this section is now clearer. 

- C37. Sample size: The justification of the sample size is not very clear. Would it not be better to either calculate it based on prior or expected effect sizes and add a percentage of participants in case of BCI illiteracy? Alternatively, you could use a Bayesian approach and determine a cut-off criterion.

→ R37. We agree with R2 but we  did not have  a strong prior regarding the expected effect sizes since no previous studies used a similar method and since we had no idea how many participants would be lower or higher than the chance level with our BMI. We did not have to exclude participants with a classification performance score lower than 50% and thus did not have to remove participants. Future studies using a similar approach will include a-priori computation of the sample size.

---

## [Decision Letter · Decision Letter 1]

30 Oct 2020

PONE-D-20-06849R1

How using brain-machine interfaces influences the human sense of agency

PLOS ONE

Dear Dr. Caspar,

Thank you for submitting your manuscript to PLOS ONE. The reviewers only require some additional small changes to the manuscript. Therefore, we invite you to submit a revised version of the manuscript that addresses the points raised during the review process.

We look forward to receiving your revised manuscript.

Kind regards,

Jane Elizabeth Aspell, PhD

Academic Editor

PLOS ONE

Reviewers' comments:

Reviewer's Responses to Questions

**Comments to the Author**

1. If the authors have adequately addressed your comments raised in a previous round of review and you feel that this manuscript is now acceptable for publication, you may indicate that here to bypass the “Comments to the Author” section, enter your conflict of interest statement in the “Confidential to Editor” section, and submit your "Accept" recommendation.

Reviewer #1: All comments have been addressed

Reviewer #2: All comments have been addressed

2. Is the manuscript technically sound, and do the data support the conclusions?

Reviewer #1: Yes

Reviewer #2: Yes

3. Has the statistical analysis been performed appropriately and rigorously? 

Reviewer #1: Yes

Reviewer #2: Yes

4. Have the authors made all data underlying the findings in their manuscript fully available?

Reviewer #1: Yes

Reviewer #2: Yes

5. Is the manuscript presented in an intelligible fashion and written in standard English?

Reviewer #1: Yes

Reviewer #2: Yes

6. Review Comments to the Author

Reviewer #1: On page 48 of the pdf file, line 21, it reads that "each training session lasted 2.5 minutes..." This seems a bit short for 40 trials. Are you sure you didn't mean 25 minutes?

Also you noted that for the time-frequency data you normalized the power estimates by subtracting the global mean and dividing the result by the global SD. What did you mean by "global" here? You should subtract the mean power within each frequency band and also divide by the standard deviation within each frequency band. This should be clarified.

Reviewer #2: Dear authors, thank you for your detailed feedback.

The current version of the manuscript is much easier to follow and a number of items have been clarified. Overall, I appreciate the BCI-approach to questions of embodiment and the effort that has gone into these two studies. Unfortunately, the lack of a control condition and of control items for the questionnaire limit the interpretation of the findings. However, I agree with reviewer 1 that the claims you are making are not very strong, and given the revised manuscript, are supported by your findings.

In my opinion, it would be worth briefly discussing the actual interval estimates. I think the fact that IEs for shorter delays are smaller for "biological" movement than for robotic movements, whereas the reverse is true for IEs after longer delays, is quite interesting.

Along these lines - have you checked wether the reported SoA differs depending on the delay? Even though the agency question is phrased clearly, it would be worth checking if the responses are affected by the subsequent interval.

7. PLOS authors have the option to publish the peer review history of their article (what does this mean?). If published, this will include your full peer review and any attached files.

Reviewer #1: **Yes: **Aaron Schurger

Reviewer #2: No

---

## [Author Response · Author response to Decision Letter 1]

7 Nov 2020

Reviewer #1: On page 48 of the pdf file, line 21, it reads that "each training session lasted 2.5 minutes..." This seems a bit short for 40 trials. Are you sure you didn't mean 25 minutes?

 For each trial, there was 3s for motor imagery, 3s for “thinking about nothing” and 2s for a break (see figure1A). We thus have 8s per trial * 20 trials =160 s. = 2.5 min. But we agree it’s confusing in the way this section is currently written and have made some modifications, as follows: 

Each training session lasted 2.5 minutes and was composed of 20 trials. Each trial was composed of 3s for the rest phase (i.e. thinking about nothing), 3s for the motor imagery phase and a 2s-break (see Figure 1A).

Also you noted that for the time-frequency data you normalized the power estimates by subtracting the global mean and dividing the result by the global SD. What did you mean by "global" here? You should subtract the mean power within each frequency band and also divide by the standard deviation within each frequency band. This should be clarified.

 This is indeed what we did, we are sorry that it was not clear enough with the word “global”. We have now modified this sentence. 

Data were normalized by subtracting the global mean within each frequency band from each data point and by dividing the result by the global SD within the same frequency band.

Reviewer #2: Dear authors, thank you for your detailed feedback.

The current version of the manuscript is much easier to follow and a number of items have been clarified. Overall, I appreciate the BCI-approach to questions of embodiment and the effort that has gone into these two studies. Unfortunately, the lack of a control condition and of control items for the questionnaire limit the interpretation of the findings. However, I agree with reviewer 1 that the claims you are making are not very strong, and given the revised manuscript, are supported by your findings.

In my opinion, it would be worth briefly discussing the actual interval estimates. I think the fact that IEs for shorter delays are smaller for "biological" movement than for robotic movements, whereas the reverse is true for IEs after longer delays, is quite interesting.

Along these lines - have you checked whether the reported SoA differs depending on the delay? Even though the agency question is phrased clearly, it would be worth checking if the responses are affected by the subsequent interval.

 We have now added the statistical result regarding interval estimates for each delay in the result section and briefly discuss this finding in the discussion section.

We also explore if the reported interval estimates would vary regarding actual action-tone intervals, depending on the experimental condition. We thus run an repeated-measures ANOVA with Action (real hand, robotic hand) and Delays (100, 400, 700 ms) as within-subject factors on the reported interval estimates. We observed a significant interaction between Action and Delay (F(2,52)=12.956, p < .001, η2partial = .358). Paired-comparisons indicated that interval estimates were shorter when participants performed theaction with the real hand (210 ms, SD=163) compared to when they used the robotic hand (290 ms, SD=137) for the 100-ms action-tone delay (t(26)=-2.717, p=.012, Cohen’s d=-.523). We also observed that interval estimates were longer when participants used their real hand (624 ms, SD=133) compared to the robotic hand (541 ms, SD=142) for the 700ms action-tone delay (t(26)=3.339, p=.003, Cohen’s d=.643). The difference was not significant for the 500 ms action-tone delay. 

We also observed that for small action-tone delays, the reported interval estimates were smaller for biological movements executed with one’s own hand than for movements executed through the robotic hand. However, this pattern reversed for longer action-tone delays. This is suggestive that different mechanisms drive agency for biological and non-biological movements, an aspect of our findings that warrants further research.

 The perceived control over a movement was not statistically influenced by the action-tone delays.

---

## [Decision Letter · Decision Letter 2]

15 Dec 2020

PONE-D-20-06849R2

How using brain-machine interfaces influences the human sense of agency

PLOS ONE

Dear Dr. Caspar,

Thank you for submitting your manuscript to PLOS ONE. Reviewer 2 has some remaining minor issues that s/he would like to be addressed. Therefore, we invite you to submit a revised version of the manuscript that addresses the points raised.

We look forward to receiving your revised manuscript.

Kind regards,

Jane Elizabeth Aspell, PhD

Academic Editor

PLOS ONE

Reviewers' comments:

Reviewer's Responses to Questions

**Comments to the Author**

1. If the authors have adequately addressed your comments raised in a previous round of review and you feel that this manuscript is now acceptable for publication, you may indicate that here to bypass the “Comments to the Author” section, enter your conflict of interest statement in the “Confidential to Editor” section, and submit your "Accept" recommendation.

Reviewer #1: All comments have been addressed

Reviewer #2: All comments have been addressed

2. Is the manuscript technically sound, and do the data support the conclusions?

Reviewer #1: Yes

Reviewer #2: Yes

3. Has the statistical analysis been performed appropriately and rigorously? 

Reviewer #1: Yes

Reviewer #2: Yes

4. Have the authors made all data underlying the findings in their manuscript fully available?

Reviewer #1: Yes

Reviewer #2: Yes

5. Is the manuscript presented in an intelligible fashion and written in standard English?

Reviewer #1: Yes

Reviewer #2: Yes

6. Review Comments to the Author

Reviewer #1: (No Response)

Reviewer #2: Thank you for the revisions. I only have a few comments that should be addressed.

General Discussion (page 36, lines 22/24): You state that “[agency] did not differ between a condition in which participants used their own hand and a condition in which they used the robotic hand to perform a keypress”

This is not true as stated and should be removed/edited. As before, you should distinguish between the JoA and the FoA here. For the JoA, no comparison between the real hand and the BCI was made. So this statement is simply not supported by any data. One would have to use the same questionnaire for the real hand condition to determine if the JoA differed between the two conditions. Similarly, FoA (via delay estimates) was only compared in study 1, and there you have an interesting interaction between the factors delay and condition.

You could rephrase this by stating that participants report a high/strong JoA for BCIs (but leave out a comparison which was not part of the study design). FoA, in study 1, was comparable in the two conditions. (You could add a Bayesian test to see if there is evidence to “support the null”).

Study 2 – Interval estimates. Please include the actual estimate results here, as for study 1. What are the means ±SDs and do participants distinguish between the levels? These will be relevant values to report for subsequent studies investigating Intentional Binding paradigms for BCIs/prostheses, too.

Page 15, ownership question – “I felt as if the robotic hand belonged to me”. (missing -ed)

Page 27, line 20 – I am not sure that “volition” can be used in this way. Maybe something like a greater cognitive effort/more concentration etc. would be better.

7. PLOS authors have the option to publish the peer review history of their article (what does this mean?). If published, this will include your full peer review and any attached files.

Reviewer #1: **Yes: **Aaron Schurger

Reviewer #2: **Yes: **Oliver A Kannape

---

## [Author Response · Author response to Decision Letter 2]

17 Dec 2020

Reviewer #2: Thank you for the revisions. I only have a few comments that should be addressed.

General Discussion (page 36, lines 22/24): You state that “[agency] did not differ between a condition in which participants used their own hand and a condition in which they used the robotic hand to perform a keypress”

This is not true as stated and should be removed/edited. As before, you should distinguish between the JoA and the FoA here. For the JoA, no comparison between the real hand and the BCI was made. So this statement is simply not supported by any data. One would have to use the same questionnaire for the real hand condition to determine if the JoA differed between the two conditions. Similarly, FoA (via delay estimates) was only compared in study 1, and there you have an interesting interaction between the factors delay and condition.

You could rephrase this by stating that participants report a high/strong JoA for BCIs (but leave out a comparison which was not part of the study design). FoA, in study 1, was comparable in the two conditions. (You could add a Bayesian test to see if there is evidence to “support the null”).

 We have now reported the BF for this comparison, which indeed supported H0 (BF10=.207). We have added this result in the result section and we have also added a note regarding how this value was obtained using JAPS and its default priors. 

 In the general discussion section, we have modified the sentence as follow: “We found in Study 1 that the absence of sensorimotor information was not detrimental for the feeling of agency, as measured through interval estimates, which did not differ between a condition in which participants used their own hand and a condition in which they used the robotic hand to perform a keypress.”

Study 2 – Interval estimates. Please include the actual estimate results here, as for study 1. What are the means ±SDs and do participants distinguish between the levels? These will be relevant values to report for subsequent studies investigating Intentional Binding paradigms for BCIs/prostheses, too.

 Participants who did not discriminate between levels were excluded, as reported in the participant section with the exclusion criteria. Out of 30 participants, 4 did not discriminate with a significant trend the three intervals. So, the 26 remaining participants for this experiment did significantly discriminate between the 3 delays.

 We agree it could be interesting but it is not the same comparison as in Study 1 since here there was not real hand condition, so we did not run the same analysis than in Study 1. We have nonetheless added the means and SDs for the 3 intervals in the “interval estimates” section. 

Page 15, ownership question – “I felt as if the robotic hand belonged to me”. (missing -ed)

 This has been corrected

Page 27, line 20 – I am not sure that “volition” can be used in this way. Maybe something like a greater cognitive effort/more concentration etc. would be better.

 We have replaced the word “volition” by “a greater cognitive effort”

---

## [Decision Letter · Decision Letter 3]

26 Dec 2020

How using brain-machine interfaces influences the human sense of agency

PONE-D-20-06849R3

Dear Dr. Caspar,

We’re pleased to inform you that your manuscript has been judged scientifically suitable for publication and will be formally accepted for publication once it meets all outstanding technical requirements.

Kind regards,

Jane Elizabeth Aspell, PhD

Academic Editor

PLOS ONE

Additional Editor Comments (optional):

Reviewers' comments:

Reviewer's Responses to Questions

**Comments to the Author**

1. If the authors have adequately addressed your comments raised in a previous round of review and you feel that this manuscript is now acceptable for publication, you may indicate that here to bypass the “Comments to the Author” section, enter your conflict of interest statement in the “Confidential to Editor” section, and submit your "Accept" recommendation.

Reviewer #2: All comments have been addressed

2. Is the manuscript technically sound, and do the data support the conclusions?

Reviewer #2: Yes

3. Has the statistical analysis been performed appropriately and rigorously? 

Reviewer #2: Yes

4. Have the authors made all data underlying the findings in their manuscript fully available?

Reviewer #2: Yes

5. Is the manuscript presented in an intelligible fashion and written in standard English?

Reviewer #2: Yes

6. Review Comments to the Author

Reviewer #2: Thank you for taking my feedback into account and for clarifying the statement in the discussion.

7. PLOS authors have the option to publish the peer review history of their article (what does this mean?). If published, this will include your full peer review and any attached files.

Reviewer #2: **Yes: **Oliver A Kannape

---

## [Editor Report · Acceptance letter]

30 Dec 2020

PONE-D-20-06849R3 

How using brain-machine interfaces influences the human sense of agency 

Dear Dr. Caspar:

I'm pleased to inform you that your manuscript has been deemed suitable for publication in PLOS ONE. Congratulations! Your manuscript is now with our production department. 

Kind regards, 

on behalf of

Dr. Jane Elizabeth Aspell 

Academic Editor

PLOS ONE